# Reciprocal antagonism of PIN1-APC/C^CDH1 governs mitotic protein stability and cell cycle entry

Shizhong Ke[1,17], Fabin Dang [2,17], Lin Wang [1,17], Jia-Yun Chen[3,4,17], Mandar T. Naik[5], Wenxue Li [6], Abhishek Thavamani[1], Nami Kim[1], Nandita M. Naik[5], Huaxiu Sui[7], Wei Tang[8], Chenxi Qiu[1,9], Kazuhiro Koikawa[1], Felipe Batalini [1,10], Emily Stern Gatof[1], Daniela Arango Isaza[1], Jaymin M. Patel [1], Xiaodong Wang[11], John G. Clohessy[12], Yujing J. Heng [2], Galit Lahav[3], Yansheng Liu [6,13], Nathanael S. Gray [14], Xiao Zhen Zhou[15] ✉, Wenyi Wei[2] ✉, Gerburg M. Wulf [1] ✉ & Kun Ping Lu [16] ✉

Induced oncoproteins degradation provides an attractive anti-cancer modality. Activation of anaphase-promoting complex (APC/C^CDH1) prevents cell-cycle entry by targeting crucial mitotic proteins for degradation. Phosphorylation of its co-activator CDH1 modulates the E3 ligase activity, but little is known about its regulation after phosphorylation and how to effectively harness APC/C^CDH1 activity to treat cancer. Peptidyl-prolyl cis-trans isomerase NIMA-interacting 1 (PIN1)-catalyzed phosphorylation-dependent *cis-trans* prolyl isomerization drives tumor malignancy. However, the mechanisms controlling its protein turnover remain elusive. Through proteomic screens and structural characterizations, we identify a reciprocal antagonism of PIN1-APC/C^CDH1 mediated by domain-oriented phosphorylation-dependent dual interactions as a fundamental mechanism governing mitotic protein stability and cell-cycle entry. Remarkably, combined PIN1 and cyclin-dependent protein kinases (CDKs) inhibition creates a positive feedback loop of PIN1 inhibition and APC/C^CDH1 activation to irreversibly degrade PIN1 and other crucial mitotic proteins, which force permanent cell-cycle exit and trigger anti-tumor immunity, translating into synergistic efficacy against triple-negative breast cancer.

Mammalian cell division is oriented by sequential activation of proline-directed CDKs, whose dysregulation contributes to unchecked proliferation and cancer[1–6]. Two major regulators required for G1/S transition are the retinoblastoma (RB) protein and the anaphase-promoting complex (APC/C^CDH1), a multi-subunit E3 ubiquitin ligase that is activated by its co-activator CDH1 (encoded in humans by *FZR1*), prevents cell cycle entry by targeting a plethora of crucial mitotic proteins and DNA replication factors for degradation[7–10]. Notably, a functional collaboration between APC/C^CDH1 and RB restrains cell cycle entry, as forced pRB-E2F expression alone is not sufficient to drive cell cycle entry, and an additional loss of APC/C^CDH1 E3 ubiquitin ligase activity is required to trigger proliferation[11,12]. Moreover, the inactivation of APC/C^CDH1, but not activation of pRB-E2F, represents the commitment point of no return for cell-cycle entry[7].

APC/C^CDH1 E3 ligase activity is largely inactivated in various types of human cancer and *CDH1^+/−* animals show increased susceptibility to

spontaneous tumors, supporting that APC/C[CDH1] functions as a tumor suppressor[7,9,13]. Although proline-directed phosphorylation of CDH1 modulates its E3 ligase activity[9,14], it remains unknown whether its activity is subject to further regulation after phosphorylation and how to harness this E3 ligase for cancer therapy. Hence, to effectively reactivate APC/C[CDH1] E3 ligase activity in cancer via inhibiting negative regulators, activating positive regulators, or both, might hold promise as a potential therapeutic approach to induce the degradation of a group of crucial mitotic regulators that promote oncogenesis. This therapeutic approach might become even more significant when the RB pathway is functionally disrupted in many aggressive human cancers, including triple-negative breast cancer (TNBC), which represents the most aggressive subtype of breast cancer with limited treatment options[15–17].

Proline-directed phosphorylation of serine or threonine residues (pSer/Thr-Pro) is a central common signaling mechanism in cancer, governing a wide array of oncoproteins and tumor suppressors[18], which is further regulated by PIN1[19,20]. Originally identified as an essential mitotic regulator[21], PIN1 is a unique proline isomerase that binds and catalyzes *cis-trans* prolyl isomerization of specific Ser/Thr-Pro motifs after phosphorylation to regulate protein structure and function, including protein stability, interaction, and activity[20,22]. However, the exact molecular mechanism underlying PIN1's critical role in cell cycle regulation is not fully understood. Notably, PIN1 is overexpressed and correlated with poor outcomes in most human cancers[19,23]. PIN1 overexpression activates over 70 oncoproteins[19,24,25] and inactivates over 30 tumor suppressors[26–28], thereby promoting cancer, cancer stem cells and an immunosuppressive tumor microenvironment[29,30]. As such, pharmacologic ablation of PIN1 including using the approved drugs offers a unique and promising approach to eradicate aggressive cancer[19,24,30–35]. Strikingly, most effective PIN1 inhibitors discovered to date not only inhibit PIN1 catalytic activity, but also induce its degradation in both *RB*-proficient and *RB*-deficient contexts[24,30,32,35,36]. However, it is still unknown how PIN1 protein stability is physiologically regulated during the cell cycle and how PIN1 inhibitors induce PIN1 protein degradation.

In this study, utilizing proteomic screens, genetic analyses, and structural characterizations, we reveal that APC/C[CDH1], a cell-cycle inhibitor, and PIN1, a cell-cycle promoter, directly interact and negatively regulate each other. Specifically, APC/C[CDH1] is identified as the physiological E3 ligase responsible for controlling PIN1 protein stability and drug-induced PIN1 degradation. Conversely, PIN1 in concert with CDKs regulates APC/C[CDH1] activity to govern the stability of a large number of mitotic proteins, controlling cell cycle entry independently of RB. Mechanistically, the reciprocal antagonism of PIN1-APC/C[CDH1] is mediated by domain-oriented phosphorylation-dependent dual interactions in which PIN1 switches from an upstream inhibitor to a downstream substrate of APC/C[CDH1]. Importantly, the combination of PIN1 inhibitors with CDK4 inhibitors creates a positive feedback loop of PIN1 inhibition and APC/C[CDH1] activation to irreversibly degrade PIN1 and other crucial mitotic proteins, resulting in permanent cell cycle exit and an anti-cancer immune response, which translates into synergistic efficacy against TNBC, including those *RB*-deficient tumors.

## Results

### Cell cycle regulator APC/C[CDH1] is a physiological E3 ligase for PIN1

To understand whether clinically significant PIN1 regulation occurs at the protein or mRNA levels, we first analyzed a dataset by ref. 37 that provided patient-level mRNA, protein and outcomes data. We found a significant correlation between elevated levels of PIN1 protein, but not mRNA, and poor prognosis in breast cancer (BC) patients, irrespective of age, estrogen receptor (ER) status and tumor grade (Fig. 1a, Supplementary Fig. 1a, Supplementary Table 1). Moreover, we evaluated PIN1 protein levels in a Tissue Microarray (TMA) comprising 160 BC samples with patient survival data. Those patients with high PIN1

protein levels had a worse prognosis compared to those with low PIN1 protein levels (Fig. 1b, Supplementary Data 1). To better understand whether PIN1 levels were not only associated with poor prognosis but also developed with treatment resistance, we performed multiplex immunohistochemistry (mIHC) assays to comprehensively examine the protein levels of PIN1 at the single-cell resolution in paired breast cancer tissue specimens collected from 9 patients before treatment with the combination of an estrogen receptor blocker and the CDK4/6-inhibitor Palbociclib, as well as after disease progression on the treatment and subsequent discontinuation of the treatment (Supplementary Fig. 1b, Supplementary Table 2). Pan-cytokeratin (PanCK) was used to distinguish tumor cells from non-tumor cells. As expected, compared to non-tumor cells, PIN1 protein levels were notably higher in tumor cells and even further increased after disease progression (Fig. 1c, Supplementary Fig. 1c) and positively correlated with the proliferation marker Ki67 (Supplementary Fig. 1d). These data show that high PIN1 protein levels are associated with aggressive BC.

As shown previously, effective PIN1 inhibitors, including the highly selective covalent inhibitor Sulfopin[30,36] and the approved drug combination of ATRA and ATO (AApin)[30,32], induce PIN1 degradation, which was rescued by proteasome inhibitors (Supplementary Fig. 1e, f), indicating that PIN1 degradation occurs via the proteasome pathway. To identify the physiological E3 ubiquitin ligase responsible for PIN1 degradation, we used immunoprecipitation coupled with mass spectrometry (IP-MS) to conduct two orthogonal screens to identify the physiological E3 ubiquitin ligase for PIN1 (GST-PIN1 and Flag-PIN1 pull-down). The APC/C E3 ligase complex was the most enriched and common one in both screens (Fig. 1d, Supplementary Fig. 1g, h, Supplementary Data 2). As APC/C activators, CDH1 and CDC20 regulate the activity and substrate specificity of the APC/C E3 ligase[38]. We found that PIN1 had a much higher affinity for CDH1 than its close homologue CDC20 (Fig. 1e, Supplementary Fig. 1i, j). Overexpression of CDH1 enhanced PIN1 inhibitor-induced PIN1 degradation (Supplementary Fig. 1k). Moreover, knockdown of endogenous *CDH1*, but not *CDC20* or other candidate E3 ubiquitin ligases identified by our IP-MS, rescued the PIN1 degradation induced by the PIN1 inhibitor (Fig. 1f, Supplementary Fig. 1l, m, 2a–j). These results show that CDH1 specifically interacts with PIN1 and likely governs its protein stability.

Phosphorylation of CDH1 during the G1/S transition reduces its binding to APC/C, resulting in the inactivation of APC/C[CDH1] and facilitating S phase entry[14,39]. In situ proximity ligation assay (PLA) revealed a substantial augmentation in PLA signals for PIN1-CDH1 when the cells progress into the S/G2 phase, signifying an elevated expression and interactions of PIN1-CDH1 (Supplementary Fig. 3a). To precisely assess the causal relationship between PIN1 and APC/C[CDH1] enzymatic activity in individual cells during the cell cycle, we used non-transformed MCF-10A breast epithelial cells stably expressing the Fucci reporter system with mCherry-conjugated to a Geminin fragment (aa1-110) containing the APC/C[CDH1] degron motif (RXXL), as described[7]. Notably, the promoter region of the reporter construct is unregulated, and the reporter degradation is primarily regulated by APC/C[CDH1] (Supplementary Fig. 3b). In this experimental model, an increase in the reporter signal directly reflects a decrease in APC/C[CDH1] E3 ligase activity and vice versa, allowing for real-time tracking of APC/C[CDH1] activity at the single cell level[7]. When cells were synchronized in G1 followed by releasing back into the cell cycle, PIN1 levels strongly correlated with APC/C-degron reporter levels across the cell cycle, revealing a negative correlation between PIN1 and APC/C[CDH1] activity (Fig. 1g, h). Consistently, in the CPTAC human breast cancer dataset[40,41], the protein levels of Geminin, a well-characterized APC/C[CDH1] substrate, were positively correlated with PIN1 protein abundance, whereas low levels of APC/C[CDH1] were associated with poor prognosis of BC (Supplementary Fig. 3c, d). To further determine the effects of CDH1 on PIN1 expression, we compared the dynamics of PIN1 levels during the cell cycle between non-transformed *CDH1* knockout

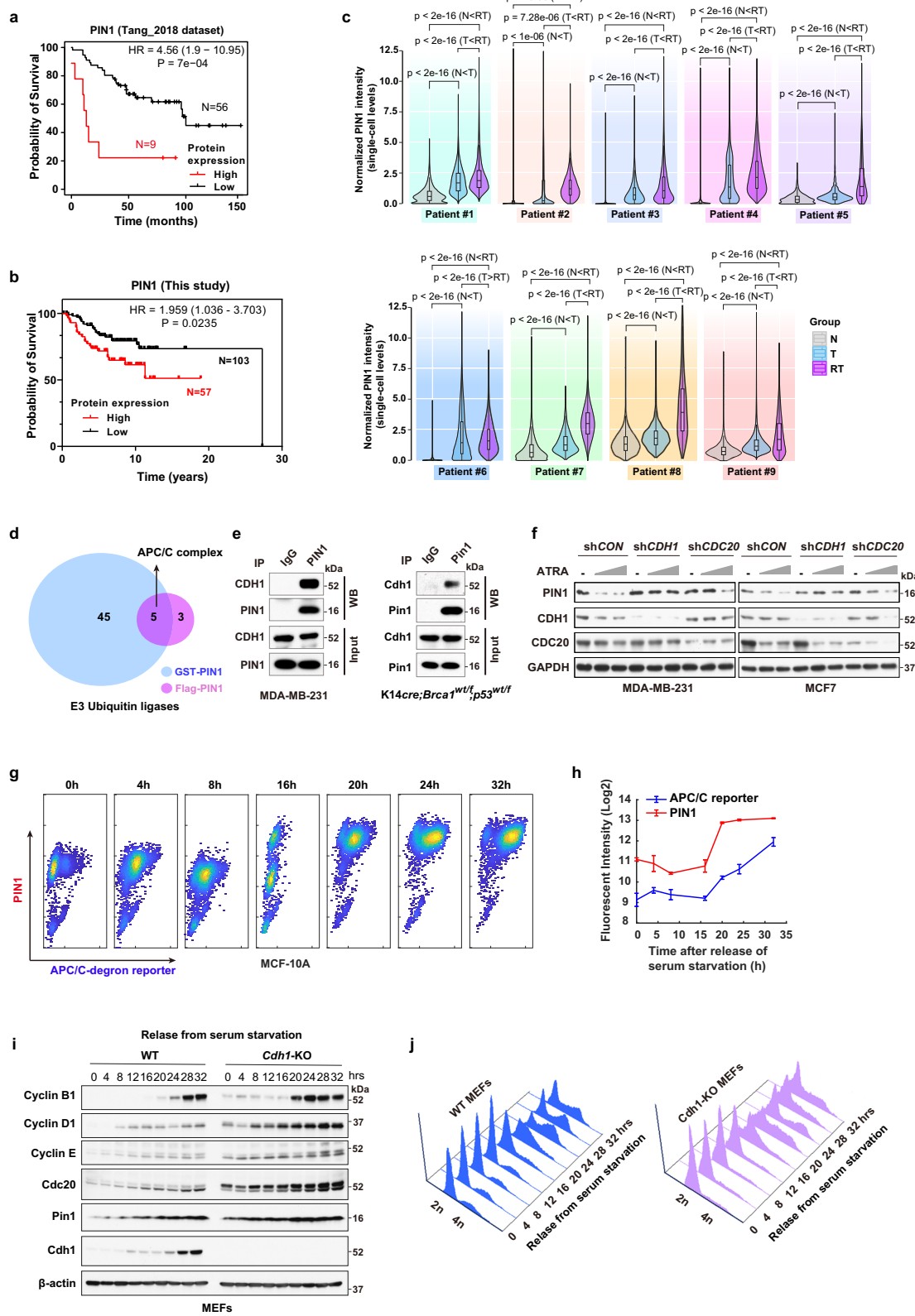

(KO) and wild-type (WT) cells. In WT cells, PIN1 and other mitotic proteins were relatively low in G1 and started to accumulate at the onset of the S phase, coincident with the inactivation of APC/C$^{CDH1}$ (Fig. 1i, j, Supplementary Fig. 3e). In contrast, *CDH1* KO led to stabilization of PIN1 and other mitotic proteins, whereas it showed modest reductions in their mRNA levels (Fig. 1i, j, Supplementary Fig. 3e–g), confirming a negative correlation between PIN1 levels and APC/C$^{CDH1}$

activity. Thus, APC/C$^{CDH1}$ likely is the prime candidate responsible for PIN1 degradation.

**Constitutively active APC/C$^{CDH1}$ targets PIN1 and other mitotic proteins for degradation to provoke cell cycle exit**

APC/C$^{CDH1}$ E3 ligase activity is frequently inhibited in cancer cells, likely due to hyperphosphorylation of CDH1 induced by increased CDKs

**Fig. 1 | Cell cycle regulator APC/C^CDH1 is a physiological E3 ligase for PIN1.**
**a** Overall survival for BC with low and high PIN1 protein abundance in Tang et al. dataset. PIN1-Low, n = 56 patients; PIN1-High, n = 9 patients. **b** Overall survival for BC with low and high PIN1 protein levels in a TMA slide. PIN1-Low, n = 103 patients; PIN1-High, n = 57 patients. Log rank test, p = 7e-04 (**a**), p = 0.0235 (**b**). **c** Violin plots with overlaid box plots showing PIN1 intensity at the single-cell level in the whole BC tissue specimens grouped into Non-tumors (N), Tumors (T) or Refractory Tumors (RT). Box plots, the central line marks 50th percentile, the box edges indicate the 25th and 75th percentiles, and the whiskers stretch to the minima and maxima within 1.5x the IQR (Inter-Quartile Range) from the quartiles. n > 5000 cells. One-way ANOVA followed by Tukey's multiple comparisons test. p values are shown. **d** Venn diagram showing the number of hits obtained from the two IP-MS screens for potential PIN1-related E3 ubiquitin ligases. **e** Co-IP of endogenous CDH1 with endogenous PIN1. MDA-MB-231 cells (left) and transgenic mouse cells (right) were treated with 10 μM MG132 for 12 hrs and precipitated with IgG or anti-PIN1 antibodies. Input is 5% of the total lysates. **f** IB analysis of indicated proteins from shCDH1 or shCDC20 MDA-MB-231 and MCF-7 cells treated with increasing concentrations of ATRA (5 μM, 20 μM) for 3 days. **g** Intensity plots of PIN1 and APC/C reporter in MCF-10A cells synchronized in G1 phase, followed by releasing back into the cell cycle at indicated time points. **h** Fluorescent intensity of PIN1 and APC/C reporter were quantified and log2 transformed across the time courses. Data in graphs are mean ± SD. **i** IB analysis of indicated proteins from WT and *CDH1* KO MEFs synchronized in G1 phase by serum starvation, followed by releasing back into the cell cycle at indicated time points. **j** Cell-cycle profiles corresponding to (**i**) monitored by flow cytometry. The images were representative images from 3 independent experiments (**e, f, i, j**). Source data are provided as a Source Data file.

activity, resulting in decreased APC/C^CDH1 activity[9,14,42]. Thus, we mutated potential CDKs phosphorylation sites that are also potential sites for prolyl isomerization, all of which are located at the N-terminus of CDH1 flanking the C-box (Fig. 2a). Ectopic expression of the phospho-deficient CDH1-7A mutants that can bind to the APC/C core and are constitutively active[14], induced a senescence-like state, indicative of an irreversible G0 state in both *RB*-proficient and *RB*-deficient BC cell lines (Fig. 2b, c, Supplementary Fig. 4a), which indicates an RB-independent cell cycle exit.

Introduction of CDH1-7A mutants induced dose-dependent decrease of PIN1 protein levels and increase of PIN1 ubiquitination (Fig. 2d, e). This observation was further confirmed by in vitro ubiquitination assays, which unequivocally demonstrated the direct ubiquitination of PIN1 by APC/C^CDH1 (Fig. 2f). Inactivating site-directed mutations of PIN1's PPIase enhanced its interaction between CDH1-7A and PIN1, and also promoted CDH1-7A-mediated PIN1 degradation and ubiquitination (Supplementary Fig. 4b–d). Notably, as Flag-PIN1 expression is driven by the constitutively active CMV promoter, Flag-PIN1 degradation is primarily regulated by APC/C^CDH1. Stable expression of CDH1-7A mutants in BC cells reduced protein levels of PIN1 and other mitotic proteins including PLK1, CDC20, Cyclin B1 and Geminin with minimal effects on mRNA levels in both *RB*-proficient and *RB*-deficient cells, which was rescued by the proteasome inhibitor (Fig. 2g, h). Of note, distinct from CDH1-WT, phosphorylation-deficient CDH1-7A mutant preferentially bound to the PIN1 PPIase domain, which may mediate PIN1 degradation through a phosphorylation-independent interaction (Fig. 2i). To explore this possibility, we examined if PIN1 contains a destruction box (D-box) since most APC/C^CDH1 substrates contain a D-box with the conserved consensus RXXL sequence (X presents any amino acid)[43]. Indeed, PIN1 has a putative D-box within its PPIase domain (Supplementary Fig. 4e), recognizable by the CDH1 WD40 domain. To confirm this possibility, we performed co-IPs using a point mutation W34A in the PIN1 WW domain and a D-box mutation RLAA in the PIN1 PPIase domain as well as their dual mutations. Indeed, the W34A mutation, which disrupts the ability of WW domain bind to a pSer/Thr-Pro motif[14], prevented the PIN1 interaction with CDH1-WT (Supplementary Fig. 4f), whereas, the RLAA mutation within the PPIase domain did not interfere with the interaction with CDH1-WT (Supplementary Fig. 4f). In contrast, PIN1-W34A still interacted with CDH1-7A, indicative of a phosphorylation-independent interaction (Supplementary Fig. 4g). Only RLAA mutations in the PIN1 PPIase domain prevented the PIN1 interaction with CDH1-7A (Supplementary Fig. 4g), demonstrating a D-box-mediated interaction. In keeping with these findings, the RLAA mutation in PIN1, but not the W34 mutation, conferred resistance to CDH1-7A-mediated PIN1 degradation and ubiquitination (Fig. 2j, k). Our data also demonstrated that this D-box mutant of PIN1 maintained the enzymatic activity of PIN1 (Supplementary Fig. 4h) and was more stable throughout the cell cycle (Supplementary Fig. 4i). Thus, active APC/

C^CDH1 targets PIN1 for degradation via the D-box localized in the PPIase domain of PIN1.

To gain structural insight into the interaction between the D-box of PIN1 and the WD40 domain of CDH1, we generated a docking model of the complex using HADDOCK, based on the available structures of PIN1 (PDB: 1PIN)[45] and the WD40 domain of CDH1 (PDB: 4UI9_R). Molecular modeling suggested that the formation of electrostatic interaction between R119 of the PPIase domain with CDH1 residues D180 and E465 drove a conformational change of the second β-strand of the PIN1 PPIase domain. This change led to L122 of the PPIase domain swinging in the hydrophobic pocket formed by CDH1 residues L179, A181 and L467, while residues K117 and G123 of the PPIase domain formed hydrogen bonds with the backbone of V219 and the side chain of W212, respectively (Fig. 2l, Supplementary Fig. 4j). These modeling results further support the notion that the D-box in the PIN1 PPIase domain is critical for CDH1 to interact with PIN1 and target PIN1 for degradation. Thus, APC/C^CDH1 is likely the physiological E3 ubiquitin ligase for PIN1, and its activity is primarily inhibited by phosphorylation of CDH1 in cancer cells. These data support that constitutively active APC/C^CDH1 acts as a tumor suppressor, which targets PIN1 and other mitotic proteins for degradation to provoke cell cycle exit independent of RB-mediated signaling.

## Loss of *PIN1* reactivates APC/C^CDH1 and destabilizes mitotic proteins

Given the negative relationship between PIN1 protein levels and APC/C^CDH1, to further explore their causal relationship, we knocked out *PIN1* to examine whether PIN1 may reciprocally inhibit APC/C^CDH1 activity. *PIN1* KO dramatically reduced cell viability in long-term clonogenic assays in both *RB*-proficient and *RB*-deficient BC cell lines (Fig. 3a), indicating *PIN1* KO induced RB-independent effects on tumor cells growth. The quantitative proteomic analysis of *PIN1* KO in the MDA-MB-231 cells showed that *PIN1* KO exhibited noticeable effects on cell cycle and G1/S transition[32] (Fig. 3b). Notably, *PIN1* KO de-stabilized mitotic proteins, indicative of APC/C^CDH1 activation, resulting in prolonged G0/G1 phases (Fig. 3c–f). To more precisely define the effects of *PIN1* KO on APC/C^CDH1 activation kinetics at the single cell level, we analyzed changes in the reporter levels during the cell cycle by tracking MCF-7 wild-type or *PIN1* KO single cells over 72 h and found that *PIN1* KO triggered the reactivation of APC/C^CDH1, accompanied by G0/G1 arrest, was evidenced by a significant decrease of the APC/C-reporter intensity and cell division. Alternatively, a small number of cells experienced extended G2/M phases, as PIN1 has other substrates in the cell cycle[19] (Fig. 3g, h). To further separate transcriptional regulation from post-transcriptional, we conducted the cycloheximide (CHX) chase assay and found that *PIN1* KO markedly reduced the half-lives of mitotic proteins (Fig. 3i). Thus, loss of PIN1 enhances APC/C^CDH1 activity, resulting in the destabilization of mitotic proteins and the induction of cell cycle arrest,

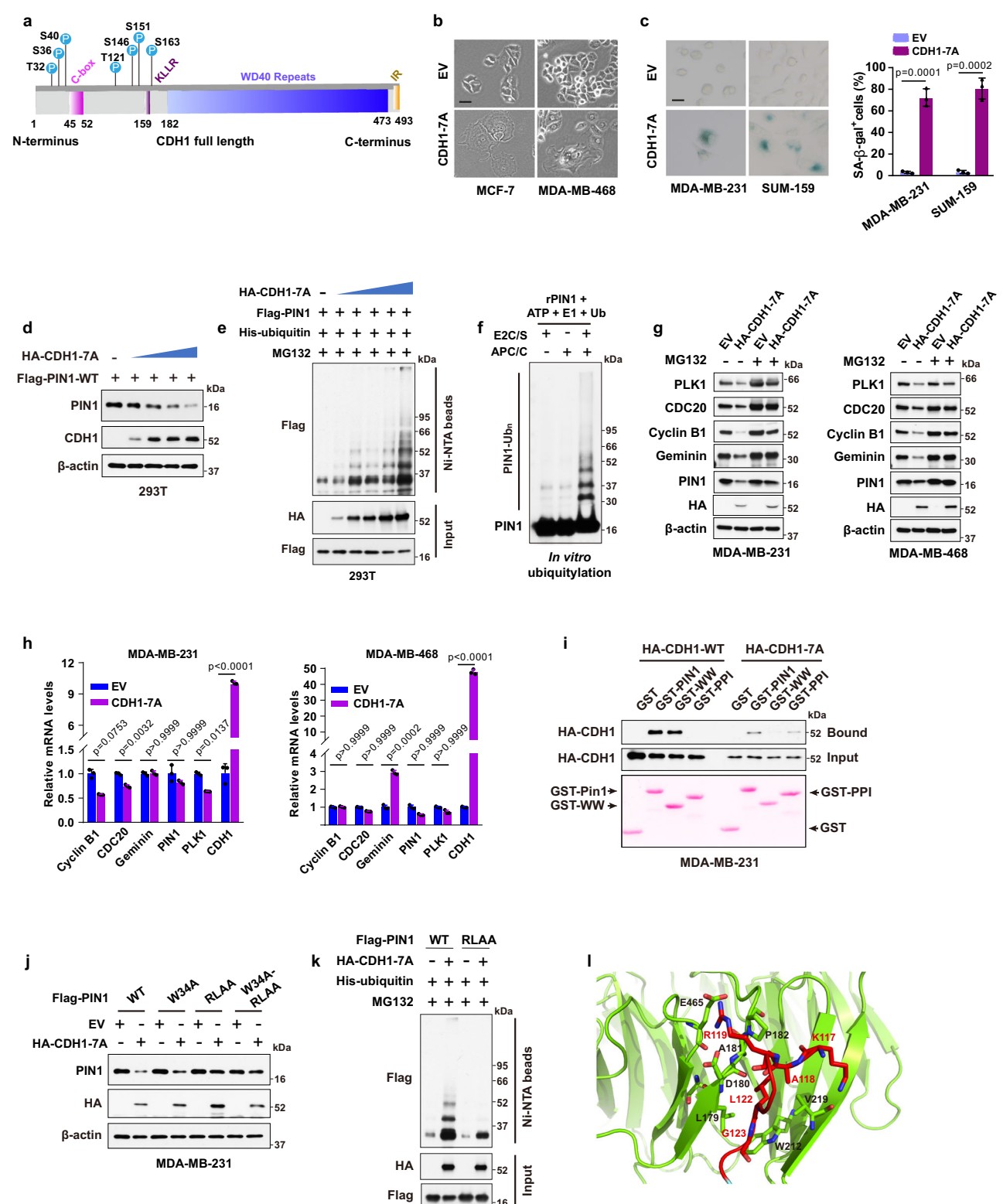

indicating that PIN1 likely is a negative regulator of APC/C$^{CDH1}$ E3 ligase.

## PIN1 catalyzes *trans* to *cis* isomerization of the pS163-P motif in CDH1 to prevent CDH1 dephosphorylation

We then investigated the underlying mechanism of the reciprocal regulation between PIN1 and APC/C$^{CDH1}$. In the APC/C$^{CDH1}$ complex, CDH1 can be directly phosphorylated and inactivated by several

kinases, resulting in its dissociation from the APC/C core complex[7,9]. We found that CDK2 and CDK4, but not CDK6, strongly interacted with CDH1 (Fig. 4a, Supplementary Fig. 5a). Moreover, CDK4 specifically phosphorylated CDH1 at S163 and significantly increased PIN1 and other mitotic protein abundance in serum-free conditions (Fig. 4b, Supplementary Fig. 5b). By performing a site-directed mutation analysis of S163 in CDH1, we found that compared to cells with the CDH1-WT, those with the S->A phospho-deficient mutation of CDH1-S163

**Fig. 2 | Constitutively active APC/C^CDH1 targets PIN1 and other mitotic proteins for degradation to provoke cell cycle exit. a** Domain architecture of CDH1 highlighting potential phosphorylation sites for CDKs. **b** Morphology of MCF-7 and MDA-MB-468 cells stably expressing CDH1-7A. **c** Left, SA-β-gal stain in MDA-MB-231 and SUM-159 cells expressing CDH1-7A. Scale bars, 20 μm. Right, Quantification of SA-β-gal positive cells. $n = 3$ independent experiments. Data in graphs are mean ± SD and analyzed by unpaired two-sided $t$-test. EV vs. CDH1-7A, $p = 0.0001$ (MDA-MB-231), $p = 0.0002$ (SUM-159). **d** IB analysis of indicated proteins from 293T cells transfected with Flag-PIN1 and a gradient of CDH1-7A constructs. **e** IB analysis of ubiquitination of Flag-PIN1 from 293T cells transfected with the indicated constructs for 48 hours and treated with 2 μM MG132 for 12 h and pulled down by Ni-NTA agarose. **f** In vitro ubiquitination assay using APC/C from cells in G1 phase and recombinant PIN1. E2C/S, E2 enzymes UBE2C and UBE2S. **g** IB analysis of indicated proteins from MDA-MB-231 or MDA-MB-468 cells stably expressing CDH1-7A and

treated with 10 μM MG132 for 12 h. **h** RT-PCR analysis of indicated mRNA of WT and CDH1-7A MDA-MB-231 and MDA-MB-468 cells. $n = 3$ independent experiments. Data in graphs are mean ± SD and analyzed by two-way ANOVA followed by Bonferroni's multiple comparisons test. $p$ values are shown. **i** GST-PIN1 pull-down precipitates derived from MDA-MB-231 cells stably expressing HA-CDH1 or HA-CDH1-7A, treated with 10 μM MG132 for 12 h. **j** IB analysis of indicated proteins from MDA-MB-231 cells stably co-expressing CDH1-7A or empty vector (EV) in the presence of Flag-PIN1 and its mutants. **k** IB analysis of ubiquitination of Flag-PIN1 and its RLAA mutant from 293T cells transfected with the indicated constructs for 48 h and treated with 2 μM MG132 for 12 h and pulled down by Ni-NTA agarose. **l** Docking model of the interaction between D-box of PIN1 and WD40 domain of CDH1. The images were representative images from 3 independent experiments (**b–g**, **i–k**). Source data are provided as a Source Data file.

---

exhibited G1 arrest with enhanced APC/C activity, as evidenced by the reduced fluorescence intensity of APC/C-degron reporter (Fig. 4c, Supplementary Fig. 5c). In contrast, cells carrying the S- > E phospho-mimic mutation of CDH1-S163 displayed decreased APC/C activity (Fig. 4c). Moreover, cells with the CDH1-S163E mutation notably impeded the CDK4 inhibitor-induced reduction in mitotic proteins (Fig. 4d, Supplementary Fig. 5d). Since another critical target of CDK4 is the RB protein, to confirm whether CDK4-mediated APC/C^CDH1 inactivation was independent of RB-mediated transcriptional signaling, we knocked down *RB* and its two homologs (Supplementary Fig. 5e, f). Although *RB* loss confers a certain level of resistance to CDK4 inhibitors, knockdown of *RB* alone couldn't fully overcome cell cycle arrest induced by CDK4 inhibitors, as shown before[11]. Indeed, the inhibition of CDK4 by relatively higher doses of Palbociclib still potently decreased the levels of PIN1 and other mitotic proteins even in *RB*-deficient cells (Supplementary Fig. 5e, f). These results support that CDH1 is an alternative CDK4 substrate and CDK4-mediated phosphorylation directly inactivates APC/C^CDH1, leading to accumulation of mitotic proteins.

PIN1 specifically recognizes pSer/Thr-Pro motifs and catalyzes sequence-specific phosphorylation-dependent proline isomerization[22,44]. We found that the CDH1-S163A mutation, abolishing phosphorylation at this location, reduced its interaction with PIN1, whereas the phosphomimic CDH1-S163E mutation restored the binding to PIN1 (Fig. 4e, Supplementary Fig. 5g). To directly visualize PIN1 binding and isomerization of phosphorylated CDH1, we synthesized a CDH1-pS163 peptide and mapped the interaction with PIN1 using nuclear magnetic resonance (NMR). The perturbation data indicated that CDH1-pS163 peptide binds to the WW domain with strong affinity (Fig. 4f). PIN1 residue R17 showed the most substantial perturbation, and along with residues S18, Y23, W34 and E35 formed a continuous patch that interacts with phosphoserine, pS163, and the adjoining proline, P164. Our experiment-guided model suggests that the phosphate group from pS163 has a charge: charge interaction with R17, while P164 stacks in the pocket formed by Y23 and W34 (Fig. 4g, h, Supplementary Fig. 5h), which was confirmed by GST pull-down assays using PIN1 point mutations (Supplementary Fig. 5i). Interestingly, weak perturbation was observed in the PPIase domain active site, which may mediate PIN1-catalyzed isomerization of CDH1-pS163 peptide. To confirm such isomerization, we used specific $^{13}C$ enrichment of the P164 and a 2D-$^{13}C$ HSQC spectrum to directly measure the P164 isomerization states (Supplementary Fig. 5j). Our results showed that 7% *cis*-P164 isomer was present in the free peptide, but PIN1-catalyzed isomerization doubled this population to 14.2% (Fig. 4i). This is significant as an enzyme usually does not change the equilibrium of the reaction, indicating that PIN1 preferentially catalyzes the *trans* to *cis* isomerization of pS163-P motif in CDH1, which may lead to increase of phosphorylated CDH1 because CDH1-specific phosphatase does not engage with *cis* proline[46]. To support these NMR findings, we examined the phosphorylation status of CDH1-S163 using Mass spectrometry. We

found that in *PIN1* KO cells, the phosphorylation levels of CDH1-S163 decreased by nearly two-fold compared to WT cells (Fig. 4j, Supplementary Data 3). Furthermore, our data revealed that *PIN1* KO significantly increased the binding affinity of CDH1 to the APC/C complex in CDH1-WT cells, but not in CDH1-S163A or -S163E mutant (Fig. 4k), which led to an increase in APC/C^CDH1 activity (Supplementary Fig. 5k). Thus, PIN1 binds to and catalyzes the *trans* to *cis* prolyl-isomerization of the pS163-P motif in CDH1, thereby stabilizing phosphorylated CDH1 and rendering APC/C^CDH1 inactive (Fig. 4l).

Collectively, the above data demonstrate domain-oriented, phosphorylation-dependent dual interactions that mediates reciprocal antagonism of PIN1 and APC/C^CDH1. On one hand, when phosphorylated by CDKs at the late G1, CDH1 binds to the WW domain of PIN1, which catalyzes *trans* to *cis* isomerization of the pS163-P motif in CDH1 to prevent CDH1 dephosphorylation, thereby rendering APC/C^CDH1 inactive, leading to accumulation of mitotic proteins to ensure S-phase entry. On the other hand, when de-phosphorylated in the G0/G1 phase, APC/C^CDH1 becomes functionally active. The substrate-binding domain of CDH1 recognizes the degron motifs of PIN1 and other mitotic proteins, targeting them for degradation to prevent S-phase entry. This reciprocal inhibition between PIN1 and APC/C^CDH1 offers a compelling molecular rationale for combining therapies targeting both proteins.

## Pharmacologic inhibition of PIN1 and CDK4 synergistically and irreversibly reactivates APC/C^CDH1 to induce degradation of PIN1 and other mitotic proteins

The above results suggest a therapeutic opportunity of targeting both PIN1 and CDKs to synergistically and irreversibly reactivate APC/C^CDH1 E3 ligase activity to target many crucial mitotic proteins for degradation. We use CDK4 inhibitors instead of CDK2 inhibitors to reactivate APC/C^CDH1 in our following experiments, as CDK4 inhibitors are highly selective and approved by FDA[47]. To test the possibility, we first examined the effects of Palbociclib on the fate of phosphorylated CDH1 in WT and *PIN1* knockout cells. Palbociclib dramatically reduced CDH1 phosphorylation and promoted the binding of CDH1 to APC/C complex in both *RB*-proficient and *RB*-deficient cells, which was enhanced by *PIN1* knockout (Fig. 5a, Supplementary Fig. 6a). Similarly, PIN1 inhibition, which prevents of *trans* to *cis* isomerization, led to de-phosphorylation of CDH1 (Supplementary Fig. 6b), presumably by the *trans*-selective CDH1 phosphatase[46], reducing CDH1 binding to the PIN1 WW domain, but increasing its binding to the PIN1 PPIase domain (Supplementary Fig. 6c, d). This change of the binding mode might switch PIN1 from being an upstream inhibitor to a downstream substrate of APC/C^CDH1. Indeed, overexpression of PIN1 potently inhibited APC/C^CDH1 E3 ligase activity, as determined by elevated levels of mitotic proteins in the absence or presence of the CDK4 inhibitor (Supplementary Fig. 6e). By contrast, PIN1 inhibitors led to significant decreases of PIN1 and other mitotic proteins in a time-dependent manner, corresponding to an increase of G0/G1 populations (Fig. 5b, Supplementary Fig. 6f–i). To maximize the effects of PIN1 inhibitors,

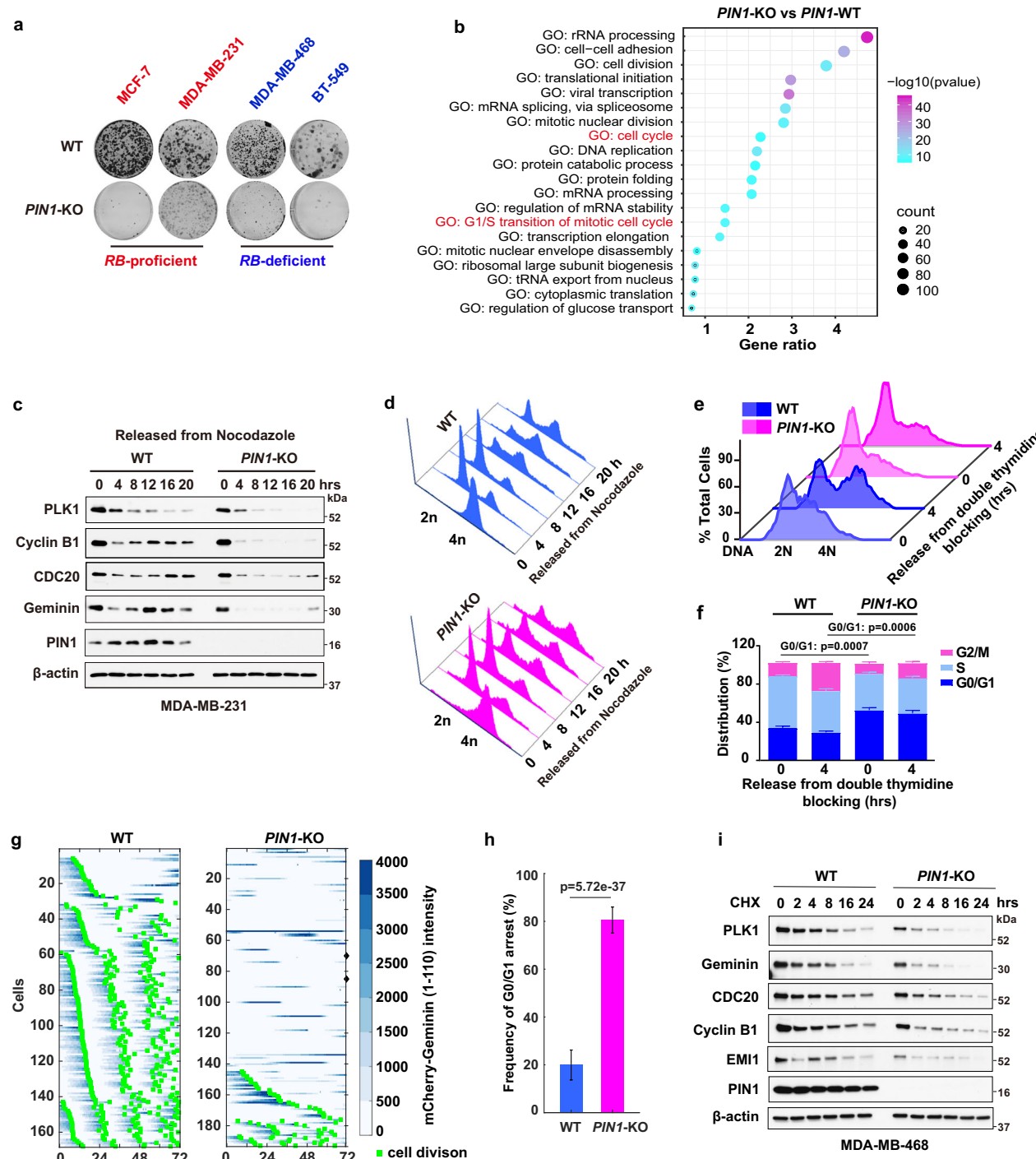

**Fig. 3 | Loss of PIN1 reactivates APC/C^CDH1 and destabilizes mitotic proteins.**
**a** Long-term colony-formation assay of indicated BC wild-type and *PIN1* KO cell lines. Cells were grown for about 2 weeks, fixed and stained with crystal violet.
**b** Gene Ontology (GO) enrichment analysis was applied to proteomics of *PIN1* KO versus WT MDA-MB-231 cells. Color codes for *p*-value and symbol size codes for the ratio of proteins related to specific GO term/total number of proteins significantly altered. Data were obtained from Kozono et al. 2018 and analyzed by one-sided hypergeometric test. **c** IB analysis for indicated proteins derived from WT and *PIN1* KO MDA-MB-231 cells synchronized in M phase by nocodazole and then released back into the cell cycle for the indicated time. **d** Cell-cycle profiles of WT (blue) and *PIN1* KO (pink) in (**c**) as determined by FACS. **e** DNA contents were measured by FACS in WT and *PIN1* KO MDA-MB-231 cells synchronized at the G1/S boundary by double thymidine block and then released back into the cell cycle for 4 h. **f** Cell cycle phase distribution of WT and *PIN1* KO MDA-MB-231 cells from (**e**). n = 3

independent experiments. Data in graphs are mean ± SD and analyzed by unpaired two-sided *t*-test. WT vs. *PIN1*-KO, *p* = 0.0007 (0 h), *p* = 0.0006 (4 h). **g,** Tracking cell division (green square) and cell death (black rhomboid) at the single cell level. Asynchronous cultures of MCF-7 WT and *PIN1* KO cells expressing the APC-degron reporter were followed for 72 h for single cell expression of mCherry-Geminin (shades of blue). **h** Frequency of G0/G1 arrest (ratio of G0/G1 arrested cells to total cells) in WT and *PIN1* KO MCF-7 cells stably expressing the APC/C-degron reporter from (**g**). WT, n = 168 cells; *PIN1* KO, n = 191 cells. The error bar indicates 95% confidence interval determined by bootstrapping. **i** Cycloheximide (CHX) chase assay for indicated proteins derived from WT and *PIN1* KO MDA-MB-468 cells treated with 50 μg/ml CHX for the indicated time. The images were representative images from 3 independent experiments (**a**, **c**–**e**, **i**). Source data are provided as a Source Data file.

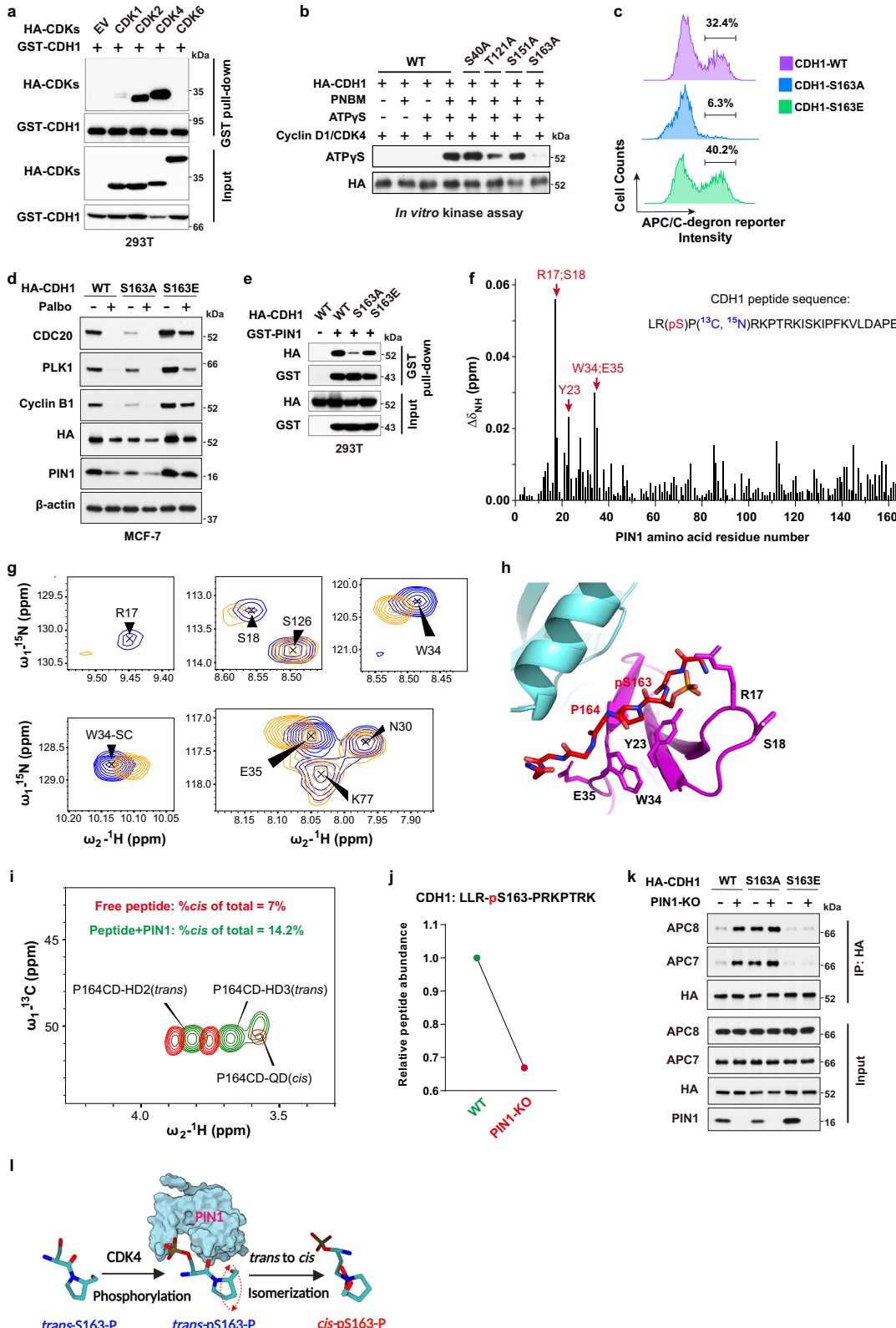

i.e., dramatic PIN1 degradation, we applied a 3-day treatment for in vitro experiments.

To further confirm these results, we measured APC/C$^{CDH1}$ activation kinetics in single cells upon the CDK4 or PIN1 inhibitors treatment by analyzing changes of the degron reporter levels. CDK4 inhibition caused an immediate reactivation of APC/C$^{CDH1}$ reflected by decreased reporter intensity along with G0/G1 arrest (Fig. 5c, middle panel). While both PIN1 inhibitors (Sulfopin and AApin) led to an extended G2/M phase and cell death in some cells, they eventually caused a reactivation of APC/C$^{CDH1}$ to induce G0/G1 arrest (Fig. 5c, d, right panel, Supplementary Fig. 6j, Supplementary Movie 1–4). Significantly, the combination of the CDK4 inhibitor and the PIN1 inhibitors markedly reduced the expression and interactions of PIN1-CDH1 and CDK4-CDH1 (Fig. 5e, f), and induced striking activation of APC/C$^{CDH1}$ (Supplementary Fig. 6k).

**Fig. 4 | PIN1 catalyzes _trans_ to _cis_ isomerization of the pS163-P motif in CDH1 to prevent CDH1 dephosphorylation. a** 293T cells were transfected with indicated constructs for 36 hrs. Input is 5% of the total lysates used in IP. **b** In vitro kinase assay showing that CDK4 phosphorylates CDH1 at Ser163. **c** FACS analysis of APC/C-degron reporter levels in MCF-7 cells expressing either WT CDH1, S163A or S163E mutants CDH1. **d** IB analysis of indicated proteins derived from MCF-7 cells expressing either WT CDH1, S163A or S163E mutants CDH1 and treated with 1 μM palbociclib for 3 days. **e** IB analysis of GST pull-down precipitates derived from 293T cells transfected with GST-PIN1 and HA-CDH1 mutants for 36 h. **f** NMR analysis of phosphorylated peptide bound to PIN1. Average chemical shift perturbation in PIN1 backbone amide resonances on the binding of the CDH1 phosphopeptide. **g** Overlay of two-dimensional (2D) ¹H-¹⁵N Heteronuclear single quantum coherence (HSQC) spectrum from the backbone of R17, S18, W34, and E35, and the W34 sidechain of ¹⁵N-labeled PIN1 (blue) and its complex with the CDH1

phosphopeptide (orange). **h** HADDOCK model demonstrating putative interaction between the CDH1 phosphopeptide shown as red sticks and PIN1 WW (magenta) and PPIase domain (cyan; PDB: 1PIN). **i** Overlay of ¹³C-HSQC spectra acquired on free peptide (red) and its complex with PIN1 (green). The peak volumes were used to derive isomer population estimates. **j** DIA-MS analysis of the relative abundance of the peptide containing phosphorylation site of CDH1-S163 derived from WT or _PIN1_ KO MCF-7 cells stably expressing HA-CDH1. **k** IB analysis of indicated immuno-precipitates derived from WT or _PIN1_ KO MCF-7 cells expressing either WT CDH1, S163A or S163E mutants CDH1 and pulled down by anti-HA antibody. Input is 5% of the total lysates used in IP. **l** Schematic diagram illustrating PIN1-catalyzed _trans_ to _cis_ prolyl-isomerization of the CDH1-pS163-P motif. The images were representative images from 3 independent experiments (**a**–**e**, **k**). Source data are provided as a Source Data file.

---

It has been previously reported that Cyclin D, RB, CDK2 or EMI1 either directly or indirectly regulate APC/C^CDH1 activity[7,9,13]. However, knockout or knockdown of each of these genes did not block PIN1 inhibitor- and CDK4 inhibitor-induced activation of APC/C^CDH1, although triple _cyclin D_ knockout blocked the effects of CDK4 inhibitor (Supplementary Fig. 5e, f, 7a–d). Instead, loss of _CDK2_ or _EMI1_ enhanced CDK4 inhibitor- and PIN1 inhibitor-induced activation of APC/C^CDH1 (Supplementary Fig. 7c, d). By contrast, knockout of _CDH1_ largely blocked PIN1 inhibitor- and CDK4 inhibitor-induced activation of APC/C^CDH1, evident from the increased levels of mitotic proteins, but not mRNA levels, in both _RB_-proficient and _RB_-deficient BC cell lines (Fig. 5g, h, Supplementary Fig. 7e–j), supporting the direct role of PIN1 and CDK4 in the regulation of APC/C^CDH1. To further separate transcriptional regulation from post-translational one, we measured mitotic protein stability by the cycloheximide chase assay. Notably, _CDH1_ KO significantly prolonged the protein half-lives of PIN1 and other mitotic proteins with or without PIN1 and CDK4 inhibitors treatment (Fig. 5i, Supplementary Fig. 7k). Combining the PIN1 inhibitor with the CDK4 inhibitor caused even more pronounced PIN1 poly-ubiquitination, which was diminished by _CDH1_ KO (Fig. 5j).

These above results not only uncover that CDK4-mediated phosphorylation in concert with PIN1-catalyzed _trans_-to-_cis_ isomerization inactivates APC/C^CDH1, leading to accumulation of PIN1 and mitotic proteins and S-phase entry (Fig. 5k), but also suggest that combined inhibition of PIN1- and CDK4 irreversibly reactivates APC/C^CDH1 to induce the degradation of PIN1 and other mitotic proteins and permanent cell-cycle exit (Fig. 5l). This combination creates a positive feedback loop of PIN1 inhibition and APC/C^CDH1 activation, which may lead to synergistic anti-tumor efficacy.

### PIN1 inhibitors synergize with CDK4 inhibitors against TNBC in human cells and immune-compromised mouse models

Selective CDK4/6 inhibitors have emerged as effective ER + BC treatment but only have limited efficacy in TNBC, in which _RB_ gene loss occurs predominantly[48,49]. Besides, most patients achieved only partial remissions and eventually experienced disease progression, indicative of primary and secondary resistance mechanisms, such as genetic alterations in the RB signaling pathway[50–52]. Hence, more effective CDK4/6 inhibitor combinations for ER+ or TNBC, independent of RB-mediated cell cycle regulation, are urgently needed.

Our findings demonstrate that inhibition of PIN1- and CDK4 cooperatively and irreversibly reactivates APC/C^CDH1 to induce robust degradation of PIN1 and other mitotic proteins, suggesting potential synergistic effects of combining the PIN1 inhibitors with the CDK4 inhibitors to facilitate the efficacy of PIN1 inhibitors or CDK4 inhibitors in TNBC, especially in _RB_-deficient tumors. Indeed, the combination of CDK4 inhibitors and PIN1 inhibitors resulted in a synergistic anti-proliferative effect in both _RB_-proficient and _RB_-deficient TNBC cell

lines (Fig. 6a–f, Supplementary Fig. 8a–f). However, _CDH1_ KO strongly reduced the efficacy of PIN1 and CDK4 inhibition (Supplementary Fig. 8g, h). These data suggest that PIN1 inhibitors synergize with CDK4 inhibitors, resulting in effective response in TNBC irrespective of _RB_ status.

To further confirm whether our in vitro findings can be translated in vivo for clinically relevant anti-tumor treatment, we used _RB_-proficient TNBC patient-derived orthotopic xenograft (PDOX) and _RB_-deficient MDA-MB-468 xenograft mouse models to evaluate the anti-tumor efficacy of the two-drug regimen. To this end, although Sulfopin and Palbociclib had limited single-agent efficacy in the _RB_-proficient mouse model, the combination of Sulfopin and the two CDK4 inhibitors nearly entirely suppressed tumor growth, with 2/7 PDOX tumors achieving a complete remission (Fig. 6g, h, Supplementary Fig. 9a, b). These results were supported by immunofluorescent analysis of tumors revealing a significant decrease in the levels of PIN1 and other mitotic proteins in tumors from mice treated with Sulfopin and CDK4 inhibitors, consistent with APC/C^CDH1 activation (Fig. 6i, Supplementary Fig. 9c, d). It was reported that Abemaciclib inhibits kinases other than CDK4/6 including CDK2[47], which is also an APC/C^CDH1 upstream kinase. Therefore, Abemaciclib has the therapy advantage over Palbociclib. As for safety and tolerability, the two-drug regimen showed no bone marrow suppression and was well tolerated with maintenance of body weight (Supplementary Fig. 9e–n). The observed low toxicity is consistent with the findings that Pin1 null mice develop normally[19,53]. In the _RB_-deficient MDA-MB-468 xenograft mouse model, Palbociclib alone didn't significantly suppress tumor growth, as expected, but the combination of Sulfopin and Palbociclib elicited a complete inhibition of tumor growth (Fig. 6j, k). Similarly, immunofluorescent analysis of tumors revealing the diminished signals of PIN1 and Geminin in tumors from mice upon combination treatment, indicating degradation of mitotic proteins due to constant APC/C^CDH1 activation (Fig. 6l).

To further evaluate the anti-tumor efficacy of the two-drug regimen in allogeneic immune-compromised mouse models, we generated two cohorts of genetically engineered TNBC mouse models: K14_cre; Brca1wt/f; p53wt/f_ BT1 and K14_cre; Brca1wt/f; p53wt/f_ BT3. They were _Rb_-deficient and _Rb_-proficient respectively and resembled aggressive human TNBC with the growth of highly proliferative and poorly differentiated mammary carcinomas in allogeneic immune-compromised recipients[54] (Supplementary Fig. 10a, b, Supplementary Table 3). The transgenic tumors were transplanted orthotopically in nude mice to generate allogeneic tumor mouse model. We also observed highly effective anti-tumor activity of the combination treatment in both _Rb_-proficient and _Rb_-deficient mouse tumor model (Fig. 6m, n). Thus, PIN1 inhibitors efficiently synergize with CDK4 inhibitors against TNBC in human cells and immune-compromised mouse models.

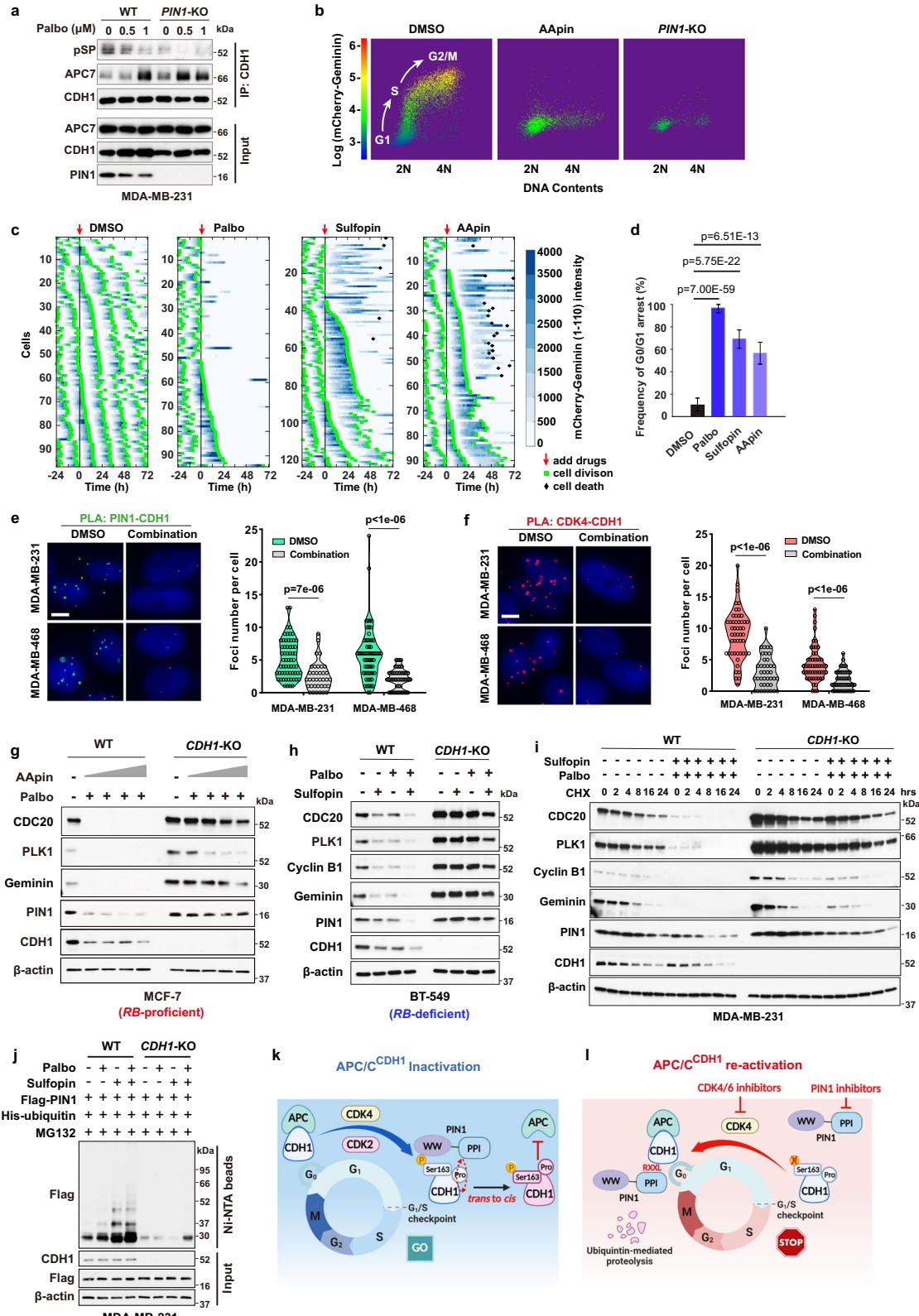

## Combination of PIN1- and CDK4-inhibitors achieves synergistic anti-tumor immunity and efficacy against *RB*-proficient or *RB*-deficient TNBC in immune-competent mouse models

Inhibition of either CDK4 or PIN1 has been reported to trigger anti-tumor immunity[30,55]. We therefore sought to study the combination therapy in immune-competent mouse models, more clinically relevant to human patients. K14*cre; Brca1wt/f; p53wt/f* BT3 transgenic tumors

were orthotopically transplanted in FVB mice to generate syngeneic tumor mouse model. Notably, syngeneic TNBC models are historically much more challenging to treat with chemotherapy or targeted agents than PDOX and allograft models in immune-compromised hosts[56]. Indeed, CDK4 inhibitors or PIN1 inhibitors alone had limited efficacy in this hard-to-treat, highly undifferentiated, and proliferative murine TNBC (Fig. 7a–d). However, their combination was highly effective and

**Fig. 5 | Pharmacologic inhibition of PIN1 and CDK4 synergistically and irreversibly reactivates APC/C^CDH1 to induce degradation of PIN1 and other mitotic proteins. a** IB analysis of immunoprecipitates from WT and *PIN1*-KO MDA-MB-231 cells treated with Palbociclib. **b** Dot plots of FACS for *PIN1*-KO or WT MCF-7 cells stably expressing the APC/C-degron reporter and treated with AApin (1.5 μM ATO + 15 μM ATRA) for 3 days. Dots color, reporter levels. **c** Tracking cell division and cell death of MCF-7 cells in response to Palbociclib (4 μM), Sulfopin (10 μM) or AApin (1.5 μM ATO + 15 μM ATRA). **d** Frequency of G0/G1 arrest in MCF-7 cells from (**c**). *n* > 90 cells. The error bar indicates 95% confidence interval determined by bootstrapping. Detection of endogenous PIN1-CDH1 (**e**) and CDK4-CDH1 interactions (**f**) by PLA in the indicated cells treated with combination of 1 μM Palbociclib and 10 μM Sulfopin for 3 days. Nucleus were stained by DAPI. Scale bars, 5 μm. (Right) Quantification of PLA signals. *n* > 30 cells. Data are analyzed by unpaired

two-sided *t*-test. *p* values are shown (**d**–**f**). **g** IB analysis for indicated proteins from WT and *CDH1*-KO MCF-7 cells treated with combination of 1 μM Palbociclib and AApin (ATO (0.5, 1, 1.5, 2 μM) + ATRA (5, 10, 15, 20 μM)) for 3 days. **h** IB analysis for indicated proteins derived WT and *CDH1*-KO BT-549 cells treated with 5 μM Sulfopin, 2.5 μM Palbociclib or their combination for 3 days. **i** CHX assay for indicated proteins from WT and *CDH1*-KO MDA-MB-231 cells pre-treated with combination of 10 μM Sulfopin and 1 μM Palbociclib for 36 h followed by 50 μg/ml CHX for the indicated time. **j** IB analysis of ubiquitinated PIN1 from WT and *CDH1*-KO MDA-MB-231 cells treated with 1 μM Palbociclib and 10 μM Sulfopin for 3 days and 2 μM MG132 for last 12 h and pulled down by Ni-NTA agarose. **k**, **l** Schematic diagrams showing the reciprocal inhibition of PIN1-APC/C^CDH1 governs cell-cycle entry and exit. The images were representative images from 3 independent experiments (**a**, **b**, **e**–**j**). Source data are provided as a Source Data file.

well-tolerated and significantly delayed tumor progression and increased overall survival compared to either monotherapy (Fig. 7a–d). To understand the effects of combination therapy in tumor microenvironment, we analyzed the immune landscape of *Brca1*-proficient tumors using cytometry by time-of-flight (CyTOF). Our 16-marker panel was designed to identify tumor cells and the major tumor infiltrating immune cell subtypes (Supplementary Data 4). Two-dimensional maps of the data were generated for a comprehensive view of the tumor-immune ecosystem using the dimensionality reduction algorithm UMAP[57]. Tumor cells were the main population with a mean of 30% across samples and distinct from tumor-infiltrating leukocytes (TILs), which were characterized by EpCAM and CD45, respectively (Supplementary Fig. 10c). Ten different cell types of TILs were identified based on expression of their phenotypic markers (Fig. 7e, Supplementary Fig. 10d). We observed a significant decreased percentage of proliferating tumor cells and regulatory T cells (Tregs) after combination treatment for 2 weeks. In contrast, increased percentage of infiltrating CD8 + T cells, CD4 + T cells, EpCAM+ macrophages and B cells were observed in the combination group, but not with either monotherapy (Fig. 7f–l). The combination of PIN1 inhibitor and CDK4 inhibitor thus enhances anti-tumor immunity, with decreased Tregs, increased cytotoxic T cells and evidence for tumor cell phagocytosis, raising the possibility that this rewiring of the immune system could further improve the therapeutic effect.

To confirm such anti-tumor immune response induced by the combination treatment, we performed RNA sequencing of TNBC tumors after treatment with Sulfopin, Abemaciclib or their combination. Indeed, the combination treatment had much larger effects on the gene sets of the immune response signature compared with single-drug treatment (Supplementary Fig. 10e, f). The gene set enrichment analyses revealed a more significant positive enrichment of adaptive immune response, lymphocyte-mediated immunity and enhanced tumor antigen presentation upon combination treatment (Supplementary Fig. 10e, f), which was consistent with the immune response signatures in *PIN1* KO cells (Supplementary Fig. 10g), suggesting that the combination treatment not only decreased PIN1 levels, but also increased tumor immunogenicity. Thus, the combination of PIN1- and CDK4-inhibitors achieves synergistic anti-tumor activity against *Rb*-proficient or -deficient TNBC in immune-compromised or -competent mice, with an excellent safety profile, making it a strong candidate for clinical development.

## Discussion

Epithelial cells execute an active program to prevent cells from entering the cell cycle that relies on reducing the levels of continuously accumulating mitotic proteins, including cyclins, through ubiquitin-mediated degradation[1,58]. A major ubiquitin ligase is APC/C^CDH1, which targets a range of mitotic proteins for degradation[7–9]. APC/C^CDH1 has been identified as a tumor suppressor[59,60]. Inactivation of APC/C^CDH1 is achieved via multi-site proline-directed phosphorylation[14], but

whether APC/C^CDH1 activity is subject to further regulation after phosphorylation is unknown. On the other hand, through catalyzing *cis-trans* isomerization after proline-directed phosphorylation, PIN1 promotes tumorigenesis by acting as a common regulator of numerous oncogenic signaling pathways[19,23]. Notably, PIN1 was originally identified as an essential mitotic regulator[21] and most effective PIN1 inhibitors including successful leukemia drugs also induce PIN1 protein degradation[24,30,32,35]. However, relatively little is known about the mechanisms regulating PIN1 protein stability during the cell cycle and PIN1 inhibitor-induced degradation.

In *RB*-proficient tumors, RB-mediated transcriptional regulation and APC/C^CDH1-mediated post-transcriptional regulation of mitotic proteins collaboratively control cell cycle entry. However, since RB-pathway is frequently disrupted in tumors, APC/C^CDH1 becomes the sole cell cycle regulator in G0/G1 phase that can be modulated by CDKs-mediated phosphorylation. In this study, we uncovered that PIN1-catalyzed *trans* to *cis* isomerization of CDH1 is indispensable for stabilizing the phosphorylation status of APC/C^CDH1 so that PIN1 in concert with CDKs inactivates APC/C^CDH1 to accumulate mitotic proteins. Unexpectedly, we also revealed that PIN1 can be targeted for degradation by active APC/C^CDH1 in a reciprocal mechanism and that combined PIN1 and CDKs inhibition could synergistically and irreversibly activate APC/C^CDH1 to induce the robust degradation of PIN1 and other mitotic proteins. The mechanism of action is distinct from classical inhibitors or protein degraders and especially significant for the development of more effective cancer treatments.

Three inhibitors of CDK4 are currently approved for treating metastatic BC. As a single agent, neither Palbociclib nor Ribociclib has activity in TNBC[61], while Abemaciclib with a broader target specificity[47] is under investigation in this setting (NCT03130439). Despite an increased overall survival in Paloma-3 and Monaleesa-2[62,63], resistance to CDK4-inhibition inevitably emerges. Despite its mechanism of action, however, RB expression is not associated with and predictive of response in ER + BC[64] or in TNBC[65]. Hence, more effective CDK4 inhibitor combinations for ER+ or TNBC, irrespective of RB status, are urgently needed.

Our data show that combined inhibition of CDK4 and PIN1 permits deeper and longer-lasting remissions, even resulting in complete remission in some tumor-bearing mice. Surprisingly, this drug regimen did not have detectable toxicity or even affect hematopoiesis, as shown by normal complete blood cell counts. In cancer, PIN1 is frequently overexpressed[19,23] and APC/C^CDH1 frequently inactivated through phosphorylation[7,9,13], creating a wide therapeutic window for combined inhibition of CDK4 and PIN1. Combined inhibition synergistically and irreversibly reactivates APC/C^CDH1 and induces the degradation of PIN1 and other mitotic proteins, leading to irreversible cell cycle exit, which translates into synergistic anti-tumor activity against triple-negative breast cancer both in immune-compromised and -competent and/or *RB*-deficient or -proficient mouse models. Consistent with the previous findings that inhibition of either CDK4 or PIN1 triggers anti-

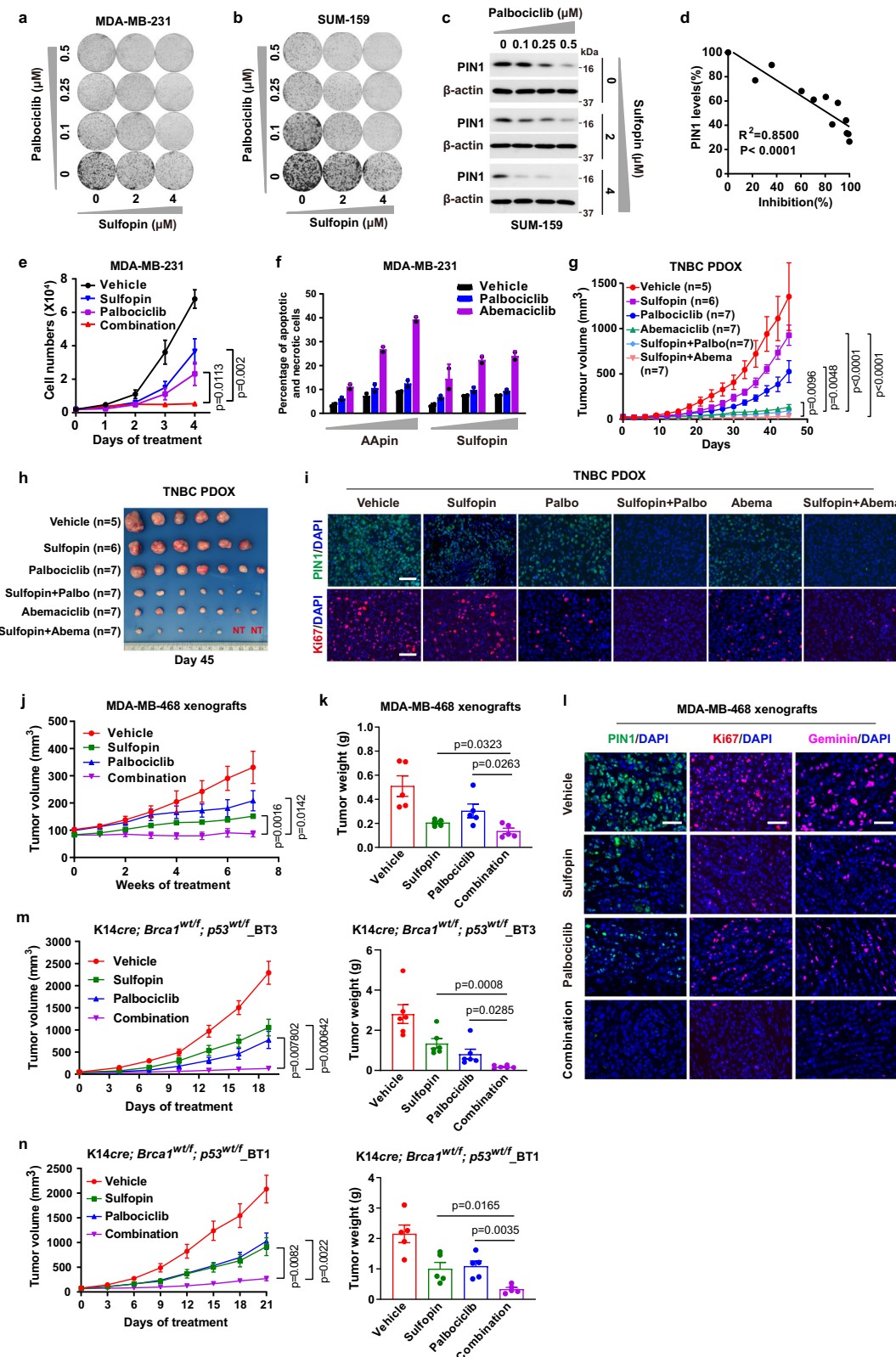

tumor immunity[30,55], we also found that the combination of PIN1 inhibitors and CDK4 inhibitors also create an immune environment with increased infiltration of cytotoxic CD8 + T cells and decreased infiltration of Tregs, raising the possibility that this rewiring of the immune system could further improve the therapeutic effects and increase survival by disrupting the immunosuppressive tumor microenvironment.

In summary, our work uncovers a reciprocal antagonism between PIN1 and APC/C$^{CDH1}$ as a fundamental cell cycle mechanism to regulate mitotic protein stability, whose aberration causes APC/C$^{CDH1}$ inhibition and PIN1 overactivation in a vicious feedback loop, leading to unchecked proliferation and cancer. Importantly, we further develop an effective therapeutic strategy using clinically available PIN1 inhibitors and CDK4 inhibitors to break this deleterious cycle by irreversibly

**Fig. 6 | PIN1 inhibitors synergize with CDK4 inhibitors against TNBC in human cells and immune-compromised mouse models.** Colony formation of MDA-MB-231 (**a**) and SUM-159 cells (**b**) treated with Sulfopin and Palbociclib for 2 weeks. **c** IB analysis of PIN1 in SUM-159 cells treated as in (**b**). **d** Correlation of cell growth inhibition (**b**) and PIN1 abundance (**c**) in SUM-159 cells. Two-sided p value for Pearson correlation coefficient. **e** Cell counts of MDA-MB-231 cells treated with 1 μM Palbociclib, 10 μM Sulfopin or their combination for 4 days. $n = 3$ independent experiments. Data in graphs are mean ± SD and analyzed by unpaired two-sided *t*-test. p values are shown. **f** MDA-MB-231 cells were treated with increasing concentrations of indicated drugs for 3 days, followed by analyzing apoptotic and necrotic cells by FACS. $n = 2$ independent experiments. **g** Tumor growth of TNBC PDOX from different treatments. Data in graphs are mean ± SEM and analyzed by unpaired two-sided t-test. Mice numbers and *p* values are shown. **h** Tumor sizes were shown when mice were euthanized after 45 days. NT, no tumor detectable.

**i** Representative immunofluorescence images for PDOX tumors stained with PIN1 (green) and Ki67 (red). Scale bars, 50 μm. **j** Tumor growth of MDA-MB-468 xenografts from different treatments. **k** Tumor weights were measured when mice were euthanized after 7 weeks. $n = 5$ mice per group (**j**, **k**). Data in graphs are mean ± SEM and analyzed by unpaired two-sided *t*-test. *p* values are shown (**j**, **k**).
**l** Representative immunofluorescence images for MDA-MB-468 xenografts tumors stained with PIN1 (green), Ki67 (red) and Geminin (pink). Scale bars, 50 μm. **m** Growth curves (left) and tumor weights (right) of K14*cre; Brca1wt/f; p53wt/f*_BT3 tumors from different treatments, $n = 6$ mice per group. **n** Growth curves (left) and tumor weights (right) of K14*cre; Brca1wt/f; p53wt/f*_BT1 tumors from different treatments, $n = 5$ mice per group. Data in graphs are mean ± SEM and analyzed by unpaired two-sided *t*-test (**m**, **n**). *p* values are shown (**m**, **n**). The images were representative images from 3 independent experiments (**a**–**c**). Source data are provided as a Source Data file.

reactivating APC/C^CDH1 and inducing degradation of PIN1 and many other mitotic proteins. These effects force permanent cell cycle exit and disrupt immunosuppressive tumor microenvironment, translating synergistic efficacy against *RB*-proficient and *RB*-deficient TNBC, paving the way for new clinical trials to evaluate their clinical impact on patients with TNBC.

## Methods
Our research complies with all relevant ethical regulations established by Beth Israel Deaconess Medical Center and Dana Farber Harvard Cancer Center (DF/HCC). All animal protocols were approved by the IACUC of the Beth Israel Deaconess Medical Center. The specimens were collected according to protocol DF/HCC 17-503 approved by the Institutional Review Board (IRB) at DF/HCC.

### Cell lines and plasmids
MDA-MB-468, BT-549, MDA-MB-231, MCF-7 and HEK293T cells were obtained from ATCC. Wild-type and *Cdh1*^−/− Mouse embryonic fibroblasts (MEFs) were kind gifts from Dr. Wenyi Wei. MCF-10A cells were gifts from the S. D. Cappell. SUM-159 cells were purchased from BioIVT. K14*cre; Brca1wt/f; p53wt/f* mouse cells were isolated from genetically engineered mouse model (K14*cre; Brca1wt/f; p53wt/f*). Among them, MDA-MB-231, MCF-7, HEK293T, MDA-MB-468, BT-549, MEFs and K14*cre; Brca1wt/f; p53wt/f* mouse cells were cultured in Dulbecco's modified Eagle's medium (DMEM) supplemented with 10% fetal bovine serum (FBS). SUM-159 cells were cultured in RPMI-1640 medium supplemented with 10% FBS. MCF-10A cells were cultured in MEBM^TM Basal Medium and Supplements (Lonza, CC-3150). All the cells used for the experiments were tested negative for mycoplasma contamination.

pLenti-HA-CDH1 was purchased from Applied Biological Materials Inc. pLKO-shRNF219, pLKO-shUBR5, pLKO-shRNF149, pLKO-shSMURF2, pLKO-shWWP2, pLKO-shUBE3A, pLKO-shUBE3B, pLKO-shNEDD4, pLKO-shKEAP1 and pLKO-shFBXO7 were purchased from Sigma-Aldrich. HA-CDH1-S163A, HA-CDH1-S163E, pLenti-HA-CDH1-7A, pLenti-3×Flag-PIN1, pLenti-3×Flag-PIN1-W34A, -W34A-RLAA, -RLAA, -M130L, -M130I, -C113A and -C113S were generated in our lab. PIN1 CRISPR/Cas9 KO construct was provided by Dr. Shingo Kozono. pLenti-HA-CDK4 and pLenti-HA-CDK6 lentiviral constructs were provided by Dr. Wenyi Wei. mCherry-Geminin (1-110) and Histone H2B-Turquoise lentiviral constructs were provided by Dr. Jia-Yun Chen. His-ubiquitin constructs was provided by Dr. Yu-Ru Lee.

### Reagents and antibodies
ATO (A1010), ATRA (R2625), MG132 (M7449), Thymidine (T1895), Nocodazole (M1404), Glutathione-agarose (G4510), Carboxymethylcellulose sodium salt (CMC-Na, C4888) and Senescence Cells Histochemical Staining Kit (CS0030) were purchased from Sigma-Aldrich. Palbociclib (PD0332991, S1116) and Abemaciclib (S5716) from Selleckchem. Hoechst 33342 Solution and Dead Cell Apoptosis Kit with

Annexin V FITC and PI from Thermo. Tumor Dissociation Kit (mouse) from Miltenyi Biotec. Sulfopin was provided by Dr. Nathanael Gray.

Antibodies for western blot: Anti-Pin1 mouse monoclonal antibody was provided by Dr. Xiao Zhen Zhou (homemade). Anti-Cdh1 (sc-56312, 1:000), anti-Cyclin E (C-19, sc-198, 1:1000) and anti-CDC20 (sc-13162, 1:1000) antibodies were purchased from Santa Cruz. Anti-RB (ab181616, 1:2000), anti-APC7 (ab4171, 1:500) and anti-Thiophosphate ester (ab92570, 1:5000) antibodies were purchased from Abcam. Monoclonal anti-Flag M2 antibody (F1804, 1:10000) from Sigma. Anti-HA-Tag rabbit mAb (3724, 1:1000), anti-HA-Tag mouse mAb (2367, 1:1000), anti-CDK2 rabbit mAb (2546, 1:1000), anti-CDK4 (D9G3E) rabbit mAb (12790, 1:1000), anti-APC8 rabbit mAb (15100, 1:1000), anti-Cyclin B1 antibody (4138, 1:1000), anti-Cyclin A2 mouse mAb (4656, 1:1000), anti-Cyclin D1 rabbit mAb (55506, 1:1000), anti-PLK1 rabbit mAb (4513, 1:1000), anti-Phospho-MAPK/CDK Substrates (PXS*P or S*PXR/K) (2325, 1:1000) and anti-Geminin rabbit mAb (52508, 1:1000) antibodies were purchased from Cell Signaling Technology. Anti-Emi1 mouse mAb (376600, 1:500) was purchased from Thermo.

Antibodies for immunoprecipitation: Anti-PIN1 rabbit monoclonal antibody (ab192036, 1:50) and anti-CDK4 antibody (ab68266, 1:50) were purchased from Abcam.

Antibodies for Immunofluorescent staining: Anti-PIN1 rabbit monoclonal antibody (ab192036, 1:50) and anti-PanCK (ab86734, 1:100) were purchased from Abcam, Anti-Ki67 (#50-828-02, 1:400) antibody was purchased from Biocare Medical.

Antibodies for CyTOF: Metal-conjugated antibodies used for CyTOF were purchased from Fluidigm, the antibodies details were provided in Supplementary Data 4.

### Tissue microarray staining, image acquisition and analysis
Breast tissue microarrays (TMAs) (BrCaStg1) were acquired from Cooperative Human Tissue Network's Mid-Atlantic Division. Each TMA underwent a dual staining process using panCK (Invitrogen 41-9003-82) and PIN1 (Abcam, ab192036) per manufacturer's recommendations. Image capture was executed using the CyteFinder high-throughput imaging system (RareCyte WA). We saved 10x magnification images from each TMA-slide in a 3-channel (DAPI/ Cy3/Cy5) configuration as Bio-format stacks. These stacks were background-corrected using the rolling ball technique and merged into a single image montage for each channel via ImageJ. We manually segmented each tissue sample and quantified the intensities of DAPI, PanCK, and PIN1 using ImageJ (NIH, MD). The PanCK marker was used to differentiate tumor cells from stromal and other non-tumor components in the tissue sample. This allowed for a more precise normalization of PIN1 expression relative to the presence of tumor cells. Supplementary Data 1 presents the PIN1 values adjusted based on this normalization, alongside the associated follow-up periods. For our study, we set a threshold for PIN1: values >=2 was categorized as 'PIN1 high' (approximating the upper tertile cut-off), while those <2 were deemed 'PIN1 low'.

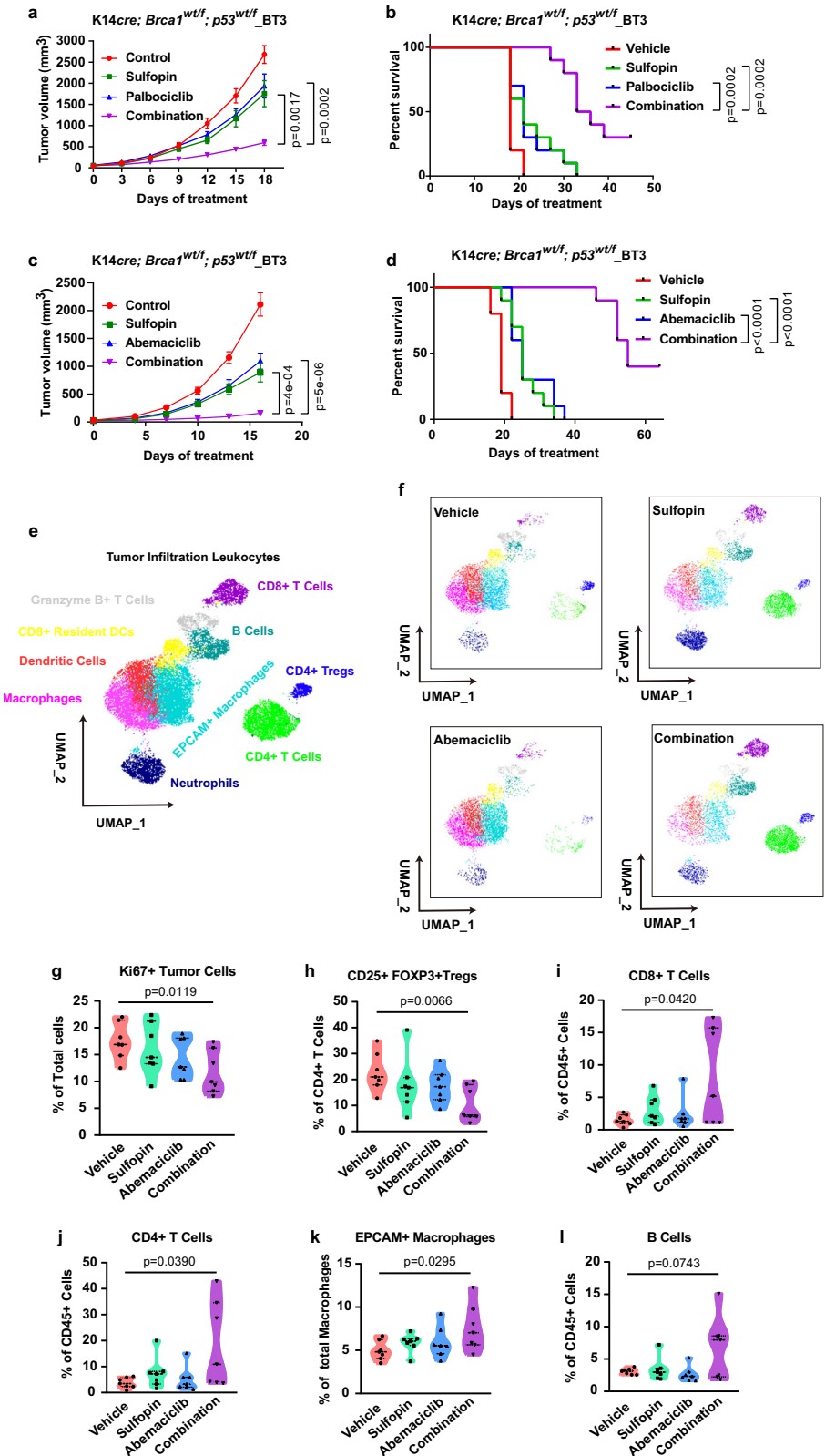

## Human BC specimens

Tissues were from nine female patients diagnosed with metastatic, ER + BC between 1999 and 2021 and treated with the combination of an estrogen receptor blocker and the CDK4/6-inhibitor Palbociclib. The specimens were collected according to protocol DF/HCC 17-503 approved by the Institutional Review Board (IRB) at Dana Farber Harvard Cancer Center. Specimens were de-identified and logged in a RedCap database and retrieved from the pathology archive. Given the minimal risk of the study, the requirement for informed consent was waived by the IRB. See also Supplementary Table 2.

## Multiplex immunohistochemistry (mIHC) assays

We performed the Opal™ multiplex IHC assay (Akoya Biosciences) to stain for three markers and nuclei: (1) PIN1 (1:50; Abcam, ab192036), (2)

**Fig. 7 | Combination of PIN1- and CDK4-inhibitors achieves synergistic anti-tumor immunity and efficacy against *RB*-proficient or -deficient TNBC in immune-competent mouse models.** Growth curve (**a**) and survival curve (**b**) generated from FVB mice bearing K14*cre; Brca1wt/f; p53wt/f*_BT3 tumors treated with vehicle (median survival of 18 days), Sulfopin (median survival of 21 days), Palbociclib (median survival of 21 days) or their combination (median survival of 34.5 days), n = 10 mice per group. Growth curve (**c**) and survival curve (**d**) generated from FVB mice bearing K14*cre; Brca1wt/f; p53wt/f*_BT3 tumors treated with vehicle (median survival of 19 days), Sulfopin (median survival of 25 days), Abemaciclib median survival of 25 days) or their combination (median survival of 55 days), n = 10 mice per group. Data are mean ± SEM and analyzed by two-sided unpaired student's *t*-test (**a**, **c**) or log-rank test (**b**, **d**). *p* values are shown (**a**–**d**). **e** Concatenated UMAP plots displaying 24,000 CD45+ cells derived from K14*cre; p53wt/f; Brca1wt/f* mouse tumors treated with Sulfopin, Abemaciclib or their combination for two weeks and colored by the main cell populations based on manual annotation of PhenoGraph clustering. **f** Individual UMAPs for CD45+ cells from different treatments. **g–l** The violin plots generated by CyTOF data showing percentages of indicated cells from different treatments. n = 7 per group. Data in graphs are analyzed by unpaired two-sided *t*-test. Vehicle vs. Combination, *p* values are shown. Source data are provided as a Source Data file.

PanCK (1:100; Abcam, ab86734), (3) Ki67 (1:400; Biocare Medical, #50-828-02), (4) nucleus with 4,6-diamidino-2-phenylindole (DAPI; Akoya Biosciences). The pairings of each of the 6 markers with an Opal™ fluorophore and the order of antibody staining were optimized as follows: cycle 1 was PIN1 paired with Opal™ fluorophore 620, cycle 2 was Ki67 with 690, cycle 3 was PanCK with 780. Opal™ 780 is an antibody-based reaction that requires the use of tyramide signal amplification-digoxigenin (TSA-DIG) for signal amplification, as such, PanCK paired with Opal™ 780 had to be stained last.

The sections were baked at 65 °C for three hours before placing the slides into the Bond RX fully automated research stainer (Leica Biosystems, Deer Park IL) for dewaxing and the Opal™ assay. The Opal™ assay began with a 40 min of heat-induced epitope retrieval step at 100 °C using the Bond epitope retrieval solution 2 (pH 9), followed by the five cycles of blocking (5 min), primary antibody incubation (30 min), incubation with Opal™ polymer horseradish peroxidase (HRP) reagent (10 min), signal amplification with the marker's paired Opal™ fluorophore (10 min), and antibody stripping using Bond epitope retrieval solution 1 (pH 6) at 95 °C for 20 min. The fourth cycle to stain for PanCK was slightly modified: blocking, PanCK antibody incubation, HRP incubation, using the Opal™ TSA-DIG for signal amplification (10 min), stripping of TSA-DIG, and incubation with Opal™ 780 for signal generation (60 min). The slides were last stained for DAPI (5 min). There were three to four washes in between each step. Slides were mounted with ProLong™ gold antifade mountant (Thermo Fisher Scientific). IF images were visualized using Phenochart (Akoya Bioscience). IF-stained slides were digitized at 40X by PhenoImager HT™ (formerly Vectra® Polaris™). The images were analyzed using inForm software (version 2.5) by building custom algorithms to detect and quantify the whole tissue sections.

### In gel digestion, mass spectrometry and data procession
For GST pull-down: MDA-MB-231 cells were lysed in pull-down buffer (20 mM Tris pH 8.0, 150 mM NaCl, 1 mM EDTA pH 8.0, 0.5% Nonidet-P40). The cell extracts were pre-cleared by glutathione agarose beads and incubated with 1 μM GST or GST-PIN1 overnight at 4 °C. Protein complexes were recovered on glutathione agarose beads for 2 h at 4 °C, washed four to six times with pull-down buffer and eluted by boiling in SDS−containing sample buffer. The eluted proteins (from both GST and GST-PIN1 samples, one for each) were resolved by SDS-PAGE on 4-12% gradient gel (Invitrogen) for staining with Coomassie blue.

For Flag-tag immunoprecipitation, MDA-MB-231 cells stably expressing Flag-PIN1. Cells were lysed in Co-IP lysis buffer (Thermo) and pre-cleared by Mouse IgG-Agarose (Sigma) for 1 h at 4 °C and then incubated with IgG-Agarose or anti-Flag M2-Agarose (Sigma) for 2 h at 4 °C. The agaroses were washed four times with Co-IP lysis buffer and eluted by 3 x Flag Peptide from the anti-Flag M2-Agarose and boiled in SDS−containing sample buffer. The eluted proteins (from both IgG and Flag-PIN1 samples, one for each) were resolved by SDS-PAGE on 4–12% gradient gel (Invitrogen) for staining with Coomassie blue.

For HA-tag immunoprecipitation, MCF-7 WT or *PIN1*-KO cells were stably expressed HA-CDH1. Cells were lysed in RIPA lysis buffer (Thermo) and pre-cleared by Mouse IgG-Agarose (Sigma) for 1 h at 4 °C and then incubated with IgG-Agarose or anti-HA-Agarose (Sigma) for 2 h at 4 °C. The agaroses were washed four times with RIPA lysis buffer and boiled in SDS−containing sample buffer. The immunoprecipitated HA-CDH1 proteins (from both WT and *PIN1*-KO, one for each) were resolved by SDS-PAGE on 4–12% gradient gel (Invitrogen) for staining with Coomassie blue.

The gel lanes stained with Coomassie blue were unevenly excised into several sections. Each section was cut into approximately 1-mm$^3$ pieces. The gel slices were first destained with the 30% acetonitrile in 100 mM NH$_4$HCO$_3$, dried by speedvac, and then incubated with 10 mM Dithiothreitol (DTT) for 1 h at 56 °C and then 20 mM iodoacetamide (IAA) in dark for 45 min at room temperature. After reduction and alkylation, the samples were digested with trypsin (Promega) at 10 ng/μL overnight at 37 °C. For the CDH1 phosphorylation site pS163 quantification the Lys-C (Promega) was used for digestion. The supernatant was collected and then combined with peptides digested and extracted from the gel slices with 80% acetonitrile containing 0.1% TFA. Peptide purification was performed on C18 column (MarocoSpin Columns, NEST Group INC) and 1 μg of the peptide was injected for mass spectrometry analysis.

The samples were measured by data-independent acquisition (DIA) mass spectrometry method as described previously[66–68]. The Orbitrap Eclipse Tribrid mass spectrometer (Thermo Scientific) instrument coupled to a nanoelectrospray ion source (NanoFlex, Thermo Scientific) and EASY-nLC 1200 systems (Thermo Scientific, San Jose, CA). A 120-min gradient was used for the data acquisition at the flow rate at 300 nL/min with the temperature controlled at 60 °C using a column oven (PRSO-V1, Sonation GmbH, Biberach, Germany). All the DIA-MS methods consisted of one MS1 scan and 33 MS2 scans of variable isolated windows with 1 m/z overlapping between windows. The MS1 scan range is 350–1650 m/z and the MS1 resolution is 120,000 at m/z 200. The MS1 full scan AGC target value was set to be 500% and the maximum injection time was 100 ms. The MS2 resolution was set to 30,000 at m/z 200 with the MS2 scan range 200–1800 m/z and the normalized HCD collision energy was 28%. The MS2 AGC was set to be 4000% and the maximum injection time was 50 ms. The default peptide charge state was set to 2. Both MS1 and MS2 spectra were recorded in profile mode. DIA-MS data analysis was performed using Spectronaut v16[69–71] with directDIA algorithm by searching against the Uniprot[72] downloaded human fasta file. The oxidation at methionine was set as variable modification, whereas carbamidomethylation at cysteine was set as fixed modification. For the CDH1 phosphorylation site pS163 quantification, the phosphorylation (S/T/Y) was also set as variable modification with the PTM score >0.75. Both peptide and protein FDR cutoffs (Qvalue) were controlled below 1% and the resulting quantitative data matrix were exported from Spectronaut. All the other settings in Spectronaut were kept as Default.

### In situ proximity ligation assay
In situ proximity ligation assay (PLA) was performed using Duolink Anti-Rabbit PLUS and Anti-Mouse MINUS PLA Probes (Sigma), according to the manufacturer's instructions. Briefly, prepare samples on slides, then

treat for fixation, retrieval, and permeabilization. Block slides with Duolink® Blocking Solution at 37 °C for 60 min. Apply diluted antibodies CDH1 (Abcam, ab227784) and PIN1 (Santa Cruz, sc-46660) or CDK4 (Santa Cruz, sc-23896), followed by diluted PLUS and MINUS PLA probes, each with specific washes in between. Mix and incubate with a ligation solution, and then with an amplification buffer-polymerase mix, ensuring protection from light during amplification. After final washes, mount slides and analyze using fluorescence microscope.

### In vitro treatment

Sulfopin and AApin (ATRA + ATO) treatment was described as previously[32,36]. Briefly, cells were seeded in 6-well plates and treated with increasing concentrations of AApin (ATO (0.5, 1, 1.5, 2 µM) plus ATRA (5, 10, 15, 20 µM) in 1:10 ratio) for 3 days. Cells were treated with increasing concentration of Sulfopin (2, 4, 8, 10 µM) or Palbociclib (0.5, 1, 2, 4 µM) for 3 days. Drugs were replenished in media every 24 h to ensure that PIN1 and CDK4/6 inhibition was maintained for the duration of the experiment.

### STED imaging

Except where indicated otherwise the steps were performed at room temperature. Cells were rinsed with PBS twice and fixed with 2% PFA for 15 min. Fixative was removed by washing with PBS 3 times. Cells were then permeabilized with 0.1% Triton for 10 min. After removing Triton, cells were blocked with 5% BSA for 1 h and then incubated with anti-PIN1 (Abcam) and anti-CDH1 (Santa Cruz) antibodies overnight at 4 °C. After three washes with PBS, the cells were then incubated with Alexa Fluor® 514 Goat Anti-Mouse (Invitrogen) and Alexa Fluor® 568 Goat Anti-Rabbit (Abcam) antibodies for 1 h. Following the incubation, the cells were washed 3X with PBS and mounted for STED imaging. Colocalization rates were calculated using the LAS X software (Leica).

### Real-time PCR

Total RNAs were extracted using the QIAGEN RNeasy mini kit. cDNA synthesis was performed using Maxima Universal First Strand cDNA Synthesis Kit from Thermo Scientific. qPCR reactions were performed with FastStart Universal SYBR Green Master (Rox) from Roche. The experiments were performed according to the manufacturer's instructions. The sequences of the primers used for qRT-PCR analyses were provided in Supplementary Table 4.

### RNA sequencing and data analysis

Total RNAs were extracted from the BC cell lines WT and *PIN1* KO MCF-7 and MDA-MB-468 respectively, or from mouse tumor tissues after treatment with Sulfopin, Abemaciclib or their combination. RNA-sequencing samples were prepared as previously described[73]. Gene set enrichment analysis (GSEA) was performed using GSEA software 4.2.3 (Broad). Normalized counts were used for GSEA analysis against the biological process related gene sets. Normalized enrichment scores (NES) were used to generate bar graphs for visualization of the functional transcriptional outputs of the three cell lines.

### Drug combination test and synergy calculations

MDA-MB-231, MDA-MB-468 and SUM-159 cells were seeded out in appropriate dilutions and treated with increasing concentrations of two drugs to form colonies in 1–3 weeks. Colonies are fixed with methanol (100% v/v), stained with crystal violet (0.5% w/v) and counted using Celigo Image Cytometer. The percentage of growth inhibition was calculated based on colony numbers and areas. The inhibition heatmaps and ZIP synergy scores were generated and calculated by SynergyFinder[74].

### Time-lapse live imaging and single-cell tracking

MCF-7 cells were stably expressed with mCherry-Geminin (1-110) and a histone H2B-Turquoise. Cells were then plated 24 h before starting the

microscope acquisition. PIN1 inhibitors or CDK4/6 inhibitor were added in the medium and imaged using a Nikon Eclipse TE-2000 inverted microscope with a 10X Plan Apo objective and a Hamamatsu ORCA-ER camera, equipped with environmental chamber controlling temperature, atmosphere (5% CO2) and humidity. Images were acquired every 30 min using the MetaMorph Software. For each condition filmed, 4 different fields were selected.

p53Cinema single cell analysis package was used for semi-automatic tracking of individual cells in live cell imaging datasets as described previously[75]. Tracking data were then used to quantify intensity of fluorescent reporters from background subtracted images by averaging 10 pixels within the cell nucleus. Cells were tracked using only information about a constitutively expressed nuclear marker, such as H2B-Turquoise, and were thus blind to the dynamics of molecular players of interest, such as mCherry-Geminin. Only cells that remained within the field of view throughout the entire duration of the experiment were considered for downstream analyses. We defined the frequency of G1 arrest as those cells that arrested in G1 phase for at least 20 h after drugs were added. S/G2 durations were calculated by the time that cells spent in S/G2 phase after drugs were added.

### Cell synchronization and cell cycle profiling

Double thymidine block: cells were grown in the presence of 2 mM thymidine for 18 h, washed with PBS, and grown in fresh media without thymidine for 8 h. 2 mM thymidine was added again for another 18 h to block cells at G1/S. Nocodazole block: cells were arrested in M phase by growth in 100 ng/ml nocodazole for 18 h, washed with PBS, and grown in fresh media. Synchronized cells were collected at the indicated time points and fixed by 75% ethanol at −20 °C overnight. After fixation, the ethanol was completely removed via centrifugation, and the cells were washed three times with cold PBS. Then, the cells were resuspended in propidium iodide (PI) staining solution provided by cell cycle kit (Beckman Coulter, C03551) according to the manufacturer's instructions. Stained cells were sorted with CytoFLEX LX1 Flow Cytometer. The results were analyzed by FSC Express software.

### Annexin V-FITC−PI double staining

For detection of apoptosis, cells treated with indicated inhibitors were co-stained with Annexin V-FITC and PI (Dead Cell Apoptosis Kit, Invitrogen) according to the manufacturer's instructions. Stained cells were sorted with CytoFLEX LX1 Flow Cytometer.

### Immunoblot and immunoprecipitation analyses

For IB analysis, cells were lysed in RIPA buffer (Thermo) supplemented with protease inhibitors (Sigma) and phosphatase inhibitors (Sigma). Protein concentrations were measured using Protein Assay Dye Reagent (Bio-Rad) and a Beckman Coulter. Equal amounts of protein were resolved by SDS-PAGE and probed with indicated antibodies. For immunoprecipitations analysis, cells were lysed in IP lysis buffer (Thermo) and pre-cleared by Mouse IgG-Agarose (Sigma) for 1 h at 4 °C and then incubated with anti-Flag M2-Agarose (Sigma) for 2 h at 4 °C. The agaroses were washed four times with IP lysis buffer and boiled in standard Laemmli-Buffer with 5% final concentration of β-mercaptoethanol before being resolved by SDS−PAGE and probed with indicated antibodies. Uncropped versions of the western blots were provided in the Supplementary Data 1.

### GST pull-down assay

Cells were stably expressing indicated proteins and lysed in pull-down buffer (20 mM Tris pH 8.0, 150 mM NaCl, 1 mM EDTA pH 8.0, 0.5% Nonidet-P40). The cell extracts were pre-cleared by glutathione agarose beads and incubated with 1 µM GST or GST fusion proteins overnight at 4 °C. Protein complexes were recovered on glutathione agarose beads for 2 h at 4 °C, washed four to six times with pull-down

buffer and eluted by boiling in SDS−containing sample buffer. Bound proteins were resolved by SDS−PAGE.

### In vivo ubiquitination assay
293 T cells were transfected with His-ubiquitin and the indicated constructs. Thirty-six hours after transfection, cells were treated with 2 μM MG132 for 12 h and lysed in buffer A (6 M guanidine-HCl, 0.1 M Na$_2$HPO$_4$/NaH$_2$PO$_4$, and 10 mM imidazole pH 8.0). After sonication, the lysates were incubated with Ni−NTA beads (QIAGEN) for 3 h at 4 °C. Subsequently, the His pull-down products were washed twice with buffer A, twice with buffer A/TI (1 volume buffer A and 3 volumes buffer TI), and once with buffer TI (25 mM Tris-HCl and 20 mM imidazole pH 6.8). The pull-down proteins were resolved by SDS−PAGE for IB.

### In vitro ubiquitination assay
In vitro ubiquitination assay was performed as described previously[9]. Briefly, 30 μl of mouse anti-CDC27 antibody (Santa Cruz, sc-9972) coupled to 30 μl of protein G-agarose (Roche) was incubated with 2 mL extracts from nocodazole arrested and released G1 (3 h post release) cells, and mixed for 2 h at 4 °C. The beads were washed three times with 1 mL swelling buffer (SB; 25 mM HEPES, pH7.5, 1.5 mM MgCl$_2$, 5 mM KCl, 0.1% Nonidet P-40) and twice with SB. Finally, In an 8-μl reaction volume, 0.5 μl of 10 μM E1 (625 nM final), 1 μl of 10 μM UBE2C (1.25 μM final), 1 μl of 10 μM UBE2S (1.25 μM final), 1 μl of 10 mg/ml ubiquitin (1.25 mg/ml final), 1 μl of 8 × ubiquitylation assay buffer (250 mM Tris 7.5, 500 mM NaCl and 100 mM MgCl$_2$), 1 μl of 100 mM DTT, 1.5 μl of energy mix (150 mM creatine phosphate, 20 mM ATP, 20 mM MgCl$_2$, 2 mM EGTA, pH to 7.5 with KOH) and 1 μl of recombinant human PIN1 was mixed with 5 μl of APC/C resin. Reactions were carried out at 30 °C for 0.5 h. Reactions were stopped by the addition of SDS sample loading buffer, resolved on a 4−15% SDS-acrylamide gel.

### PIN1 enzymatic activity assay
Briefly, Flag-tagged PIN1 WT and RLAA mutants were transfected into HEK293T cells, followed by being immunoprecipitated with anti-Flag M2-Agarose (Sigma) and eluted by competition with FLAG peptide (Sigma). PIN1 activity was determined by SensoLyte® Green PIN1 Activity Assay Kit (AnaSpec) according to the manufacturer's instructions. Fluorescence intensity was measured at Ex/Em=490 nm/520 nm and data were recorded every 15 min for 120 min using BIOTEK Synergy H1.

### In vitro kinase assay
HA-tagged CDH1 WT and mutants were transfected into HEK293T cells, followed by being immunoprecipitated with monoclonal Anti-HA-Agarose antibody (Sigma, A2095). The purified HA-CDH1 proteins were then incubated with 500 uM of ATPγS (Abcam, ab138911) and 0.5 ug of recombinant human cyclin D1 + CDK4 proteins (Abcam, ab55695) in the kinase reaction buffer (50 mM Tris-HCl, 10 mM MgCl$_2$, 0.1 mM EDTA, 2 mM DTT, 0.01% Brij 35, pH 7.5) for 30 min at room temperature. Then adding 2 mM of PNBM (Abcam, ab138910) and allowing the alkylating reaction proceed for additional 2 h at room temperature. The reaction was then terminated by adding 5x SDS loading buffer and boiled for 10 min. Samples were then subjected to IB using anti-Thiophosphate ester antibody (Abcam, ab92570).

### In vivo therapy for immunocompromised TNBC mouse models
All animals were housed under controlled conditions with an ambient temperature set at 70 °F and relative humidity ranging from 40% to 60% under a 12/12-h light/dark cycle with unrestricted access to food and water throughout the duration of the experiment. All animal experiments were approved by the IACUC of the Beth Israel Deaconess Medical Center. Maximum permitted diameter of tumors was 20 mm. Toward the end of the study, there were instances where some mice marginally surpassed the predefined maximum tumor burden. These mice were closely monitored for signs of distress and euthanasia was promptly performed at any indication of significant distress. Pieces from PDOX or K14*cre; Brca1wt/f; p53wt/f* tumors were subcutaneously implanted into the mammary fat pads of 6-week-old BALB/c female nude mice. Mice were randomly assigned to six groups with comparable average tumor size. Treatments were started once the tumors reached 3–5 mm in diameter and continued until 45 days. Sulfopin treatment was given by intraperitoneal injection with a dosage of 40 mg/kg (dissolved solution: 5% DMSO in D5W, 7 days/week), Palbociclib treatment was given by oral gavage with a dosage of 100 mg/kg (dissolved solution: saline, 5 days/week), Abemaciclib treatment was given by oral gavage with a dosage of 100 mg/kg (dissolved solution: 0.5% CMC-Na, 5 days/week), or drug combinations in which each compound was administered at the same dose and scheduled as a single agent. Tumor sizes were measured every three days by caliper after implantation and tumor volume was calculated by the modified ellipsoidal formula: tumor volume = ½ length × width$^2$. The investigators were not blinded to allocation during experiments and outcome assessment.

### In vivo therapy for immunocompetent TNBC mouse models
All animals were housed under controlled conditions with an ambient temperature set at 70 °F and relative humidity ranging from 40% to 60% under a 12/12-h light/dark cycle with unrestricted access to food and water throughout the duration of the experiment. All animal experiments were approved by the IACUC of the Beth Israel Deaconess Medical Center. Maximum permitted diameter of tumors was 20 mm. Toward the end of the study, there were instances where some mice marginally surpassed the predefined maximum tumor burden. These mice were closely monitored for signs of distress and euthanasia was promptly performed at any indication of significant distress. Pieces from breast tumors generated in K14*cre; Brca1wt/f; p53wt/f* female mice with FVB/129P2 mixed genetic background were transplanted into the mammary pads of 6-week-old FVB female mice. For survival studies, treatments were started once the tumors reached 3–5 mm in diameter and continued until mice were symptomatic or tumors reached around 20 mm. Sulfopin treatment was given by intraperitoneal injection with a dosage of 60 mg/kg (7 days/week), Abemaciclib treatment was given by oral gavage with a dosage of 100 mg/kg (7 days/week), or drug combinations in which each compound was administered at the same dose and scheduled as a single agent. Tumor sizes were measured every three days by caliper after implantation and tumor volume was calculated by the modified ellipsoidal formula: tumor volume = ½ length × width$^2$. The investigators were not blinded to allocation during experiments and outcome assessment.

### NMR spectroscopy
All NMR experiments were acquired on Bruker NEO 600 MHz spectrometer equipped with a TCI cryoprobe at 25 °C. 0.1 mM $^{13}$C, $^{15}$N-enriched PIN1 sample dissolved in pH 6.6 buffer made-up of 20 mM Potassium Phosphate, 100 mM NaSO$_4$ and 10% D$_2$O was used to study peptide interaction. NMR assignments of PIN1 were taken from the BMRB database (accession number 27579) and confirmed using 3D-HNCA experiment. Synthetic peptides CDH1-pS163 (comprised of CDH1 residues 161-183 with phosphorylated Ser163 and isotope labeled Pro164, LR(pS)P($^{13}$C, $^{15}$N)RKPTRKISKIPFKVLDAPE) were purchased from Pepmic. A systematic titration between PIN1 and CDH1 phosphopeptide was performed by acquiring series of 2D- HSQC spectra to attain resonance assignments of the peptide-bound form of PIN1. The absolute average chemical shift perturbation was calculated by using an equation, $[(\Delta\delta_H^2 + (\Delta\delta_N/5)^2)/2]^{1/2}$, available in software NMRFAM Sparky version 1.414.

## Experiment-guided model

The chemical shift perturbation was interpreted as ambiguous iterative restrains used for docking a random conformation of the phosphopeptide on PIN1 (PDB: 1PIN)[45] using HADDOCK2.2 webserver[76]. The restrains were derived by marking two strongly perturbed PIN1 residues, R17 and W34 as active residues and three moderately perturbed residues S18, Y23 and E35 as passive residues. The peptide was assumed to be fully flexible with the phosphoserine, pS163, and its adjacent proline, P164, being the active residues that interact with PIN1. In subsequent runs, the model was refined using ambiguous distance restraints based on the interpretation of previously solved crystal structures of similar phosphopeptides bound to the WW domain of PIN1[77].

## Proline isomerization study

Commercially synthesized specific $^{13}C$, $^{15}N$- P164 labeled CDH1 phosphopeptide was used to facilitate direct quantitative determination of the cis and trans proline populations. The strong $^{13}C$-HSQC peaks originating from Pro164 can be easily distinguished from the weak peaks due to ~1% natural abundance $^{13}C$ present in the rest of the peptide. Two isolated sets of peaks were observed for P164. Based on the interpretation of the chemical shifts, the major peaks were assigned as trans isomer and the minor peaks were assigned as cis isomer[78]. The proline resonance assignments were further confirmed using a 2D-$^{13}C$-HSQCTOCSY experiment while no attempts were made to stereospecifically assign proton resonances, thus the assignment of HB2 and HB3, HG2 and HG3, and HD2 and HD3 are interchangeable. 58 μM free peptide and its complex with a 4-fold molar excess of PIN1, dissolved in the above-mentioned NMR buffer were used to estimate the cis and trans isomer populations at 25 °C.

## Docking model

A docking model was built to explain the interaction between PIN1 (PDB: 1PIN)[45] and the WD40 domain of CDH1 (PDB: 4UI9_R)[79] using the HADDOCK2.2 webserver[76]. The docking was performed using ambiguous iterative restraints between PIN1 residues K117 to G128 and FRZ1 residues on the D-box binding interface, as observed in the anaphase-promoting complex (PDB: 4UI9)[79]. The model was refined using additional weak ambiguous distance restraints between the two canonical PIN1 residues, R119 and L122, and CDH1 residues D180, P182, E465 and L467.

## Immunofluorescence analysis

PDOX tumor tissue sections were boiled in 10 mM sodium citrate (pH 6.0), for antigen retrieval after deparaffinization. The sections were permeabilized with PBS containing 0.1–0.5% Triton X-100 and blocked with PBS containing 5% Goat serum for 30 min RT. The primary antibodies were diluted in PBS containing 1% Goat serum (1:200) and incubated in slides for overnight at 4 °C. The slides were rinsed by PBS three times, each time for 5 min. Secondary antibodies were diluted in PBS (1:1000) and incubated for 20 min at room temperature. 20 mg/ml DAPI was used to label nuclear of cells. Slides were scanned at least three different representative areas at 20X magnification using BZ-X800 fluorescence microscope (KEYENCE).

## Antibody staining for mass cytometry

Except where indicated otherwise sample staining and acquisition were carried out at room temperature. Mouse tumor tissues were dissociated into single-cell suspension using the Tumor Dissociation Kit (Miltenyi Biotec) and the gentleMACS™ Octo Dissociator following manufacturer's instructions. Cells were stained with Cisplatin-195Pt at a final concentration of 1 μM for 5 min. After viability staining, cells were incubated with Fc-Receptor blocking solution. Fifteen minutes later, the surface staining antibody cocktail was added into each cell suspensions and incubated for 30 min without washing out the Fc

blocking. The cells were then washed with Maxpar Cell Staining Buffer (CSB) (Fluidigm) for a total of two wash. Then cells were incubated with Nuclear Antigen Staining Buffer (Fluidigm) with gentle vortex for 30 min. After two washes with Nuclear Antigen Staining Perm (Fluidigm), cells were stained with secreted and nuclear antigen antibody cocktail for 30 min. Following the staining, cells were washed twice with Nuclear Antigen Staining Perm and fixed by freshly made 1.6% paraformaldehyde for 10 min. Afterwards, cells were incubated with Cell-ID Intercalator-Ir (Fluidigm) overnight at 4 °C. Cells were then washed in CSB buffer and with subsequent washes in Cell Acquisition Solution (CAS) (Fluidigm) to remove buffer salts and cell debris for total of two washes. Immediately prior to sample acquisition, cells were resuspended at $5 \sim 6 \times 10^5$ cells per mL in CAS containing EQ™ Four Element Calibration Beads (1:5) (Fluidigm) and filtered through a 40 μm cell strainer.

## Mass cytometry acquisition setting and data analysis

For quality control, the acquisition event rate was maintained under 500 events/s, and the EQ™ beads were confirmed to have clustered events >10,000 and median Eu151 and Eu153 intensity were over 1000 to ensure appropriate mass sensitivity. Original data acquired by CyTOF (Fluidigm Helios) were randomized and normalized using the FSC processing function of the CyTOF software (Fluidigm Helios). The Gaussian Parameters were applied to gating the FSC processed files using FlowJo. Standard gating strategy were used for single cell analysis with multiple markers.

To visualize the high-dimensional data and identify clusters of cells with a similar expression of cell surface markers in CyTOF, the populations of interest were gated and UMAP algorithm was applied on data from a certain number of randomly selected cells from each sample. Clustering analysis was performed using the PhenoGraph implementation in the FlowJo plugins. The resulting PhenoGraph clusters were projected onto the UMAP. ClusterExplorer plugin in FlowJo was performed to define the cell clusters by typical marker expression. For hierarchical clustering, the distances between clusters were computed using Eucledian measurement method. Dendrograms were generated using average linkage. A normalized heatmap for each marker within all generated clusters was displayed.

## Statistics and reproducibility

Quantitative data were presented as either mean ± SD or mean ± SEM derived from multiple independent experiments. Statistical analyses were conducted using unpaired two-sided $t$-tests, one-way or two-way ANOVA, or Log-rank test for Kaplan–Meier survival curves. Statistical details of experiments, including statistical tests and sample sizes used, can be found in the Figure legends. Randomization applies to all statistical analyses and the allocation of mice to treatment groups. Data collection and analysis procedures were not conducted blind to the experiment operators.

## Reporting summary

Further information on research design is available in the Nature Portfolio Reporting Summary linked to this article.

# Data availability

Full list of PIN1-inteacting proteins identified by mass spectrometry in this study are provided in Supplementary Data 2. The mass spectrometry raw data generated in this study have been deposited in the ProteomeXchange Consortium under accession code PXD046325. The PIN1 human breast cancer protein data were obtained from Tang, W. et al. dataset[37]. The data for the correlation of Pin1 and Geminin protein levels were obtained from cBioPortal for Breast Invasive Carcinoma (TCGA, PanCancer Atlas, mass spectrometry by CPTAC). The Cdh1 (FZR1) human breast cancer mRNA data were obtained from UCSC

Xena (https://xenabrowser.net/) for GDC TCGA Breast Cancer. The proteomics data of PIN1 KO versus WT MDA-MB-231 cells were obtained from Kozono, S. et al. dataset[32]. Pin1 (PDB: 1PIN) and Cdh1 (PDB: 4UI9) PDB data are used for docking models. The RNA sequencing data generated in this study have been deposited in the NCBI Gene Expression Omnbus (GEO) under accession codes GSE232285 and GSE232422. The remaining data are available within the Article, Supplementary Information or Source Data file. Source data are provided with this paper.

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

## Acknowledgements

We are grateful to the animal facility staff at Beth Israel Deaconess Medical Center (BIDMC) for assistance with in vivo experimentation. The work is supported by funding from the Ludwig Center at Harvard, the Breast Cancer Research Foundation (BCRF), SPORE grant 1P50CA168504, and 5R01CA226776 to G.M.W., R35CA253027 to W.W., 1K99CA263194 to F.D., R01CA205153 to K.P.L., N.S.G., and X.Z.Z., R01AG055559 to K.P.L., and X.Z.Z., R01GM137031 to Y.L., R35GM139572 to G.L., the Jane Coffin Child Memorial Fund for Medical Research fellowship to J.-Y.C., the Canada Foundation for Innovation (CFI) grants (#43257 and #43822) and Canadian Institutes of Health Research (#502128, #506475 and #506500), Ontario Institute for Cancer Research grant (#1240) and philanthropic donation from Keith Samitt to K.P.L., and X.Z.Z.

## Author contributions

Conceptualization: S.K., F.D., X.Z.Z., W.W., G.M.W. and K.P.L.; Methodology: S.K., F.D., X.Z.Z., W.W., G.M.W. and K.P.L.; Investigation: S.K., F.D., L.W., J.Y.C., M.T.N., W. L., A.T., N.K., N.M.N., H.S., W. T., C.Q., K.K., F. B., E.S.G., D.A.I., J.M.P., and X.W.; M.T.N. and W. L. contributed equally; Writing – Original Draft: S.K. and G.M.W.; Writing – Review & Editing, S.K., F.D., X.Z.Z., W.W., G.M.W. and K.P.L.; Resources: J.G.C., Y.J.H., G.L., Y. L. and G.S.N.; Project Administration: X.Z.Z., W.W., G.M.W., and K.P.L.;

Funding Acquisition: X.Z.Z., G.S.N. W.W., G.M.W. and K.P.L.; Supervision: X.Z.Z., W.W., G.M.W. and K.P.L.

## Competing interests

S.K., G.M.W., N.S.G., X.Z.Z., and K.P.L. are inventors of several issued patents and/or pending patent applications on PIN1, PIN1 biomarkers, PIN1 inhibitors and PIN1 inhibitor combination to treat human diseases; X.Z.Z. and K.P.L. are the scientific founders and former scientific advisors of and own equity in Pinteon. Their interests were reviewed and managed by BIDMC in accordance with its conflict-of-interest policy. G.M.W. reports research funding from Glaxo Smith Kline (institutional funding). W.W. is a co-founder and consultant for the Rekindle Therapeutics. N.S.G. is a founder, science advisory board member (SAB) and equity holder in Syros, C4, Allorion, Lighthorse, Inception, Voronoi, Matchpoint, Shenandoah (board member), Larkspur (board member) and Soltego (board member). The Gray lab receives or has received research funding from Novartis, Takeda, Astellas, Taiho, Jansen, Kinogen, Arbella, Deerfield and Sanofi. All other authors do not have any competing interests.

## Additional information

[1]Division of Hematology/Oncology, Department of Medicine and Cancer Research Institute, Beth Israel Deaconess Medical Center, Harvard Medical School, Boston, MA 02215, USA. [2]Department of Pathology, Beth Israel Deaconess Medical Center and Cancer Research Institute, Harvard Medical School, Boston, MA 02215, USA. [3]Department of Systems Biology, Harvard Medical School, Boston, MA 02215, USA. [4]Laboratory of Systems Pharmacology, Harvard Medical School, Boston, MA 02215, USA. [5]Department of Molecular Biology, Cell Biology & Biochemistry, Brown University, Providence, RI 02912, USA. [6]Yale Cancer Biology Institute, West Haven, CT 06516, USA. [7]Key Laboratory of Functional and Clinical Translational Medicine, Fujian Province University, Xiamen Medical College, Xiamen 361023, China. [8]Data Science & Artificial Intelligence, R&D, AstraZeneca, Gaithersburg, MD, USA. [9]Department of Genetics, Harvard Medical School, Boston, MA 02115, USA. [10]Department of Medicine, Division of Medical Oncology, Mayo Clinic, Phoenix, AZ, USA. [11]Molecular and Integrative Physiological Sciences, Department of Environmental Health, Harvard T.H. Chan School of Public Health, Boston, MA 02215, USA. [12]Preclinical Murine Pharmacogenetics Facility, Beth Israel Deaconess Medical Center, Harvard Medical School, Boston, MA 02215, USA. [13]Department of Pharmacology, Yale University School of Medicine, New Haven, CT 06510, USA. [14]Department of Chemical and Systems Biology, Chem-H and Stanford Cancer Institute, Stanford University, Stanford, CA 94305, USA. [15]Departments of Pathology and Laboratory Medicine, Biochemistry, and Oncology, and Lawson Health Research Institute, Schulich School of Medicine and Dentistry, Western University, London, ON N6A 3K7, Canada. [16]Departments of Biochemistry and Oncology, and Robarts Research Institute, Schulich School of Medicine and Dentistry, Western University, London, ON N6A 3K7, Canada. [17]These authors contributed equally: Shizhong Ke, Fabin Dang, Lin Wang, Jia-Yun Chen. ✉e-mail: xzhou659@uwo.ca; wwei2@bidmc.harvard.edu; gwulf@bidmc.harvard.edu; klu92@uwo.ca

