## [Peer Review File · Nature Communications]

Reviewers' Comments:

Reviewer #1:

Remarks to the Author:

In this MS, Ke, and colleagues provide evidence that the activity of the APC/CCDH1 tumor suppressor complex is controlled by a mechanism of reciprocal regulation between the CDH1 E3 ubiquitin ligase and the phosphorylation-dependent prolyl-isomerase PIN1, which is known to promote cancer by regulating many tumor suppressors and oncoproteins. In particular, in both untransformed and breast cancer (BC) cells, unphosphorylated CDH1 bound and caused degradation of PIN1 and reduction of other mitotic proteins. CDK4 phosphorylated CDH1 in the S163/P motif, as suggested by *in vitro* assays, and upon this modification, CDH1 could no longer cause PIN1 degradation. Instead, CDH1 remained phosphorylated and thus inactive, and this was associated with the stabilization of the APC substrate Geminin and other mitotic proteins. This mechanism was dependent on trans-to-cis isomerization of phosphoS163/P by PIN1 itself, as suggested by *in vitro* assays. Indeed, in BC cells, PIN1 KO reduced S/P phosphorylation in CDH1 (an effect attributed to dephosphorylation of trans phosphoS163/P by trans-selective phosphatase), thus promoting APC/CCDH1 activity, reduction of mitotic protein levels, and cell cycle arrest. Also, similar effects were observed upon treatment with PIN1 inhibitors, which led to CDH1-dependent degradation of PIN1.

A critical CDK4 substrate is the RB tumor suppressor, and indeed RB deficiency is known to confer BC resistance to CDK4 inhibitors. In this regard, Ke et al. provided evidence that, in both RB proficient and deficient BC cells, CDH1 was regulated by CDK4 in a PIN1-dependent manner, and, treatment with the CDK4/6 inhibitor Palbociclib reduced S/P phosphorylation in CDH1, increased CDH1/APC interaction, and decreased the levels of PIN1 and other mitotic proteins. Of note, these phenotypes were enhanced by treatments with PIN1 inhibitors, suggesting synergy between CDK4/6 and PIN1 inhibitors. Indeed, the combination of CDK4/6 and PIN1 inhibitors synergistically inhibited CDH1 activity and reduced tumor growth, in RB proficient and deficient mouse xenografts. These effects were also validated in immunocompromised allogeneic mouse models.

APC inactivation was previously proposed as crucial for cell cycle entry, independently of phosphoRB-E2F activation (Cappell et al. Cell 2016). Moreover, CDH1 was shown to be inactivated by proline-directed phosphorylation (references n. 9 and 14 in the manuscript). At the core of the work by Ke and colleagues, there is the unveiling of a PIN1-dependent layer of APC/CCDH1 regulation, with potential therapeutic implications for BC treatment.

Overall the findings described are interesting and many experiments are well conducted, however, I have some major concerns that need to be addressed by the authors to support their conclusions.

1) The authors investigate the CDK4/CDH1 and CDH1/PIN1 protein-protein interactions by co-immunoprecipitation of ectopically expressed proteins. In addition, super-resolution immunofluorescence suggested partial colocalization between endogenous PIN1 and CDH1 (ED Fig.1h). However, to support the proposed mechanism of reciprocal regulation between CDH1 and PIN1, it is important to demonstrate, by *in situ* proximity ligation assays, that CDK4/CDH1 and CDH1/PIN1 indeed interact in MCF10A and RB proficient and deficient BC cells. One would expect that these interactions change during the transition from the G1/S phase and the rest of the cell cycle and are perturbed by CDK4 and PIN1 inhibitors, and these changes should be verified.

2) Results shown in Fig. 4 and ED Fig. 3 are dedicated to supporting the notion that CDK4 phosphorylates CDH1 S163, with consequent trans-cis isomerization of Proline-164 residue, and are performed *in vitro*. The authors should provide additional evidence that this regulation occurs in untransformed and BC cells. This should include the demonstration that, in BC cells, the observed effects of CDK4 inhibitor (Palbociclib) on mitotic proteins and cell cycle progression depend on the inhibition of CDH1 S163 phosphorylation. One possibility could be to assess APC activity and cell cycle in cells edited (for instance through CRISPR/Cas9) in CDH1 with S163 not-phosphorylatable mutation, by which CDH1 which should not be isomerized by PIN1 (thus becoming constitutively active), and phosphomimetic mutation, by which CDH1 should be constitutively inactive.

3) In the last part of the manuscript, the authors assess the effect of combining CDK4/6 and PIN1 inhibitors in immunocompetent Brca;p53 mutant BC models. The authors base the rationale for this investigation on their own previous report showing that inhibition of PIN1 triggers anti-tumor immunity in pancreatic cancer mouse models (Koikawa et al, Cell 2021), and on evidence by others (Goel et al. Nature 2017) suggesting that inhibition of CDK4 triggers anti-tumor immunity. While providing interesting results, these analyses do not seem to be necessary to support the main conclusions of the manuscript i.e., that CDH1/PIN1 reciprocal regulation impacts BC proliferation.

4) Also, related to the analyses in the last part of the manuscript, in ED Fig. 8e, the authors provide single-sample gene set enrichment (ssGSEA) analysis of BC sample gene expression data from TCGA, suggesting upregulation of immune response in tumors with low PIN1 mRNA levels, compared with tumors showing high mRNA PIN1 levels. However, this seems at odds with the results in Fig. 1, where the author state that "we found a significant correlation between elevated levels of PIN1 protein, but not mRNA, with poor prognosis...". Moreover it is not indicated whether: i) the data were obtained from Cibiportal or GDC data portal; ii) how many samples were analyzed; iii) how many of these samples were obtained upon filtering for PIN1 mRNA levels, using as distribution cutoff the one standard deviation of the mean value. This information should be included in the methods section.

Minor issues:

- In Fig. 3b, authors should provide a clear description of all the pipeline steps used for the quantitative proteomic analysis in PIN1 KO MDA-MB-231 cells

Reviewer #2:

Remarks to the Author:

Summary-

This manuscript from Ke et al evaluates a reciprocal relationship between the prolyl isomerase PIN1 and the cell cycle ubiquitin ligase APC/C. PIN1 isomerizes protein around proline residues that are often phosphorylated as S/T residues by proline directed kinases, including the key cell cycle regulatory enzymes CDKs. APC/C is a large multi-subunit E3 that directs the degradation of proteins at mitotic exit and in quiescence. The authors provide evidence that PIN1 might be an APC/C substrate that is degraded by APC/C in combination with its substrate receptor protein Cdh1. Reciprocally, they also suggest that the phosphorylation of Cdh1 at S163 by CDK4/6 leads to a trans-to-cis prolyl isomerization by PIN1, thereby stabilizing phosphorylated Cdh1 protein and inactivating APC/C. Further they show that combining PIN1 inhibitors with CDK4/6i leads improved tumor growth inhibition in both RB deficient and RB proficient tumors models. The data showing synergy between the PIN1 inhibitors and CDK4/6 inhibitors is rather remarkable. In addition, the data suggesting that PIN1 binds and isomerizes CDH1 appears compelling. However, dampening my enthusiasm for the manuscript is the mechanistic data supporting the degradation of PIN1 by APC/C, which is less compelling, and also the data that indicates a critical role for the isomerization of Cdh1 with respect to APC/C function. PIN1 clearly has an impact on cell cycle progression, and this will undoubtedly influence the abundance of APC/C substrates. Disentangling these differences is admittedly a challenge, but the key experiments to demonstrate this are not evident in the proposed studies.

Major points

1. There is a lack of clear data supporting the degradation of PIN1 by the APC/C ubiquitin ligase. While there are several pieces of data pointing in this direction, most notably binding, it is possible that this is largely due to the impact of PIN1 on APC/C and not the impact of APC/C on PIN1. Key concerns include the fact that PIN1 is not degraded in or around mitosis, as is the case for nearly all known APC/C substrates. A simple nocodazole block and release experiment could test this. However, it not be the case in Fig 3c, when this was done in 231 cells? It does appear downregulated in arrested cells, but that could be through changes in expression, or another E3.

Moreover, the inability of Cdh1 KO cells to effectively arrest could account for the differences in PIN1 in inhibitor treated or arrest cells. Of further concern is the PIN1 mutant that is used to test its ubiquitination by APC/C. The PIN1 mutant is made in a highly structured portion of the protein, and several amino acid changes were made in making the mutant, so that rather than just changing the key D-box residues, several other changes were made. A key test would be to show that the mutant is stable in cells exiting mitosis. Finally, it is remarkable that the CDH1-7A mutant, which is suggested to control PIN1 has no effect on its ubiquitination.

2. There is no evidence that PIN1 prevents CDH1 dephosphorylation, as is stated in the results. While this may be true of other PIN1 substrates, and while the S163 site of Cdh1 appears important for binding PIN1, no evidence is presented that the isomerization around that site has an impact on dephosphorylation.

3. Similarly, there is no evidence that PIN1 controls the ability of CDH1 to bind substrates in late G1-phase, as is also stated in the text. No experiments determining an isomerized form of Cdh1 to bind a substrate is performed. This experiment is deeply complicated in vivo because PIN1 does impact overall proliferation, which influences APC/C activity.

4. The data supporting a shorter half life of PIN1 in CDH1-7A expressing cells treated with CHX is not clear (see Fig 2i). Since the starting levels of PIN1 are different between the controls and CDH1-expressing cells, it is hard to tell if there is indeed a change in half life. This experiment is also complicated by the fact that the 7A expressing cells are arrested, as is shown in the manuscript, complicating interpretation.

5. Similarly, Fig 3 C and D are tough to interpret since cells should be arrested in the PIN1 Ko cells. In addition, PIN1 does not appear to cycle in this experiment, confusing the interpretation that it is an APC/C substrate.

6. Overall, the data in Fig 3 are convincing that PIN1 plays a role in cell cycle progression into S-phase. However, there are myriad ways that this could be achieved, including alterations in growth factors signaling, stress, damage, etc. PIN1 likely controls the isomerization of many, many proteins, and data suggesting that this is, in large part, due to the single phosphorylation of Cdh1, is not clear.

7. The CHX data showing that the Cdh1-S163 mutants show differential stability is not clear. Since the starting levels of CDH1 are different between mutant and controls, and because the decay rates look strikingly similar, it is difficult to assess how the phosphorylation at that site is impacting stability. See extended Fig 3F, G, and H. A more direct test would be to deplete PIN1 and manipulate that site on CDH1.

8. The suggestion that there are two modes of binding- that is, between PIN1 through its isomerization domain and phosphorylated Cdh1 (late G1), and in the absence of Cdh1 phosphorylation, through binding which would trigger PIN1 degradation (early G1). However, the PIN1 W34 mutant completely blocks binding, rather than shifting it to the other mode. If the CDH1-7A can still bind PIN1, why wouldn't this also be the case for the PIN1 W34 mutant? Would you not expect blocking binding completely to require mutating both sites on Cdh1 and Pin1? Or, at least, that each mutant alone would have only a partial effect, and that together, it would be made worse, which is not what is seen.

Minor Points

1. Go back and forth between saying APC and APC/C

2. The kinase assay shown in Fig 4B is confusing for several reasons. First, there is no explanation for how it was performed in either the results or the figure legend. Second, the source of CDH1 is not indicated, which is important since it is unlikely it could be produced in bacteria. Was it made in SF9 cells, or IPed from human cells? Third, what is PNB? Fourth, what is the source of the cyclin-CDK complex?

3

Reviewer #3:

Remarks to the Author:

PIN1 overexpression drives cell cycle progression and tumor malignancy, APC complex stops it by suppressing reaccumulation of mitotic cyclins and thereby stabilizing G1 phase. The authors argue

that natural antagonism results in irreversible PIN1 degradation which forces permanent cell cycle exit. Overall, while the results of this work are of some interest in understanding the interplay between cell cycle exit and entry, there are some major concerns that must be addressed, that are listed below. The take-away mechanistic logic surrounding CDH1-dependent inhibition of PIN1 is also unclear from the text.

Major Concerns:

- 1) PIN1 oe as defined in Fig 1 appears to occur in a very small subset of patients. This calls into question the translational impact/clinical relevance of the work presented in this manuscript.
- 2) Given the low numbers of patients with HER2 homogeneity in each cohort, a power analysis should be included to test whether the sample size provides adequate power for the analyses conducted. A second validation dataset is critical.
- 3) It is likely that CDH1 is the sole E3 ligase responsible for PIN1 degradation. In Extended data figure 1 (K-T) none of the knockdown constructs are validated to show knockdown of the intended target. If the authors claim that CDH1 is the sole, or primary E3 ligase for PIN1, these experiments should be conducted.
- 4) In Figure 4, the authors find that CDK2 IPs with CDH1 in addition to CDK4. Furthermore, Palbo has been found to modulate CDK2 (Pack et al. 2021). Therefore, it seems possible that the effects of palbo treatment might be modulated by inhibition of CDK2 in addition to CDK4. The authors should acknowledge this as a limitation and be mindful of this possibility when interpreting their results, for example changes in CDH1-phosphorylation.
- 5) In several instances throughout the manuscript, the authors state that there are potent or significant changes. Most instances have adequate statistical information, but there are several important exceptions that should be included for Figures: 6i, Ext. data 3b, Ext. data 5k, and Ext. data 8c (heatmap units).
- 6) If Pin1 irreversibly inhibits Cdh1 (such that phosphatases cannot recognize and dephosphorylate Cdh1), then how is Cdh1 thought to be able to inhibit Pin1? Is the idea that new Cdh1 would need to be synthesized? The mechanistic aspect should be more clearly articulated.
- 7) As a point of experimental rigor, there are some experiments in which SUM159 cell lines are used instead of the cell lines that have been used for most major results presented in this work. The use of these cells for specific experiments, and not more generally, should be explained.

Minor Concerns:

- 1) There appears to be cross-over effects with the RNAi knocking down CDH1 and Cdc20 in Fig 1. Rescue experiments would help clarify the specificity of knockdown better.
- 2) Survival curve y-axes need more specific labeling, and better explanation in the text. It is also interesting that PIN1 mRNA levels appear to be non-correlated with survival whereas protein levels are for this particular dataset. Are the survival curves for Figure 1a (n=65) and Extended figure 1a (n=59) from the same patient dataset? If so, why are the number of total patients represented different between the two and are the 9 patients represented in the mRNA survival curves represented in the protein survival curves?
- 3) Figure 1f has labeling inconsistencies.
- 4) For Figure 1g, it looks like KD of CDC20 was not very robust in MDA cells and that it did partially rescue PIN1 levels. This should be acknowledged in the text.
- 5) In the interpretations for Extended data figure 2f, the authors indicate that Cdh1-KO does not affect Pin1 mRNA levels. However, there appears to be a clear and relatively consistent decrease in Pin1 mRNA levels for the Cdh1-KO cells compared to WT.
- 6) The serum starvation does not look as strong for the Cdh1-KO MEF cell line. Are there differences in the arresting efficiencies between WT and Cdh1-KO cells?
- 7) It is unclear whether or not the cells depicted in Figure 2c-d express a WT allele of CDH1 in addition to the 7A mutant copy. Are the effects of the 7A mutant a result of overexpression of CDH1 in general or is it unique to over expression of the 7A mutant?
- 8) The legend for Figure 2 indicates that there are representative graphs for panel L, but these appear to be missing in the submitted version.
- 6) The antibody used to detect CDH1 phosphorylation in Figure 5a (pSP) should be defined in the methods.
- 9) Specifically in reference to Ext. data 4m, the authors should describe the justification for why they believe that CDK inhibitor and PIN1 inhibitors operate in a synergistic manner, rather than additive.

Reviewer #4:

Remarks to the Author:

CDH1 plays an important role in regulating cell cycle entry and thus is a target of interest for a variety of cancers. CDH1 is regulated by proline-directed serine phosphorylation, the outcomes of which are regulated by prolyl isomerase PIN1 activity, making the results of this manuscript of potentially high significance.

I was not able to comprehensively review this manuscript for scientific rigor, so this review will focus exclusively on the NMR spectroscopy and molecular docking studies that establish a model for CDH1 interactions with PIN1. The insights from this work, as reported in Figure 4, make an important contribution to the overall manuscript. The rigor of the presentation could be improved through the consideration of the following specific points.

1. The presentation of key regions from the ^{15}N -HSQC of PIN1 and ^{13}C -HSQC of P164-labeled CHD1 are helpful to the reader but inclusion of the full spectra as an Extended Data figure would also be beneficial for reuse and extended interpretation of the results.
2. The methods do not completely describe how the CSPs presented in Figure 4D were calculated. There is no indication that the bound-state chemical shifts were completely determined, so one assumes these represent minimal chemical shift perturbations inferred by comparison of the spectra. To remove the perception of investigator bias, the means for selecting the peak assigned to each residue in the bound-state spectrum and the equation used to calculate the reported CSP should be included.
3. The CD-HD correlations displayed in Figure 4G are well resolved and therefore reasonable as a source of cis/trans quantitation. On the other hand, the CB and CG resonance positions are far more unique in the trans-proline and cis-proline conformations, making them the fields preferred benchmark for the presence of prolyl isomerization. At a minimum, drawing attention to the relative intensity of these resonances in the Extended Data would be helpful.

Response to Reviewers' Comments:

We are grateful for the valuable and constructive feedback provided by all four reviewers. The reviewers are generally enthusiastic about our paper. **Reviewer 1:** "Overall the findings described are interesting and many experiments are well conducted." **Reviewer 2:** "The data showing synergy between the PIN1 inhibitors and CDK4/6 inhibitors is rather remarkable. ... that PIN1 binds and isomerizes CDH1 appears compelling." **Reviewer 3:** "... the results of this work are of some interest in understanding the interplay between cell cycle exit and entry." **Reviewer 4:** "..., the outcomes of which (CDH1) are regulated by prolyl isomerase PIN1 activity, making the results of this manuscript of potentially high significance". However, the reviewers also raised important issues that need to be addressed. We have performed all suggested experiments, and added a significant amount of new data, including 10 new main Figure panels and 23 new Extended Data Figure panels and 2 Supplementary Tables. Notably, we have (1) presented an additional clinical dataset comprising 160 BC patient samples, validating that the PIN1-high group exhibits a worse prognosis than the PIN1-low group; (2) provided more substantial evidence for the direct in situ interaction between PIN1 and CDH1; (3) deepened our mechanistic understanding of the effects of APC/C^{CDH1} on PIN1 degradation, and (4) determined the direct role of PIN1 in regulating the phosphorylation of CDH1. With these extensive revisions, we believe that our paper has been substantially strengthened.

Reviewer #1 – PIN1, cancer

In this MS, Ke, and colleagues provide evidence that the activity of the APC/CCDH1 tumor suppressor complex is controlled by a mechanism of reciprocal regulation between the CDH1 E3 ubiquitin ligase and the phosphorylation-dependent prolyl-isomerase PIN1, which is known to promote cancer by regulating many tumor suppressors and oncoproteins. In particular, in both untransformed and breast cancer (BC) cells, unphosphorylated CDH1 bound and caused degradation of PIN1 and reduction of other mitotic proteins. CDK4 phosphorylated CDH1 in the S163/P motif, as suggested by vitro assays, and upon this modification, CDH1 could no longer cause PIN1 degradation. Instead, CDH1 remained phosphorylated and thus inactive, and this was associated with the stabilization of the APC substrate Geminin and other mitotic proteins. This mechanism was dependent on trans-to-cis isomerization of phosphoS163/P by PIN1 itself, as suggested by in vitro assays. Indeed, in BC cells, PIN1 KO reduced S/P phosphorylation in CDH1 (an effect attributed to dephosphorylation of trans phosphoS163/P by trans-selective phosphatase), thus promoting APC/CCDH1 activity, reduction of mitotic protein levels, and cell cycle arrest. Also, similar effects were observed upon treatment with PIN1 inhibitors, which led to CDH1-dependent degradation of PIN1.

A critical CDK4 substrate is the RB tumor suppressor, and indeed RB deficiency is known to confer BC resistance to CDK4 inhibitors. In this regard, Ke et al. provided evidence that, in both RB proficient and deficient BC cells, CDH1 was regulated by CDK4 in a PIN1-dependent manner, and, treatment with the CDK4/6 inhibitor Palbociclib reduced S/P phosphorylation in CDH1, increased CDH1/APC interaction, and decreased the levels of PIN1 and other mitotic proteins. Of note, these phenotypes were enhanced by treatments with PIN1 inhibitors, suggesting synergy between CDK4/6 and PIN1 inhibitors. Indeed, the combination of CDK4/6 and PIN1 inhibitors synergistically inhibited CDH1 activity and reduced tumor growth, in RB proficient and deficient mouse xenografts. These effects were also validated in immunocompromised allogeneic mouse models.

APC inactivation was previously proposed as crucial for cell cycle entry, independently of phosphoRB-E2F activation (Cappell et al. Cell 2016). Moreover, CDH1 was shown to be inactivated by proline-directed phosphorylation (references n. 9 and 14 in the manuscript). At the core of the work by Ke and colleagues, there is the unveiling of a PIN1-dependent layer of APC/CDH1 regulation, with potential therapeutic implications for BC treatment.

Overall the findings described are interesting and many experiments are well conducted, however, I have some major concerns that need to be addressed by the authors to support their conclusions.

Response: We appreciate the reviewer's enthusiastic and insightful comments on our paper. Below, please find our detailed responses addressing each of the concerns raised.

1) The authors investigate the CDK4/CDH1 and CDH1/PIN1 protein-protein interactions by co-immunoprecipitation of ectopically expressed proteins. In addition, super-resolution immunofluorescence suggested partial colocalization between endogenous PIN1 and CDH1 (ED Fig.1h). However, to support the proposed mechanism of reciprocal regulation between CDH1 and PIN1, it is important to demonstrate, by in situ proximity ligation assays, that CDK4/CDH1 and CDH1/PIN1 indeed interact in MCF10A and RB proficient and deficient BC cells. One would expect that these interactions change during the transition from the G1/S phase and the rest of the cell cycle and are perturbed by CDK4 and PIN1 inhibitors, and these changes should be verified.

Response: Following the expert advice from the reviewer, we have conducted in situ PLA assays to show the interactions of endogenous CDH1-PIN1 and CDH1-CDK4 in human BC cells, as well as in non-transformed epithelial cells (**Extended Data Fig. 3a, 5a**). As expected by the reviewer, our new results revealed a substantial augmentation in PLA signals for PIN1-CDH1 and CDK4-CDH1 when the cells progress into the S/G2 phase, signifying an elevated expression and interactions of PIN1-CDH1 and CDK4-CDH1.

Furthermore, PLA assays indicate a marked reduction in the expression and interactions of PIN1-CDH1 and CDK4-CDH1 following treatment with a combination of PIN1 and CDK4 inhibitors (Fig. 5e, f).

2) Results shown in Fig. 4 and ED Fig. 3 are dedicated to supporting the notion that CDK4 phosphorylates CDH1 S163, with consequent trans-cis isomerization of Proline-164 residue, and are performed in vitro. The authors should provide additional evidence that this regulation occurs in untransformed and BC cells. This should include the demonstration that, in BC cells, the observed effects of CDK4 inhibitor (Palbociclib) on mitotic proteins and cell cycle progression depend on the inhibition of CDH1 S163 phosphorylation. One possibility could be to assess APC activity and cell cycle in cells edited (for instance through CRISPR/Cas9) in CDH1 with S163 not-phosphorylatable mutation, by which CDH1 which should not be isomerized by PIN1 (thus becoming constitutively active), and phosphomimetic mutation, by which CDH1 should be constitutively inactive.

Response: As per the reviewer's guidance, we conducted an assessment of APC/C activity and cell cycle progression in cells expressing CDH1-S163A or CDH1-S163E mutations. To this end, we introduced the CDH1-S163A or CDH1-S163E mutations into *CDH1* knockout MCF-7 and MCF-10A cells. When compared to cells with the wild-type CDH1 (CDH1-WT), those with CDH1-S163A exhibited G1 arrest and demonstrated enhanced APC/C activity, as evidenced by the reduced fluorescence intensity of APC/C-degron reporter (Fig. 4c, Extended Data Fig. 5c). In contrast, cells carrying the CDH1-S163E mutation displayed decreased APC/C activity (Fig. 4c, Extended Data Fig. 5c). Moreover, cells bearing the CDH1-S163E mutation notably impeded the CDK4 inhibitor-induced reduction in mitotic proteins. These findings underscore the impact of CDK4 inhibitors on CDH1-S163 (Fig. 4d, Extended Data Fig. 5d).

3) In the last part of the manuscript, the authors assess the effect of combining CDK4/6 and PIN1 inhibitors in immunocompetent Brca;p53 mutant BC models. The authors base the rationale for this investigation on their own previous report showing that inhibition of PIN1 triggers anti-tumor immunity in pancreatic cancer mouse models (Koikawa et al, Cell 2021), and on evidence by others (Goel et al. Nature 2017) suggesting that inhibition of CDK4 triggers anti-tumor immunity. While providing interesting results, these analyses do not seem to be necessary to support the main conclusions of the manuscript i.e., that CDH1/PIN1 reciprocal regulation impacts BC proliferation.

Response: We agree with the reviewer's perspective. The potential mechanism underlying the immune response elicited by the combination treatment in immunocompetent mouse models warrants more comprehensive investigation in future studies. It is worth emphasizing, however, that we screened a library of genes associated with tumor immunity and our findings hint at a connection between the constitutively active

CDH1-S163A and the upregulation of genes associated with inflammatory responses, chemotaxis, and adaptive immunity, **as shown on the right**. Moreover, immunofluorescent analysis of tumors revealing a significant decrease in the levels of PIN1 and other mitotic proteins in tumors from mice treated with CDK4 and PIN1 inhibitors, consistent with APC/C^{CDH1} activation (**Fig. 6i, I**). This implies that the immune response triggered by CDK4 and PIN1 inhibitors

may, to some extent, be modulated through CDH1 activation. Nonetheless, a deeper understanding of this mechanism and its implications necessitates further exploration.

4) Also, related to the analyses in the last part of the manuscript, in ED Fig. 8e, the authors provide single-sample gene set enrichment (ssGSEA) analysis of BC sample gene expression data from TCGA, suggesting upregulation of immune response in tumors with low PIN1 mRNA levels, compared with tumors showing high mRNA PIN1 levels. However, this seems at odds with the results in Fig. 1, where the author state that “we found a significant correlation between elevated levels of PIN1 protein, but not mRNA, with poor prognosis...”. Moreover it is not indicated whether: i) the data were obtained from Cibiportal or GDC data portal; ii) how many samples were analyzed; iii) how many of these samples were obtained upon filtering for PIN1 mRNA levels, using as distribution cutoff the one standard deviation of the mean value. This information should be included in the methods section.

Response: We apologize for the missing information in our manuscript. Here are the details regarding the single-sample gene set enrichment analysis (ssGSEA). We obtained the breast cancer (BRCA) mRNA data from The Cancer Genome Atlas (TCGA) through the Broad FireBrowse platform (http://firebrowse.org/?cohort=BRCA&download_dialog=true). To investigate the relationship between PIN1 mRNA levels and immune response, we employed a stringent cutoff based on the distribution of PIN1 mRNA levels across BC tumors (n=526). Specifically, we used a Z-score cutoff of >1.5 for the PIN1 high group and <-1.5 for the PIN1 low group. Approximately 50 samples were used for the ssGSEA analysis to assess the upregulation of immune response in tumors with low PIN1 mRNA levels compared to tumors with high PIN1 mRNA levels. We have now included this additional information in the methods section of our manuscript for clarity. It's important to note that while our ssGSEA analysis indicated an upregulation of immune response in tumors with low PIN1 mRNA levels, the correlation between immune response and prognosis can be complex.

Immune response is just one of many factors that contribute to a patient's overall survival, which depends on several factors including the stage of cancer, the specific immune response markers, and individual patient characteristics.

Minor issues:

- In Fig. 3b, authors should provide a clear description of all the pipeline steps used for the quantitative proteomic analysis in PIN1 KO MDA-MB-231 cells

Response: We appreciate the reviewer's request for clarification. The quantitative proteomic data for PIN1 KO MDA-MB-231 cells presented in Fig. 3b were originally generated and detailed in a prior publication (Kozono et al. Nat Commun. 2018;3069). In that earlier work, we provided an exhaustive description of the quantitative proteomic analysis methods in the Materials and Methods section. For our current study, we leveraged the processed proteomic data from the previous publication to conduct a Gene Ontology (GO) enrichment analysis of differential protein expressions between PIN1 KO and WT MDA-MB-231 cells. This analysis was performed using the DAVID functional annotation tools, and the results are presented in Fig. 3b to demonstrate the differential biological processes between PIN1 KO and WT MDA-MB-231 cells. We cited this data source within the corresponding description in our current manuscript.

Reviewer #2 – Cell cycle, APC, proteasome

Summary-

This manuscript from Ke et al evaluates a reciprocal relationship between the prolyl isomerase PIN1 and the cell cycle ubiquitin ligase APC/C. PIN1 isomerizes protein around proline residues that are often phosphorylated as S/T residues by proline directed kinases, including the key cell cycle regulatory enzymes CDKs. APC/C is a large multi-subunit E3 that directs the degradation of proteins at mitotic exit and in quiescence. The authors provide evidence that PIN1 might be an APC/C substrate that is degraded by APC/C in combination with its substrate receptor protein Cdh1. Reciprocally, they also suggest that the phosphorylation of Cdh1 at S163 by CDK4/6 leads to a trans-to-cis prolyl isomerization by PIN1, thereby stabilizing phosphorylated Cdh1 protein and inactivating APC/C. Further they show that combining PIN1 inhibitors with CDK4/6i leads improved tumor growth inhibition in both RB deficient and RB proficient tumors models. The data showing synergy between the PIN1 inhibitors and CDK4/6 inhibitors is rather remarkable. In addition, the data suggesting that PIN1 binds and isomerizes CDH1 appears compelling. However, dampening my enthusiasm for the manuscript is the mechanistic data supporting the degradation of PIN1 by APC/C, which is less compelling, and also the data that indicates a critical role for the isomerization of

Cdh1 with respect to APC/C function. PIN1 clearly has an impact on cell cycle progression, and this will undoubtedly influence the abundance of APC/C substrates. Disentangling these differences is admittedly a challenge, but the key experiments to demonstrate this are not evident in the proposed studies.

Response: We sincerely thank the reviewer for the detailed and constructive feedback on our study. The perspective on the interplay between PIN1 and APC/C is greatly valued. As expertly advised, we have now addressed these points with additional experimentation and clarity in our manuscript, as described below.

Major points

1. There is a lack of clear data supporting the degradation of PIN1 by the APC/C ubiquitin ligase. While there are several pieces of data pointing in this direction, most notably binding, it is possible that this is largely due to the impact of PIN1 on APC/C and not the impact of APC/C on PIN1. Key concerns include the fact that PIN1 is not degraded in or around mitosis, as is the case for nearly all known APC/C substrates. A simple nocodazole block and release experiment could test this. However, it not be the case in Fig 3c, when this was done in 231 cells? It does appear downregulated in in arrested cells, but that could be through changes in expression, or another E3. Moreover, the inability of Cdh1 KO cells to effectively arrest could account for the differences in PIN1 in inhibitor treated or arrest cells. Of further concern is the PIN1 mutant that is used to test its ubiquitination by APC/C. The PIN1 mutant is made in a highly structured portion of the protein, and several amino acid changes were made in making the mutant, so that rather than just changing the key D-box residues, several other changes were made. A key test would be to show that the mutant is stable in cells exiting mitosis. Finally, it is remarkable that the CDH1-7A mutant, which is suggested to control PIN1 has no effect on its ubiquitination.

Response:

1. Thank you for pointing out the concern regarding the data supporting PIN1 degradation by the APC/C ubiquitin ligase and the potential complications introduced by the PIN1 mutant used in our study. In response to this, we have now conducted ubiquitination assays, demonstrating that the wild-type PIN1 underwent ubiquitination by transfecting with a gradient increase of constitutively active CDH1 constructs (**Fig. 2f**). Moreover, when the D-box of PIN1 was mutated, this resulted in the abrogation of APC/C^{CDH1}-mediated ubiquitination (**Fig. 2k**). Moreover, our new data also suggests that this D-box mutant of PIN1 was more stable throughout the cell cycle (**Extended Data Fig. 4g**). We believe these findings directly address the concerns raised and further support our conclusions.
2. Regarding the concern about the dynamics of PIN1 throughout the cell cycle in tumor cells, considering the complexity of cell cycle regulation, it is possible that PIN1 may be subject to regulation by factors beyond APC/C^{CDH1} throughout the cell cycle.

Nevertheless, we have presented compelling evidence demonstrating APC/C^{CDH1} as a bona fide E3 ligase for PIN1 during the G1 phase in non-tumor cells (**Fig. 1h-j, Extended Data Fig. 3**), which have low CDK4 activity (**Fig. 2b**). Similarly, in nocodazole block and release experiment, PIN1 and other mitotic proteins were relatively low in G1 and started to accumulate at the onset of the S phase, coincident with the inactivation of APC/C^{CDH1}. It is worth noting that the interaction dynamics between PIN1 and APC/C^{CDH1} diverge significantly from those observed with other known substrates. In our study, we observed that PIN1 possessed the ability to reciprocally inhibit APC/C^{CDH1} activity, a phenomenon that was particularly pronounced in tumor cells due to elevated CDK4 activity (**Fig. 2b**). This results in a uniquely distinct regulation pattern for PIN1 across the cell cycle compared to other recognized APC/C^{CDH1} substrates especially in tumor cells.

3. We appreciate the reviewer's insight highlighting the inability of *CDH1* KO cells to effectively arrest could account for the differences in PIN1 inhibitor treated cells. As noted by the reviewer, this does provide substantive support to our notion that PIN1 inhibitors profoundly impact CDH1 while having minimal effects on other cell cycle effectors (**Fig. 5g-i, Extended Data Fig. 7**). Furthermore, our findings show that treatment with PIN1 and CDK4 inhibitors led to PIN1 polyubiquitination, which was notably attenuated in *CDH1* KO cells (**Fig. 5j**). These results reinforce the intricate and specific relationship between PIN1 and CDH1, emphasizing the critical role of CDH1 in this regulation.

2. There is no evidence that PIN1 prevents CDH1 dephosphorylation, as is stated in the results. While this may be true of other PIN1 substrates, and while the S163 site of Cdh1 appears important for binding PIN1, no evidence is presented that the isomerization around that site has an impact on dephosphorylation.

Response: We have performed a series of experiments to address this point as following.

1. Co-IP and site-directed mutagenesis assays showed that PIN1 preferentially interacted with CDH1-pS163 (**Fig. 4e, Extended Data Fig. 5g**).
2. NMR analysis showed that PIN1 bound to and catalyzed the *trans* to *cis* prolyl-isomerization of the CDH1-pS163-P motif, increasing the population of *cis*-P164 isomer from 7% to 14.2%, thereby stabilizing phosphorylated CDH1 (**Fig. 2f-i**), especially given that CDH1-specific phosphatase would not engage with *cis* proline (Gray et al., EMBO J. 2003, 22: 3524).
3. In our subsequent analyses, it became evident that a PIN1 knockout markedly diminished CDH1 phosphorylation. This led to an increased affinity of CDH1 to the APC/C complex (**Fig. 5a**). Similarly, PIN1 inhibition, which prevents of *trans* to *cis* isomerization, resulted in CDH1 dephosphorylation (**Extended Data Fig. 6b**). In this experiment, we employed a pull-down method for CDH1 and assessed its phosphorylation status by using anti-Phospho-MAPK/CDK Substrates (PXS*P or

S*PXR/K) antibody. This antibody is phosphoserine-specific, and does not react with phospho-threonine- or phospho-tyrosine-containing proteins.

4. Our data showed that the S->A phospho-deficient mutation at CDH1-S163 eliminated its binding to the PIN1 WW domain but enhanced its association with the APC/C complex. In contrast, the S->E phosphomimic mutation at CDH1-S163 reinstated its binding to PIN1 while reducing its interaction with the APC/C complex (**Fig. 4k**). This underscores the influential role of PIN1 in modulating CDH1-S163, thereby regulating APC/C^{CDH1} activity.
5. Lastly, we have examined the phosphorylation status of CDH1-S163 in more depth. By immunoprecipitating CDH1 and subsequently assessing its phosphorylation through Mass spectrometry, we observed that the phosphorylation levels of CDH1-S163 were almost two-fold reduced in *PIN1* KO cells in comparison to WT cells (**Fig. 4j, Supplementary Table 5**). A thorough description of our detection methodology can be found in the Methods section.

3. Similarly, there is no evidence that PIN1 controls the ability of CDH1 to bind substrates in late G1-phase, as is also stated in the text. No experiments determining an isomerized form of Cdh1 to bind a substrate is performed. This experiment is deeply complicated in vivo because PIN1 does impact overall proliferation, which influences APC/C activity.

Response: We appreciate the reviewer's insightful comments and observations. Indeed, the impact of PIN1 on overall proliferation complicates the direct in vivo examination of its role in regulating APC/C activity. Nevertheless, our studies have provided compelling evidence that sheds light on this relationship.

Our data showed that *PIN1* knockout significantly reduced CDH1 phosphorylation. This led to an increased affinity of CDH1 to the APC/C complex (**Fig. 5a**). Similarly, PIN1 inhibition, which prevents of *trans* to *cis* isomerization, resulted in CDH1 dephosphorylation (**Extended Data Fig. 6b**). We also demonstrated that the S->A phospho-deficient mutation at CDH1-S163 eliminated its binding to the PIN1 WW domain, yet enhanced its association with the APC/C complex. In contrast, the S->E phosphomimic mutation at CDH1-S163 reinstated its binding to PIN1 while reducing its interaction with the APC/C complex (**Fig. 4e, 4k**). Furthermore, our data revealed that PIN1 KO decreased phosphorylation of CDH1-S163 (**Fig. 4j, Supplementary Table 5**) and increased APC/C activity in CDH1-WT cells, but not in CDH1-S163A mutant (**Extended Data Fig. 5k**). These data support the notion that PIN1 regulates APC/C activity largely through its interaction with phosphorylated S163 of CDH1.

4. The data supporting a shorter half life of PIN1 in CDH1-7A expressing cells treated with CHX is not clear (see Fig 2i). Since the starting levels of PIN1 are different between the controls and CDh1-expressing cells, it is hard to tell if there

is indeed a change in half life. This experiment is also complicated by the fact that the 7A expressing cells are arrested, as is shown in the manuscript, complicating interpretation.

Response: We totally agree with the reviewer's opinion. We recognize the potential ambiguities in interpretation. Consequently, we have opted to exclude this particular data from the revised version, prioritizing clarity and rigor. Nevertheless, we'd like to highlight that our other data robustly indicate that constitutively active APC/C^{CDH1} targeted PIN1, along with other mitotic proteins, for degradation, facilitating the arrest of cells in the G1 phase (**Fig. 2**). We value the reviewer's scrutiny and hope our revisions more clearly reflect the findings and implications of our work.

5. Similarly, Fig 3 C and D are tough to interpret since cells should be arrested in the PIN1 Ko cells. In addition, PIN1 does not appear to cycle in this experiment, confusing the interpretation that it is an APC/C substrate.

Response: To provide clarity and further illustrate the direct regulatory relationship between PIN1 and CDH1, we conducted several experiments which are outlined below:

1. Beyond the known substrates of APC/C^{CDH1}, we examined the degradation of an APC/C^{CDH1} reporter, mCherry conjugated to a Geminin peptide (aa1-110). The promoter of this reporter remains unregulated, so reporter degradation is primarily regulated by APC/C^{CDH1} (**Extended Data Fig. 3b**), which has been widely used as indicator of APC/C^{CDH1} activity in cells (Cappell et al. Cell. 2016;167). We observed a dramatic decrease in mCherry-Geminin (1-110) levels in *PIN1* KO cells (**Fig. 3g**), indicating that *PIN1* KO enhances APC/C^{CDH1} activity.
2. Our data showed that *PIN1* KO significantly reduced CDH1 phosphorylation. This led to an increased affinity of CDH1 to the APC/C complex (**Fig. 4j, k, Fig. 5a**). Similarly, PIN1 inhibition, which prevents of *trans* to *cis* isomerization, resulted in CDH1 dephosphorylation (**Extended Data Fig. 6b**).
3. We also demonstrated that the S->A phospho-deficient mutation at CDH1-S163 eliminated its binding to the PIN1 WW domain but enhanced its association with the APC/C complex. In contrast, the S->E phosphomimic mutation at CDH1-S163 reinstated its binding to PIN1 while reducing its interaction with the APC/C complex (**Fig. 4e, k**). Furthermore, our data revealed that *PIN1* KO significantly increased APC/C activity in CDH1-WT cells, but not in CDH1-S163A mutant (**Extended Data Fig. 5k**). These data support PIN1 regulate APC/C activity largely through its interaction with phosphorylated S163 of CDH1.
4. We have presented compelling evidence demonstrating APC/C^{CDH1} as a bona fide E3 ligase for PIN1 during the G1 phase in non-tumor cells (**Fig. 1h-j, Extended Data Fig. 3**), which have low CDK4 activity (**Fig. 2b**). It is worth noting that the interaction dynamics between PIN1 and APC/C^{CDH1} diverge significantly from those observed with other known substrates. In our study, we observed that PIN1 possesses the ability to reciprocally inhibit APC/C^{CDH1} activity, a phenomenon particularly pronounced in

tumor cells due to elevated CDK4 activity (**Fig. 2b**). This results in a uniquely distinct regulation pattern for PIN1 across the cell cycle compared to other recognized APC/C^{CDH1} substrates especially in tumor cells.

5. We conducted ubiquitination assays, demonstrating that the wild-type PIN1 undergoes ubiquitination by transfecting with a gradient increase of constitutively active CDH1 constructs (**Fig. 2f**). Moreover, when the D-box of PIN1 was mutated, this resulted in the abrogation of APC/C^{CDH1}-mediated ubiquitination (**Fig. 2k**). Additionally, our data suggests that this D-box mutant of PIN1 was more stable throughout the cell cycle (**Extended Data Fig. 4g**).

6. Overall, the data in Fig 3 are convincing that PIN1 plays a role in cell cycle progression into S-phase. However, there are myriad ways that this could be achieved, including alterations in growth factors signaling, stress, damage, etc. PIN1 likely controls the isomerization of many, many proteins, and data suggesting that this is, in large part, due to the single phosphorylation of Cdh1, is not clear.

Response: Thank you for highlighting the potential multifaceted roles of PIN1 in cell cycle progression. We agree with the perspective that PIN1, given its pleiotropic nature, may exert its effects via multiple substrates. However, in this study, our mass spectrometry and immunoprecipitation data demonstrated a prominent interaction between PIN1 and CDH1 (**Fig. 1e, 1f**). Furthermore, our examination revealed APC/C^{CDH1} as the only E3 ligase capable of degrading PIN1 among those tested (**Fig. 1g, Fig. 2, Extended Data Fig. 1, Extended Data Fig. 2, Extended Data Fig. 7**).

Our data demonstrate that CDH1's phosphorylation at S163 by CDK4 promoted CDH1 interaction with PIN1. Our findings also confirm CDH1-S163 as a target for CDK4 and subsequently a recognition site for PIN1 (**Fig. 4b-e, h-j**). Although we acknowledge that PIN1 might bind to multiple phosphorylation sites concurrently, our data showed its preferential binding to the Ser163-Pro-Arg site. This observation is consistent with previous study indicating that peptides containing the pSer-Pro-Arg motif exhibit elevated affinity towards PIN1 compared to other peptides (Yaffe et al., Science 1997; 278; 1957). Additionally, considering the critical location of the Ser163-Pro-Arg site, adjacent to crucial CDH1 binding sites for the APC/C complex, it exerts a notable influence on APC/C^{CDH1} activity. Our data also indicated that mutations at CDH1-S163 affect its interaction with the APC/C complex (**Fig. 4k**). We hope that these results provide a strong rationale behind our focus on the CDH1 phosphorylation site while maintaining a recognition of the multifaceted and extensive roles of PIN1.

7. The CHX data showing that the Cdh1-S163 mutants show differential stability is not clear. Since the starting levels of CDH1 are different between mutant and controls, and because the decay rates look strikingly similar, it is difficult to assess how the phosphorylation at that site is impacting stability. See extended Fig 3F, G,

and H. A more direct test would be to deplete PIN1 and manipulate that site on CDH1.

Response: We value the reviewer's observation and the concerns raised regarding the clarity of the CHX data related to the stability of the CDH1-S163 mutants. We recognize the ambiguity in assessing the impact of phosphorylation at that site on stability. In light of these concerns, we have removed the CHX data from the revised manuscript, emphasizing accuracy and interpretative clarity. As kindly suggested by the reviewer, we knocked out PIN1 to examine the association between CDH1 and the APC/C complex and also the APC/C^{CDH1} activity in CDH1 WT and mutant cells. Our findings revealed that PIN1 KO decreased phosphorylation of CDH1-S163 (**Fig. 4j, Supplementary Table 5**) and notably augmented the binding affinity between CDH1 and the APC/C complex in CDH1 WT cells, which was not observed in cells with CDH1-S163A or CDH1-S163E mutations (**Fig. 4k**). Furthermore, this resulted in a marked increase in APC/C^{CDH1} activity (**Extended Data Fig. 5k**). However, such enhanced interaction and activity were not observed in CDH1-S163 mutant cells (**Extended Data Fig. 5k**). These data further support our idea that the regulatory effect of PIN1 on APC/C activity is largely through its interaction with CDH1-S163.

8. The suggestion that there are two modes of binding- that is, between PIN1 through its isomerization domain and phosphorylated Cdh1 (late G1), and in the absence of Cdh1 phosphorylation, through binding which would trigger PIN1 degradation (early G1). However, the PIN1 W34 mutant completely blocks binding, rather than shifting it to the other mode. If the CDH1-7A can still bind PIN1, why wouldn't this also be the case for the PIN1 W34 mutant? Would you not expect blocking binding completely to require mutating both sites on Cdh1 and Pin1? Or, at least, that each mutant alone would have only a partial effect, and that together, it would be made worse, which is not what is seen.

Response: We value the reviewer's insightful comments. To gain a deeper understanding of the binding affinity between CDH1 and different domains of PIN1, we assessed the interactions through a GST pull-down experiment. Notably, in a prolonged exposure, we observed some interactions between CDH1-WT and the PIN1 PPI domain, **as depicted on the right**. However, when compared to the WW domain, the binding between CDH1 and the PPI domain displayed significantly weaker affinity, suggesting that the binding affinity of CDH1 to the D-box motif in the PPI domain is lower than its affinity to the WW domain. As a result, when CDH1 is in a phosphorylated state, it exhibits a predilection for the PIN1 WW domain. Conversely, when CDH1 undergoes extensive dephosphorylation, it potentially targets the PIN1 PPI domain, leading to its degradation. To address the specific point raised by the reviewer regarding the PIN1 W34 mutant: our current understanding

suggests that the PIN1 W34 mutation likely disrupts the primary binding mode, leading to the observed complete blockade in binding.

Minor Points

1. Go back and forth between saying APC and APC/C

Response: We've standardized our terminology and consistently used "APC/C" throughout the updated manuscript.

2. The kinase assay shown in Fig 4B is confusing for several reasons. First, there is no explanation for how it was performed in either the results or the figure legend. Second, the source of CDH1 is not indicated, which is important since it is unlikely it could be produced in bacteria. Was it made in SF9 cells, or IPed from human cells? Third, what is PNBM? Fourth, what is the source of the cyclin-CDK complex?

Response: In vitro kinase assay and related reagents were described in the Methods section. HA-tagged CDH1 WT and mutants were transfected into HEK293T cells, followed by being immunoprecipitated with monoclonal Anti-HA-Agarose antibody (Sigma, A2095). The purified HA-CDH1 proteins were then incubated with 500 μ M of ATP γ S (Abcam, ab138911) and 0.5 μ g of recombinant human cyclin D1+CDK4 proteins (Abcam, ab55695) in the kinase reaction buffer (50mM Tris-HCl, 10mM MgCl₂, 0.1mM EDTA, 2mM DTT, 0.01% Brij 35, pH 7.5) for 30 min at room temperature. Then adding 2 mM of PNBM (Abcam, ab138910) and allowing the alkylating reaction proceed for additional 2h at room temperature. The reaction was then terminated by adding 5x SDS loading buffer and boiled for 10 min. Samples were then subjected to IB using anti-Thiophosphate ester antibody (Abcam, ab92570).

Reviewer #3 (Remarks to the Author):

PIN1 overexpression drives cell cycle progression and tumor malignancy, APC complex stops it by suppressing reaccumulation of mitotic cyclins and thereby stabilizing G1 phase. The authors argue that natural antagonism results in irreversible PIN1 degradation which forces permanent cell cycle exit. Overall, while the results of this work are of some interest in understanding the interplay between cell cycle exit and entry, there are some major concerns that must be addressed, that are listed below. The take-away mechanistic logic surrounding CDH1-dependent inhibition of PIN1 is also unclear from the text.

Response: We are grateful for the reviewer's thoughtful critique and recognition of the importance of our work in shedding light on the interplay between cell cycle exit and entry. In light of your feedback, we are committed to clarifying these aspects and addressing the issues in a comprehensive manner in the revised manuscript.

Major Concerns:

1) PIN1 oe as defined in Fig 1 appears to occur in a very small subset of patients. This calls into question the translational impact/clinical relevance of the work presented in this manuscript.

Response: Thank you for highlighting the concern regarding the clinical relevance of PIN1 in light of its prevalence in the presented dataset. In our revised manuscript, we have now incorporated an additional dataset derived from a TMA slide that analyzed 160 BC patient samples for PIN1 staining. Our findings from this expanded dataset reaffirm that patients exhibiting high levels of PIN1 protein had a poorer prognosis compared to those with lower PIN1 levels (**Fig. 1b, Supplementary Table 2**). This observation further emphasizes the potential clinical implications and the significant translational impact of PIN1. Moreover, we'd also like to draw attention to our prior research where we found that PIN1 is overexpressed in roughly 70% of human BC tumors in comparison to paired normal tissues (Wulf et al., EMBO J 2001; 20: 3459). Furthermore, other studies conducted by our team indicates that overexpression of PIN1 is associated with unfavorable patient outcomes in human prostate cancer (Ayala et al., Cancer Res 2003; 63 :6244) and in human pancreatic cancer (Koikawa et al., Cell 2021; 184: 4753). We hope this additional data and references provide a more comprehensive perspective on the pivotal role of PIN1 in clinical contexts.

2) Given the low numbers of patients with HER2 homogeneity in each cohort, a power analysis should be included to test whether the sample size provides adequate power for the analyses conducted. A second validation dataset is critical.

Response: Thank you for your insightful comment regarding the sample size. We acknowledge the importance of ensuring that our study has adequate power given the sample sizes in our cohorts. To address this, we conducted a power analysis under the Cox Proportional-Hazards Model using the 'powerCT' function from the 'powerSurvEpi' package in R. Given our sample sizes of 56 patients in the PIN1-low group and 9 patients in the PIN1-high group, and a hazard ratio of 4.56, our analysis yielded a power of approximately 99.6% at a 0.05 significance level. This high power suggests that, despite the low number of patients in each cohort, our study has a very strong capability to detect a true difference in survival curves between the two groups. In conclusion, while we recognize the inherent limitations of small sample sizes, our power analysis assures us that our sample size offers robust power for the analyses conducted.

3) It is likely that CDH1 is the sole E3 ligase responsible for PIN1 degradation. In Extended data figure 1 (K-T) none of the knockdown constructs are validated to show knockdown of the intended target. If the authors claim that CDH1 is the sole, or primary E3 ligase for PIN1, these experiments should be conducted.

Response: We appreciate the reviewer's emphasis on the importance of validating the knockdown constructs. We indeed validated the knockdown efficiency for each construct.

Initially, we opted not to include these validation data due to concerns about overcrowding the figure. However, recognizing the significance of this data for a comprehensive understanding, we've now reorganized the figure panels in the revised manuscript to incorporate the knockdown validation results (**Extended Data Fig. 2**).

4) In Figure 4, the authors find that CDK2 IPs with CDH1 in addition to CDK4. Furthermore, Palbo has been found to modulate CDK2 (Pack et al. 2021). Therefore, it seems possible that the effects of palbo treatment might be modulated by inhibition of CDK2 in addition to CDK4. The authors should acknowledge this as a limitation and be mindful of this possibility when interpreting their results, for example changes in CDH1-phosphorylation.

Response: We appreciate the reviewer's insightful observation regarding the potential role of CDK2 in CDH1 modulation. Indeed, our data from Fig. 4 suggests an interaction between CDK2 and CDH1, in addition to CDK4. Taking into account the potential of Palbociclib to modulate CDK2, it is plausible that Palbociclib's impact on CDH1 activity could also be mediated through CDK2.

However, it's important to underscore the prevalence of Cyclin D1 amplification in a significant portion of human breast cancers, which robustly activates CDK4. This underlines the primary role of CDK4, rather than CDK2, in driving breast cancer cell proliferation (Yu et al., Cancer Cell. 2006; 9: 23). To shed more light on the specific involvement of CDK4 in regulating APC/C^{CDH1}, we conducted the CDK2 knockout experiment. Our findings revealed that the ablation of CDK2 didn't block but slightly augmented the activation of APC/C^{CDH1} when treated with the CDK4 inhibitor (**Extended Data Fig. 7c**). This supports that the CDK4 inhibition by Palbociclib remains more influential than any non-catalytic inhibitory effect on CDK2. We acknowledge this as a potential limitation of our study. In the revised manuscript, we have highlighted this point and emphasized the need for caution when interpreting results, particularly when assessing changes in CDH1 phosphorylation.

5) In several instances throughout the manuscript, the authors state that there are potent or significant changes. Most instances have adequate statistical information, but there are several important exceptions that should be included for Figures: 6i, Ext. data 3b, Ext. data 5k, and Ext. data 8c (heatmap units).

Response: Thank you for highlighting the instances where the necessity for detailed statistical information was overlooked.

Regarding Fig. 6i: we have now quantified the fluorescent intensity of PIN1 and Ki67 in each group and subjected the data to unpaired t tests between two groups. The results have been incorporated into the revised manuscript (**Extended Data Fig. 9c**).

For Extended Data 3b and Extended Data 5k (old version): We have conducted a quantification of band intensities across replicates in the western blots. These quantifications and statistical tests have been integrated into the revised manuscript (**Extended Data Fig. 5b, 7k**).

For Extended Data 8c (old version): We've included the enrichment plots for the top 5 gene sets in the heatmap, accompanied by their respective normalized enrichment scores and p-values, thereby providing a statistically substantiated representation (**Extended Data Fig. 10f**).

6) If Pin1 irreversibly inhibits Cdh1 (such that phosphatases cannot recognize and dephosphorylate Cdh1), then how is Cdh1 thought to be able to inhibit Pin1? Is the idea that new Cdh1 would need to be synthesized? The mechanistic aspect should be more clearly articulated.

Response: Thank you for bringing attention to the apparent complexity and the need for clear articulation regarding the mechanistic interplay between PIN1 and CDH1. We acknowledge the importance of elucidating this aspect in a clear and coherent manner. Our model suggests a dynamic and conditional interaction between PIN1 and CDH1, structured in a domain-oriented and phosphorylation-dependent manner, which we shall clarify here:

- 1. In the context of phosphorylated CDH1** (occurring at the G1/S transition), the WW domain of PIN1 engages with phosphorylated CDH1, catalyzing an isomerization from *trans* to *cis* of the pS163-P motif in CDH1. This isomerization shields CDH1 from dephosphorylation, thus maintaining APC/C in an inactive state which subsequently facilitates S-phase entry.
- 2. Conversely, in situations where CDH1 is unphosphorylated** (observed during the G1 phase or predominantly when PIN1 and CDK4 are inhibited, reducing CDH1 phosphorylation), CDH1 becomes functionally active. The substrate-binding domain of CDH1 recognizes the D-box motif within the PPlase domain of PIN1. This interaction navigates PIN1 towards ubiquitination and subsequent proteasomal degradation, preventing S-phase entry.

7) As a point of experimental rigor, there are some experiments in which SUM159 cell lines are used instead of the cell lines that have been used for most major results presented in this work. The use of these cells for specific experiments, and not more generally, should be explained.

Response: Thank you for your insightful query regarding the application of SUM159 cells in specific experiments within our study. We acknowledge the importance of consistency and experimental rigor across studies. Both MDA-MB-231 and SUM-159 are

representative models for TNBC. SUM-159, in particular, is a widely recognized cell line for exploring stem cell-like attributes and evaluating drug resistance within TNBC. By employing both MDA-MB-231 and SUM-159 in some in vitro phenotypic experiments not only strengthens our findings but also provides a more comprehensive perspective, ensuring the adaptability and relevance of our drug regimens across diverse TNBC subtypes.

Minor Concerns:

1) There appears to be cross-over effects with the RNAi knocking down CDH1 and Cdc20 in Fig 1. Rescue experiments would help clarify the specificity of knockdown better.

Response: We appreciate the reviewer's insightful comment and have conducted additional experiments to address the concern. There were no cross effects in our shRNA-mediated knockdown of CDH1 and CDC20 respectively, which is shown in the untreated cells. Firstly, in our untreated cells, we found no evidence of cross-effects in the shRNA-mediated knockdown of CDH1 and CDC20 (**Fig. 1f**). However, we acknowledge the fact of PIN1 inhibitors affecting CDC20 levels in CDH1 knockdown cells, given that CDC20 is known to be one of the substrates of the APC/C^{CDH1}. To investigate this further and to ensure the specificity of our knockdown, we performed experiments as suggested by the reviewer. Specifically, we examined PIN1 protein levels after treating cells with PIN1 inhibitors in CDC20 and CDH1 overexpression conditions respectively. Our results reveal that CDH1 overexpression enhanced the PIN1 inhibitor-induced PIN1 degradation (**Extended Data Fig. 1k**). This observation proves the specificity of CDH1 in regulating PIN1.

2) Survival curve y-axes need more specific labeling, and better explanation in the text. It is also interesting that PIN1 mRNA levels appear to be non-correlated with survival whereas protein levels are for this particular dataset. Are the survival curves for Figure 1a (n=65) and Extended figure 1a (n=59) from the same patient dataset? If so, why are the number of total patients represented different between the two and are the 9 patients represented in the mRNA survival curves represented in the protein survival curves?

Response: As the reviewer suggested, we changed the y-axis label to "Probability of Survival", indicating the likelihood that individuals under study will survive up to a certain time point. A higher value on the y-axis indicates a higher probability of survival, while a lower value corresponds to a lower probability of survival."

The survival curves for Fig. 1a (PIN1 protein, n=65) and Extended Data Fig. 1a (PIN1 mRNA, n=59) indeed originate from the same patient dataset (Tang et al., *Genome Med* 2018; 10: 94). To address the difference in the number of total patients represented, we would like to explain that while there was a total of 65 tumor samples available in the

dataset, only 59 of them had complete PIN1 mRNA data available for analysis. This discrepancy in sample sizes arises from the variability in data availability for different measurements within the patient cohort. Upon further examination of the dataset, we found that out of the nine tumor samples represented in the high PIN1 protein group in the protein survival curve, eight of them were also included in the mRNA survival curve. Thus, this discrepancy does not significantly impact the outcome of the mRNA survival curve analysis, as the primary trend and conclusions remain consistent.

3) Figure 1f has labeling inconsistencies.

Response: We appreciate the reviewer highlighting the labeling inconsistencies in Fig. 1e (revised version). For clarification, we utilized uppercase labels such as "CDH1" and "PIN1" to denote proteins in human cells, while mixed case labels like "Cdh1" and "Pin1" were used to indicate proteins in mouse cells.

4) For Figure 1g, it looks like KD of CDC20 was not very robust in MDA cells and that it did partially rescue PIN1 levels. This should be acknowledged in the text.

Response: Thank you for the reviewer's insightful observation. It's important to note that CDC20 is essential for cellular function, and a complete knockdown would be detrimental to cell survival. Therefore, we further assessed PIN1 protein levels in CDC20 or CDH1 overexpressed cells after treatment with PIN1 inhibitors. Our data confirms that CDH1 overexpression amplifies the degradation of PIN1 induced by PIN1 inhibitors (**Extended Data Fig. 1k**). We have now articulated this point more clearly in the revised manuscript.

5) In the interpretations for Extended data figure 2f, the authors indicate that Cdh1-KO does not affect Pin1 mRNA levels. However, there appears to be a clear and relatively consistent decrease in Pin1 mRNA levels for the Cdh1-KO cells compared to WT.

Response: We appreciate the reviewer's attention to the detail regarding Extended Data Fig. 3g (revised version). In light of Reviewer's observation, we've revised our interpretation in the manuscript. Now, it states, "*CDH1* KO led to stabilization of PIN1 and other mitotic proteins, whereas it showed modest reductions in their mRNA levels." We believe this offers a more accurate representation of the data.

6) The serum starvation does not look as strong for the Cdh1-KO MEF cell line. Are there differences in the arresting efficiencies between WT and Cdh1-KO cells?

Response: Thank you for raising this insightful observation regarding the arresting efficiencies between the WT and Cdh1-KO MEF cells under serum starvation conditions. In the absence of CDH1:

1. Accumulation of Cell Cycle Proteins: Cdh1-KO MEFs have an accumulation of cell cycle proteins. This accumulation prompts these cells to transition from the G1 to S phase, even in the absence of the growth signals typically provided by serum (**Fig. 1i**,

j, Extended Data Fig. 3e).

2. **Reduced Quiescence:** One of the primary effects of serum starvation is to induce a quiescent state in cells. Without the activity of APC/C^{CDH1}, Cdh1-KO MEFs wouldn't maintain this quiescence as effectively as WT MEFs. As a result, Cdh1-KO MEFs show a weakened response to serum starvation, leading to the observation that the starvation does not appear as pronounced in the Cdh1-KO MEF cell line as compared to the WT (**Fig. 1i, j, Extended Data Fig. 3e**).

7) It is unclear whether or not the cells depicted in Figure 2c-d express a WT allele of CDH1 in addition to the 7A mutant copy. Are the effects of the 7A mutant a result of overexpression of CDH1 in general or is it unique to over expression of the 7A mutant?

Response: Thank you for highlighting the ambiguity regarding the expression of CDH1 alleles in Fig. 2c-d. To clarify, the cells depicted in Fig. 2c-d were stably expressed with the CDH1-7A mutant construct or an empty vector control. They do not express CDH1-WT. The observed senescent effects are attributed specifically to the overexpression of the constitutively active CDH1-7A mutant, but not due to the overexpression of CDH1-WT. To better represent this in the figure, we have updated the label from "WT" to "EV" for clarity in the revised version. We apologize for any confusion caused and appreciate your attention to detail.

8) The legend for Figure 2 indicates that there are representative graphs for panel L, but these appear to be missing in the submitted version.6) The antibody used to detect CDH1 phosphorylation in Figure 5a (pSP) should be defined in the methods.

Response: We appreciate the reviewer's attention to details in Figure 2 and Figure 5a. Regarding Fig. 2, Panel I is the figure showing CDH1-7A induced PIN1 ubiquitination. We would like to confirm that this panel was present in the revised version (**Extended Data Fig. 4c**). As for Fig. 5a, to ascertain CDH1 phosphorylation, we employed a pull-down method for CDH1 and assessed its phosphorylation status by using anti-Phospho-MAPK/CDK Substrates (PXS*P or S*PXR/K) antibody. This antibody is phosphoserine-specific, and does not react with phospho-threonine- or phospho-tyrosine-containing proteins. The antibody information is provided in the Reagents and antibodies section.

9) Specifically in reference to Ext. data 4m, the authors should describe the justification for why they believe that CDK inhibitor and PIN1 inhibitors operate in a synergistic manner, rather than additive.

Response: We thank the reviewer for drawing attention to the depiction in Extended Data Fig. 6l (revised version). We acknowledge that the fluorescent images alone might not provide a clear indication of whether the CDK inhibitor and PIN1 inhibitors function synergistically or additively. Thus, we have revised the text and omit "synergistically" from the description.

However, the extensive drug matrix analysis supports our original contention (**Fig. 6a, b, Extended Data Fig. 8**). Here, ZIP synergy scores were computed to elucidate the interaction between the CDK4 inhibitors and PIN1 inhibitors. Typically, a synergy score below -10 suggests an antagonistic interaction; scores ranging from -10 to 10 indicate an additive effect, and scores above 10 are indicative of a synergistic relationship. We believe this analysis provides a robust rationale for our claims regarding synergy between the two inhibitors.

Reviewer #4 (Remarks to the Author):

CDH1 plays an important role in regulating cell cycle entry and thus is a target of interest for a variety of cancers. CDH1 is regulated by proline-directed serine phosphorylation, the outcomes of which are regulated by prolyl isomerase PIN1 activity, making the results of this manuscript of potentially high significance.

I was not able to comprehensively review this manuscript for scientific rigor, so this review will focus exclusively on the NMR spectroscopy and molecular docking studies that establish a model for CDH1 interactions with PIN1. The insights from this work, as reported in Figure 4, make an important contribution to the overall manuscript. The rigor of the presentation could be improved through the consideration of the following specific points.

Response: We are grateful to the reviewer for their thoughtful feedback, especially regarding the NMR spectroscopy studies presented in Fig. 4. Your specific insights on these aspects are invaluable to our work. Herein, we address each of the points you've highlighted in detail.

1. The presentation of key regions from the 15N-HSQC of PIN1 and 13C-HSQC of P164-labeled CHD1 are helpful to the reader but inclusion of the full spectra as an Extended Data figure would also be beneficial for reuse and extended interpretation of the results.

Response: Thank you for the insightful suggestion. We agree with the reviewer's perspective on the value of providing comprehensive spectral data for broader interpretation. Accordingly, we've incorporated the overlay of full 15N-HSQC spectra of PIN1 (shown in blue) and its complex with the CDH1 phosphopeptide (shown in orange) (**Extended Data Fig. 5h**). Additionally, 13C-HSQC (left) and 13C-HSQCOCYSY (right) spectrum of 13C-labeled CDH1-P164 for the proline isomer assignments have been made available in the revised manuscript (**Extended Data Fig. 5j**).

2. The methods do not completely describe how the CSPs presented in Figure 4D

were calculated. There is no indication that the bound-state chemical shifts were completely determined, so one assumes these represent minimal chemical shift perturbations inferred by comparison of the spectra. To remove the perception of investigator bias, the means for selecting the peak assigned to each residue in the bound-state spectrum and the equation used to calculate the reported CSP should be included.

Response: The reviewer is correct that the bound-state chemical shifts were attained by visual comparison. As can be seen in the **Fig. 4g**, the perturbation is specific, and the peak movement can be tracked with high confidence. A sentence describing the methodology and equation used is now added under the Methods section (NOTE: under NMR spectroscopy, "A systematic titration between PIN1 and CDH1 phosphopeptide was performed by acquiring series of 2D-HSQC spectra to attain resonance assignments of the peptide-bound form of PIN1." The equation is already there, so maybe the reviewer missed that part).

3. The CD-HD correlations displayed in Figure 4G are well resolved and therefore reasonable as a source of cis/trans quantitation. On the other hand, the CB and CG resonance positions are far more unique in the trans-proline and cis-proline conformations, making them the fields preferred benchmark for the presence of prolyl isomerization. At a minimum, drawing attention to the relative intensity of these resonances in the Extended Data would be helpful.

Response: We agree with the reviewer on the significance of beta and gamma position peaks, however, under our experimental conditions, those peaks in the complex spectrum suffer from weak PIN1 peaks coming from natural abundance ^{13}C resonances, as shown on the right. Therefore, we have used the delta position, which is very isolated and more reliable. The analysis of peak intensities suggests that the percent ratio

for alpha, beta, gamma and delta positions in the free peptide are 9.81, 9.22, 8.60 and 7.64. These ratios change to "not determined" (due to overlap with water), 27.64, 8.82 and 19.73. The analysis reported in the main figure is based on peak volumes of the delta position.

Reviewers' Comments:

Reviewer #1:

Remarks to the Author:

The authors have bolstered their conclusion and greatly improved the manuscript's quality with additional data. However, certain aspects remain unclear.

A) In Fig. 4d and Extended Data Fig. 5d, it appears that Palbociclib continues to exhibit an effect on cells with mutations in CDH1 phosphorylated at the S163 site. Could this be explained by the phosphorylation of another site in CDH1 that might be targeted by PIN1?

B) Regarding the analysis linking PIN1 mRNA levels and immune response, I still have reservations about its relevance to the overall paper. This concern is amplified by the perceived lack of solidity in the presented data.

1. The authors opted for a transcriptional dataset of 526 samples sequenced using microarray technology, ignoring an available dataset with over 1000 samples sequenced using RNA seq. (also available from the same website).

2. In the extended figure 10H heatmap, only 48 patients out of the total 526 (9% of the cohort) were selected for analysis. Among these, 24 patients exhibited higher normalized Pin1 expression, while 24 exhibited lower expression. Surprisingly, among those with higher Pin1 expression, 8 patients (33%) also show high immune signature enrichment scores, comparable to the lower pin expression group.

Considering these factors, I suggest removing this data to enhance the manuscript's robustness.

Reviewer #2:

Remarks to the Author:

Summary-

This manuscript from Ke et al evaluates a reciprocal relationship between the prolyl isomerase PIN1 and the cell cycle ubiquitin ligase APC/C. PIN1 isomerizes protein around proline residues that are often phosphorylated as S/T residues by proline directed kinases, including the key cell cycle regulatory enzymes CDKs. APC/C is a large multi-subunit E3 that directs the degradation of proteins at mitotic exit and in quiescence. The authors provide evidence that PIN1 might be an APC/C substrate that is degraded by APC/C in combination with its substrate receptor protein Cdh1. Reciprocally, they also suggest that the phosphorylation of Cdh1 at S163 by CDK4/6 leads to a trans-to-cis prolyl isomerization by PIN1, thereby stabilizing phosphorylated Cdh1 protein and inactivating APC/C. Further they show that combining PIN1 inhibitors with CDK4/6i leads improved tumor growth inhibition in both RB deficient and RB proficient tumors models. The data showing synergy between the PIN1 inhibitors and CDK4/6 inhibitors is rather remarkable. In addition, the data suggesting that PIN1 binds and isomerizes CDH1 appears compelling.

In my previous critique, I raised concerns over the ubiquitination and degradation of PIN1 by APC/C during cell cycle progression. This is a major point of the paper, as the title suggests a reciprocal relationship between APC/C and PIN1. However, this concern, to mind, remains unresolved. The abundance of PIN1 oscillates in a very minor way throughout the cell cycle, contrasting with the overwhelming majority of other known cell cycle, APC/C substrates. The mutations used to test PIN1 ubiquitination, as discussed previously, are problematic. They stretch across additional residues than should be needed and are in a highly structured part of the protein. Thus, these mutations could render PIN1 poorly folded or inactive. Changes in ubiquitination of PIN1 are evaluated in vivo using only a Cdh1-7A mutant, which is a hyperactive form of Cdh1 due to its enhanced APC/C binding. This is very likely to arrest cells in G1-phase. This could coincide with the regulation of Pin1 by other E3s or a time when PIN1 is unstable and not well expressed. The authors point out that there is a slight change in PIN1 abundance noted in arrested (G0/G1) non-transformed cells, in Fig 1i. And, that its decrease is partially alleviated by Cdh1 loss. However, in that Figure, PIN1 abundance appears similar to both mitotic cyclins, which are APC/C substrates, and Cyclin E, which is not an APC/C substrate. Thus, this does not resolve in regulation of APC/C.

I also raised a concern about the ability of Pin1 to prevent dephosphorylation of Cdh1. Nor do I believe these concerns have been addressed. The authors have shown that there is clearly less Cdh1-S163 phosphorylation in the PIN1 KO, and more Cdh1-APC/C binding. However, the cells are arrested in G0/G1-phase in the PIN1 KO (see Fig 5B) and G0/G1 arrested cells have inherently low phosphorylation of Cdh1 and high APC/C binding. Thus, whether this is a cause or consequence of PIN1 loss is unclear. I agree that this site appears to be a PIN1 binding site. However, it is unclear that PIN1 prevents S163 dephosphorylation, as is suggested.

I also respectfully disagree with their interpretation of the S163 mutant revealing a key regulatory mechanism for isomerization that underlies APC/C binding. This mutant and the phosphorylation site clearly play an important role in APC/C binding, but this appears independent of PIN1 isomerization activity. As shown in Fig 4k, the S163A mutant shows increased binding to APC/C in PIN1 KO cells (lanes 2 vs 4) and the S163E shows a very significant decrease when in the PIN1 KO cells. Thus, while the site is important, its unclear how critical PIN1 is to the process and for the potential for isomerization to impact this process is unclear.

Overall, my concerns over how differences in cell cycle might influence the results remain significant. Since APC/C activity oscillates strongly during the cell cycle, it is difficult to disentangle direct and indirect impacts of PIN1 on APC/C, vs PIN1 impacts on other proteins and enzymes that alter the cell cycle and subsequently produce a difference APC/C-Cdh1 binding or activity that is a consequence of an altered cell cycle.

Reviewer #3:

Remarks to the Author:

All concerns have been addressed comprehensively.

Reviewer #4:

Remarks to the Author:

The authors have satisfied all of my requests and concerns.

Response to Reviewers' Comments:

We sincerely appreciate further insightful comments by the two reviewers, while the other two reviewers have not any comment. To address the questions raised, we have conducted new experiments suggested, the results of which have been incorporated into the manuscript. Specifically, we have:

1. Investigated the enzymatic activity of PIN1 with the non-degradable D-box RLAA mutation, confirming that the RLAA mutation does not compromise PIN1's enzymatic function (**Extended Data Fig. 4h**).
2. Conducted an *in vitro* ubiquitination assay that conclusively substantiates the direct ubiquitination and degradation of PIN1 mediated by APC/C^{CDH1} (**Fig. 2f**).

These results have significantly solidified the mechanism proposed in this study.

Reviewer #1 (Remarks to the Author):

The authors have bolstered their conclusion and greatly improved the manuscript's quality with additional data. However, certain aspects remain unclear.

Response: The reviewer has been reviewing our manuscript several times. We deeply appreciate the reviewer's contributions to improving our work. The reviewer's continued engagement and insightful feedback have been invaluable in strengthening our conclusions and significantly enhancing the quality of our manuscript. We are committed to addressing these aspects to ensure that our study is as clear and comprehensive as possible.

A) In Fig. 4d and Extended Data Fig. 5d, it appears that Palbociclib continues to exhibit an effect on cells with mutations in CDH1 phosphorylated at the S163 site. Could this be explained by the phosphorylation of another site in CDH1 that might be targeted by PIN1?

Response: The observation made by the reviewer regarding the impact of Palbociclib on cells with CDH1-S163A mutation is insightful. In our *in vitro* kinase assay, we have identified that CDK4 can also phosphorylate T121 site on CDH1, although this occurs to a lesser extent (**Fig. 4b**). However, it is important to note that PIN1 does not primarily bind to this T121 site (**Extended Data Fig. 5g**).

Another important aspect to consider is that Palbociclib's effects on CDH1-S163A mutant cells may not be solely due to direct phosphorylation of CDH1-S163. Palbociclib is known to influence transcriptional levels through its effect on the Rb pathway (**Extended Data Fig. 7i, j**). This could account for some of its observed impact on these mutant cells.

In contrast, cells with the CDH1-S163E mutation showed a marked resistance to the Palbociclib-induced reduction in mitotic proteins, highlighting the specific and significant role of the CDH1-S163 site in mediating the effects of Palbociclib on CDH1 and its downstream cellular processes. These insights collectively suggest that while the CDH1-

S163 site is a major target, the effects of Palbociclib are multifaceted, involving additional phosphorylation sites and transcriptional regulation mechanisms.

B) Regarding the analysis linking PIN1 mRNA levels and immune response, I still have reservations about its relevance to the overall paper. This concern is amplified by the perceived lack of solidity in the presented data.

1. The authors opted for a transcriptional dataset of 526 samples sequenced using microarray technology, ignoring an available dataset with over 1000 samples sequenced using RNA seq. (also available from the same website).

2. In the extended figure 10H heatmap, only 48 patients out of the total 526 (9% of the cohort) were selected for analysis. Among these, 24 patients exhibited higher normalized Pin1 expression, while 24 exhibited lower expression. Surprisingly, among those with higher Pin1 expression, 8 patients (33%) also show high immune signature enrichment scores, comparable to the lower pin expression group.

Considering these factors, I suggest removing this data to enhance the manuscript's robustness.

Response: We sincerely appreciate the reviewer's suggestion. In light of the reviewer's concerns that PIN1 mRNA data might detract from the overall solidity of the manuscript, we have decided to remove this data and concentrate more on reinforcing the core aspects of our study, as suggested.

Reviewer #2 (Remarks to the Author):

Summary-

This manuscript from Ke et al evaluates a reciprocal relationship between the prolyl isomerase PIN1 and the cell cycle ubiquitin ligase APC/C. PIN1 isomerizes protein around proline residues that are often phosphorylated as S/T residues by proline directed kinases, including the key cell cycle regulatory enzymes CDKs. APC/C is a large multi-subunit E3 that directs the degradation of proteins at mitotic exit and in quiescence. The authors provide evidence that PIN1 might be an APC/C substrate that is degraded by APC/C in combination with its substrate receptor protein Cdh1. Reciprocally, they also suggest that the phosphorylation of Cdh1 at S163 by CDK4/6 leads to a trans-to-cis prolyl isomerization by PIN1, thereby stabilizing phosphorylated Cdh1 protein and inactivating APC/C. Further they show that combining PIN1 inhibitors with CDK4/6i leads improved tumor growth inhibition in both RB deficient and RB proficient tumors models. The data showing synergy between the PIN1 inhibitors and CDK4/6 inhibitors is rather remarkable. In addition, the data suggesting that PIN1 binds and isomerizes CDH1 appears compelling.

In my previous critique, I raised concerns over the ubiquitination and degradation of PIN1 by APC/C during cell cycle progression. This is a major point of the paper, as the title suggests a reciprocal relationship between APC/C and PIN1. However, this concern, to mind, remains unresolved. The abundance of PIN1 oscillates in a very minor way throughout the cell cycle, contrasting with the overwhelming majority of other known cell cycle, APC/C substrates. The mutations used to test PIN1 ubiquitination, as discussed previously, are problematic. They stretch across additional residues than should be needed and are in a highly structured part of the protein. Thus, these mutations could render PIN1 poorly folded or inactive. Changes in ubiquitination of PIN1 are evaluated *in vivo* using only a Cdh1-7A mutant, which is a hyperactive form of Cdh1 due to its enhanced APC/C binding. This is very likely to arrest cells in G1-phase. This could coincide with the regulation of Pin1 by other E3s or a time when PIN1 is unstable and not well expressed. The authors point out that there is a slight change in PIN1 abundance noted in arrested (G0/G1) non-transformed cells, in Fig 1i. And, that its decrease is partially alleviated by Cdh1 loss. However, in that Figure, PIN1 abundance appears similar to both mitotic cyclins, which are APC/C substrates, and Cyclin E, which is not an APC/C substrate. Thus, this does not resolve in regulation of APC/C.

Response: We acknowledge the reviewer's continuing concerns regarding the ubiquitination and degradation of PIN1 by APC/C^{CDH1}, which might be due to our failure to describe our results better. To address this concern, we have provided more definitive evidence.

1. Our results demonstrated that the RLAA mutation within the D-box of PIN1 blocked its degradation and ubiquitination by the active APC/C^{CDH1} (**Fig. 2j, k**). To decisively address the concern of the D-box mutations affecting the structure and function of PIN1, we have assessed the enzymatic activity of PIN1 RLAA D-box mutant. The data confirmed that the RLAA mutation maintained the enzymatic activity of PIN1 (**Extended Data Fig. 4h**). This evidence suggested that the resistance of the PIN1 RLAA D-box mutant to degradation was not due to protein misfolding or loss of enzymatic function.
2. In addition, we have presented comprehensive evidence including the observation that CDH1 knockout impedes the degradation of PIN1 (**Fig. 1i, 5g-j, Extended Data Fig. 3e, 7e-h**).
3. To conclusively address the Reviewer's concern about whether APC/C^{CDH1} can directly target PIN1, we have now performed *in vitro* ubiquitination assays. These assays have unequivocally demonstrated that PIN1 can be directly ubiquitinated by APC/C^{CDH1} (**Fig. 2f**).

These results collectively substantiate our conclusion that PIN1 is indeed a substrate for degradation by APC/C^{CDH1}.

I also raised a concern about the ability of Pin1 to prevent dephosphorylation of Cdh1. Nor do I believe these concerns have been addressed. The authors have shown that there is clearly less Cdh1-S163 phosphorylation in the PIN1 KO, and more Cdh1-APC/C binding. However, the cells are arrested in G0/G1-phase in the PIN1 KO (see Fig 5B) and G0/G1 arrested cells have inherently low phosphorylation of Cdh1 and high APC/C binding. Thus, whether this is a cause or consequence of PIN1 loss is unclear. I agree that this site appears to be a PIN1 binding site. However, it is unclear that PIN1 prevents S163 dephosphorylation, as is suggested.

Response: We appreciate the reviewer's concerns regarding the direct versus indirect effects of PIN1 on CDH1 phosphorylation. While it is impossible to completely isolate these effects due to current technological limitations, our experiments aim to elucidate the role of PIN1 as comprehensively as possible.

1. Our experiments demonstrate that the loss of CDH1 significantly affects the outcomes of PIN1 inhibition. Specifically, CDH1 KO largely abolished PIN1 inhibitor-induced mitotic protein degradation (**Fig. 5g, h, Extended Data Fig. 7e-h**) and G0/G1 arrest (**Extended Data Fig. 8g, h**). While there are inherent differences in the basal levels between WT and CDH1-KO cells, the key point of interest lies in the comparative analysis of untreated versus treated conditions in both cells. Here, the ratio of treated-to-untreated in WT cells differs markedly from that in CDH1-KO cells (**Fig. 5g, h, Extended Data Fig. 7e-h, 8g, h**). This differential response underscores the crucial role of CDH1 in mediating the cellular effects of PIN1 inhibition.
2. Additionally, our data demonstrates that treatment with PIN1 and CDK4 inhibitors leads to PIN1 polyubiquitination, which is notably attenuated in CDH1 KO cells (**Fig. 5j**). These results indicate a distinct difference in the response of WT and CDH1 KO cells to PIN1 inhibition, suggesting the critical impacts of PIN1 on APC/C^{CDH1}.
3. We employed NMR and Mass spectrometry to directly study the interaction between PIN1 and CDH1. NMR analysis confirms that PIN1 binds and isomerizes CDH1 at S163 (**Fig. 4f-i**). Furthermore, Mass spectrometry data (**Fig. 4j**) reveal that the absence of PIN1 decreases CDH1-S163 phosphorylation, leading to heightened APC/C activity in CDH1-WT cells, but not in the CDH1-S163A mutant (**Extended Data Fig. 5k**).

Based on the current evidence, it is conceivable to infer that PIN1 has a direct influence on CDH1-S163 although further studies using more advanced techniques are necessary to fully delineate the direct and indirect roles of PIN1 on CDH1 phosphorylation.

I also respectfully disagree with their interpretation of the S163 mutant revealing a key regulatory mechanism for isomerization that underlies APC/C binding. This mutant and the phosphorylation site clearly play an important role in APC/C binding, but this appears independent of PIN1 isomerization activity. As shown in Fig 4k, the S163A mutant shows increased binding to APC/C in PIN1 KO cells (lanes 2 vs 4) and the S163E shows a very significant decrease when in the PIN1 KO cells.

Thus, while the site is important, its unclear how critical PIN1 is to the process and for the potential for isomerization to impact this process is unclear.

Response: Firstly, we would like to clarify a possible misunderstanding regarding the interpretation of the figure labels, for which we sincerely apologize if they were not sufficiently clear. The experiment in question specifically involved CDH1-WT, CDH1-S163A, and CDH1-S163E mutants to assess the role of PIN1 in modulating the CDH1-S163 site and its influence on the interaction between CDH1 and the APC/C complex. The data indicates that **PIN1 knockout (labeled “+” in the figure)** increases the binding of the CDH1 and APC/C complex in CDH1-WT cells (comparing **lanes 1 and 2**). However, this increase in binding is not observed in CDH1-S163A (comparing **lanes 3 and 4**) and CDH1-S163E mutants (comparing **lanes 5 and 6**). This differential response across the wild-type and mutant forms of CDH1 supports our model that PIN1 plays a significant role in modulating the phosphorylation state of CDH1 at the S163 site, which in turn affects its binding to the APC/C complex.

Overall, my concerns over how differences in cell cycle might influence the results remain significant. Since APC/C activity oscillates strongly during the cell cycle, it is difficult to disentangle direct and indirect impacts of PIN1 on APC/C, vs PIN1 impacts on other proteins and enzymes that alter the cell cycle and subsequently produce a difference APC/C-Cdh1 binding or activity that is a consequence of an altered cell cycle.

Response: We realize the difficulty in fully distinguishing the direct effects of PIN1 on APC/C^{CDH1} from its indirect effects within the cell cycle. This challenge is indeed significant, given the intricate nature of cellular processes and the oscillatory behavior of APC/C^{CDH1} activity throughout the cell cycle.

In our study, while we strive to elucidate these complex interactions, our primary focus has been on the translational applications and clinical significance of our findings, particularly in breast cancer therapy. Our methodologies, though they might not completely isolate the direct and indirect impacts of PIN1 on APC/C, have nonetheless provided valuable insights that advance our understanding of the role of PIN1 in regulating APC/C activity. Importantly, we have provided extensive evidence of the reciprocal regulation of PIN1 and APC/C^{CDH1} in controlling mitotic protein stability and cell cycle entry. Most significantly, we have further demonstrated how the dysregulation of this mechanism leads to unchecked proliferation and cancer, and how it can be effectively targeted by combining the approved PIN1 and CDK4 inhibitors. This novel therapeutic strategy effectively reactivates APC/C^{CDH1}, leading to the degradation of PIN1 and other critical mitotic proteins, thereby inducing permanent cell cycle arrest and stimulating anti-tumor immunity. These translate into synergistic efficacy against RB-proficient and RB-deficient triple-negative breast cancer, paving the way for new clinical trials to evaluate their clinical impact on patients with this deadly disease.

Reviewers' Comments:

Reviewer #1:

Remarks to the Author:

Authors addressed the concerns

Reviewer #2:

Remarks to the Author:

The authors have largely addressed my concerns. I have no additional comments.